

# Biogeochemistry and Physics of the Southern Ocean-Atmosphere System Explored With Data Science

Sebastian Landwehr[1,13], Michele Volpi[2], F. Alexander Haumann[3,4], Charlotte M. Robinson[5], Iris Thurnherr[6,7], Valerio Ferracci[8], Andrea Baccarini[1,13], Jenny Thomas[9], Irina Gorodetskaya[10,11], Christian Tatzelt[12], Silvia Henning[12], Rob L. Modini[13], Heather J. Forrer[14,15], Yajuan Lin[16,17,18], Nicolas Cassar[16,17], Rafel Simó[19], Christel Hassler[20,9], Alireza Moallemi[13], Sarah E. Fawcett[14], Neil Harris[8], Ruth Airs[21], Marzieh H. Derkani[22], Alberto Alberello[23], Alessandro Toffoli[22], Gang Chen[13], Pablo Rodríguez-Ros[19], Marina Zamanillo[19], Pau Cortés-Greus[19], Lei Xue[24], Conor G. Bolas[25], Katherine C. Leonard[11,26], Fernando Perez-Cruz[2], David Walton[4,†], and Julia Schmale[1,13]

[1]School of Architecture Civil and Environmental Engineering, École Polytechnique Fédérale de Lausanne, Switzerland
[2]Swiss Data Science Center, ETH Zurich and EPFL, Switzerland
[3]Atmospheric and Oceanic Sciences Program, Princeton University, Princeton, NJ, USA
[4]British Antarctic Survey, Cambridge, UK
[5]Remote Sensing and Satellite Research Group, School of Earth and Planetary Sciences, Curtin University, Kent Street, Bentley, WA 6102, Australia
[6]Institute for Atmospheric and Climate Science, ETH Zurich, Switzerland
[7]Geophysical Institute, University of Bergen, and Bjerknes Centre for Climate Research, Bergen, Norway
[8]Centre for Environmental and Agricultural Informatics, School of Water, Energy & Environment Cranfield University, College Road, Cranfield MK43 0AL, Bedfordshire
[9]Swiss Polar Institute, Switzerland
[10]Centre for Environmental and Marine Studies, Department of Physics, University of Aveiro, Aveiro, Portugal
[11]CRYOS, School of Architecture, Civil and Environmental Engineering, École Polytechnique Fédérale de Lausanne, Switzerland
[12]Leibniz Institute for Tropospheric Research, Leipzig, Germany
[13]Laboratory of Atmospheric Chemistry, Paul Scherrer Institute (PSI), 5232 Villigen PSI, Switzerland
[14]Department of Oceanography, University of Cape Town, 7701, Cape Town, South Africa
[15]Earth, Ocean and Atmospheric Science Department, Florida State University, Tallahassee, FL, USA, 32306
[16]Division of Earth and Climate Sciences, Nicholas School of the Environment, Duke University, Durham, USA
[17]CNRS, Univ Brest, IRD, Ifremer, LEMAR, F-29280 Plouzané, France
[18]Duke Kunshan University, China
[19]Institut de Ciències del Mar (ICM-CSIC), Barcelona, Catalonia, Spain
[20]Department F.-A. Forel for Environmental and Aquatic Sciences, University of Geneva.
[21]Plymouth Marine Laboratory
[22]Dept. Infrastructure Engineering, Faculty of Engineering and Information Technology, The University of Melbourne, Melbourne, Australia
[23]Graduate School of Frontier Sciences, The University of Tokyo, Kashiwa, Japan
[24]Department of Chemistry, College of Environmental Science and Forestry, State University of New York, Syracuse, NY, USA
[25]ITOPF
[26]CIRES, University of Colorado Boulder, USA
†deceased, 12 February 2019

**Correspondence:** Sebastian Landwehr (sebastian.landwehr@gmail.com) and Julia Schmale (julia.schmale@epfl.ch)





**Abstract.** The Southern Ocean is a critical component of Earth's climate system, but its remoteness makes it challenging to develop a holistic understanding of its processes from the small to the large scale. As a result, our knowledge of this vast region remains largely incomplete. The Antarctic Circumnavigation Expedition (ACE, austral summer 2016/2017) surveyed a large number of variables describing the dynamic state of the ocean and the atmosphere, the freshwater cycle, atmospheric chemistry, ocean biogeochemistry and microbiology. This circumpolar cruise included visits to twelve remote islands, the marginal ice zone, and the Antarctic coast. Here, we use 111 of the observed variables to study the latitudinal gradients, seasonality, shorter term variations, the geographic setting of environmental processes, and interactions between them over the duration of 90 days. To reduce the dimensionality and complexity of the dataset and make the relations between variables interpretable, we applied a sparse Principal Component Analysis (sPCA), which describes environmental processes through 14 latent variables. To derive a robust statistical perspective on these processes and to estimate the uncertainty in the sPCA decomposition, we have developed a bootstrap approach. We identified temporal patterns from diurnal to seasonal cycles, as well as geographical gradients and "hotspots" of interaction. Our results establish connections of oceanic, atmospheric, biological and terrestrial processes in an innovative way, while confirming many well known relations of the Southern Ocean system. More specifically, we identify: the important role of the oceanic circulations, frontal zones, and islands in shaping the nutrient availability that controls biological community composition and productivity; that sea ice predominantly controls sea water salinity, dampens the wave field, and is associated with increased phytoplankton growth and net community productivity possibly due to iron fertilization and reduced light limitation; and clear regional patterns of aerosol characteristics emerged, stressing the role of the sea state, atmospheric chemical processing, as well as source processes near "hotspots" for the availability of cloud condensation nuclei and hence cloud formation. A set of key variables and their combinations, such as the difference between the air and sea surface temperature, atmospheric pressure, sea surface height, geostrophic currents, upper ocean layer light intensity, surface wind speed and relative humidity, played an important role in the majority of latent variables, highlighting their importance for a large variety of processes and the necessity for Earth System Models to represent them adequately. In conclusion, our study highlights the use of sPCA to identify key ocean-atmosphere interactions across physical, chemical, and biological processes and their associated spatio-temporal scales. The sPCA processing code is available as open-access and we believe that our approach is widely applicable to other environmental field studies.



# 1 Introduction

The Southern Ocean plays an important role in Earth's climate. Comparisons of climate models suggest that the 30% of the global ocean surface south of 30°S strongly mitigates global surface warming. The region accounts for about 43% of the uptake of anthropogenic $CO_2$ (labeled as (a) in Figure 1) and 75% of the excess heat uptake by global oceans (Frölicher et al., 2014). This substantial uptake of excess heat and $CO_2$ is due to the formation of large volumes of subsurface waters by subduction in this region (Figure 1b), accounting for around 65% of all global ocean subsurface water (DeVries, 2014). The Southern Ocean is not only responsible for the subduction of water masses, it is also the region where the deepest ocean waters return to the surface, accounting for about 80% of the resurfacing of North Atlantic Deep Water and Antarctic Bottom Water (Talley, 2013) (Figure 1c). Through this upwelling, it provides the surface ocean with important macronutrients such as dissolved nitrate, phosphate, and silicate (Marinov et al., 2006). However, at the surface of the Southern Ocean itself, the consumption of these macronutrients through biological production is incomplete, due to the limited availability of iron, which determines the efficiency of the so-called biological carbon pump in this region (Tagliabue et al., 2017) (Figure 1d). The unused macronutrients are then exported to lower latitudes, where they have been estimated to fuel about 75% of the global ocean biological production (Sarmiento et al., 2004). While this overall important role of the Southern Ocean in the global climate system (through its ocean circulation and influence on the carbon and energy cycle) is widely accepted, very little is known about the local interactions between its components, i.e. the atmosphere, biosphere, sea ice, land and ocean.

The exchange of heat, freshwater, momentum, chemical species (e.g. in the form of trace gases such as $CO_2$, DMS, and aerosols) in the Southern Ocean is determined by the complex interaction of the atmosphere, ocean, land, ice, and the microbial communities (Figure 1). Between 40°S and 60°S the strong westerly wind-belt develops nearly unhindered by land masses (Figure 1e), making the Southern Ocean the stormiest and fiercest ocean of the world (Hanley et al., 2010; Derkani et al., 2020). Winds and waves strongly modulate ocean mixing (Thorpe, 2007; Toffoli et al., 2012), biological production (Nicholson et al., 2016; Uchida et al., 2020), air-sea gas exchange (Wanninkhof et al., 2009; Gruber et al., 2019), sea ice dynamics (Alberello et al., 2020; Vichi et al., 2019) and sea spray emission (Figure 1f), relevant for cloud formation (Schmale et al., 2019; Quinn et al., 2017; Bigg, 1973). This storm track region is characterised by the frequent passage of extratropical cyclones leading to the formation of a quasi-persistent low-level cloud deck and regular precipitation (Figure 1g) (Catto et al., 2012). Clouds can only form and persist when cloud condensation nuclei (CCN) and ice nucleating particles (INP) are present. Here, interactions between microbial activity and atmospheric chemistry come into play. Trace gas emissions from phytoplankton blooms can grow particles into the CCN size range (Figure 1h) (Charlson et al., 1987; Pierce and Adams, 2006; Korhonen et al., 2008; Hoffmann et al., 2016; Schmale et al., 2019). The ocean-microbiology-atmosphere-cloud interactions strongly affect the radiation balance (Figure 1i) and hydrological cycle of the region (Figure 1j) (Vergara-Temprado et al., 2018). To date, uncertainties in these processes contribute to biases in simulated cloud presence and lifetime, which lead to an overprediction of the ocean heat uptake in global climate models of up to $30\,\mathrm{W\,m^{-2}}$ with implications on the regional energy balance, momentum transport, and ocean dynamics (Trenberth and Fasullo, 2010; Flato et al., 2013).





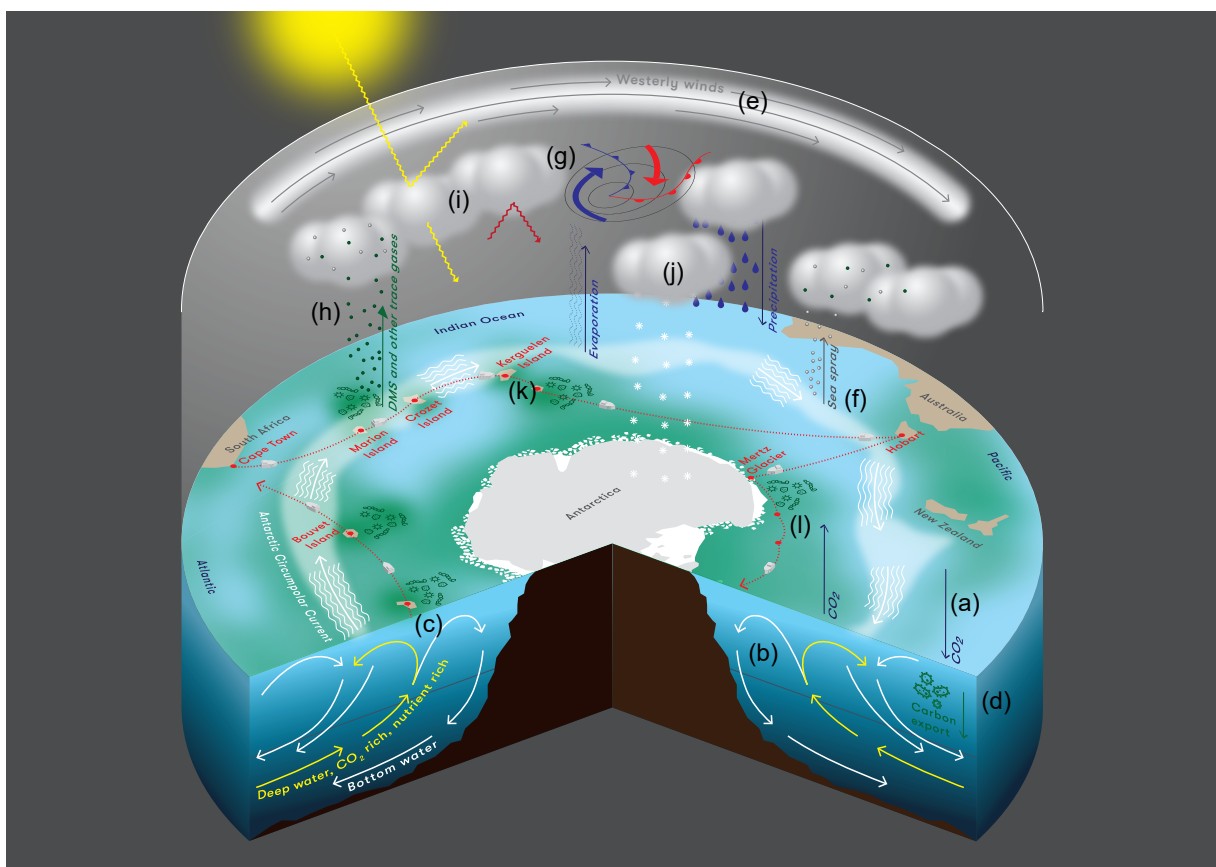

**Figure 1.** Conceptual illustration of selected Southern Ocean processes. The red dashed line represents the ACE cruise track. Letters indicate processes described in more detail in the introduction: (a) $CO_2$ uptake, (b) formation of bottom water, (c) upwelling of nutrient rich water, (d) biological carbon pump, (e) westerly storm track, (f) formation of sea spray, (g) cyclone activity and low-level cloud deck, (h) emission of microbial gases and secondary aerosol formation, (i) cloud modulated radiation budget, (j) evaporation and precipitation, (k) nutrient (iron) rich areas near islands through the island mass effect, (l) meltwater inducing phytoplankton blooms.

The presence of continents, islands, sea ice, and glaciers, can strongly modify the local atmosphere-biosphere-ocean inter-actions, especially through their influence on ocean circulation and stratification, the marine boundary layer, and the biological productivity. Islands can interrupt the zonal flow in the atmosphere and ocean leading to horizontal and vertical mixing and transport (Rintoul, 2018). For example, "hotspots" of carbon and nutrient upwelling have been identified in close proximity to shallow ocean topography or islands (Figure 1k) (e.g. Tamsitt et al., 2018). The supply of iron from sediments or deep water

can lead to "hotspots" of biological production in the close proximity of land masses and topographic features (e.g. Atkinson et al., 2001; Blain et al., 2007; Prend et al., 2019), which is known as the island mass effect (IME). Sea ice in the Southern Ocean forms and melts seasonally and alters the surface ocean stratification (Haumann et al., 2016), gas and heat exchange (Butterworth and Miller, 2016; Swart et al., 2019), and causes spring-time blooms (l) (Uchida et al., 2019; Arteaga et al.,





2020; Moreau et al., 2020). Due to the large number of processes and their different spatial and temporal scales, a direct quan-

tification of the processes that determine atmosphere-biosphere-ocean-ice-land interaction is challenging and requires large interdisciplinary datasets as well as new tools to analyze their covariance.

In a single summer season the Antarctic Circumnavigation Expedition (ACE) (Walton and Thomas, 2018; Schmale et al., 2019) covered all three Southern Ocean basins including visits to twelve remote islands, the marginal sea ice zone, and the Antarctic coast (Figure 2). The in situ observations cover a wide range of variables related to the dynamic state of the ocean

and the atmosphere, the freshwater cycle, atmospheric chemistry, ocean biogeochemistry and microbiology. The dataset covers a large range of environmental conditions and process time scales and provides a unique opportunity for an interdisciplinary study to better understand the complex Southern Ocean system and to identify relations between physical, chemical, and biological processes. Studying the above requires not only an interdisciplinary dataset, but also analytical tools capable of capturing the relations between the large number of original variables (OVs, i.e. observed and derived variables), which vary

over different spatial and temporal scales. In this work, we explore how a sparse matrix factorization approach, i.e. sparse principal component analysis (sPCA), can connect these highly heterogeneous observations.

Standard principal component analysis (PCA) (Hotelling, 1933) is a fundamental data analysis tool in many disciplines in the natural and environmental sciences. Also known as empirical orthogonal function (Denbo and Allen, 1984) in climate science and meteorology, it is mainly used to reduce the dimensionality of a dataset, as a preprocessing step for further analyses and/or

for visualization (Demsar et al., 2013). Observations are treated as points in a multidimensional space with each dimension representing an OV. The PCA rotates the input data so that the new axes, the principal components, are aligned to the direction of maximal variance. These dimensions are also known as latent variables (LVs). However, despite its value in providing uncorrelated LVs and summarizing data variance in a few principal components, PCA decompositions are hard to interpret, due to the potentially large number of input variables that contribute to the definition of the LVs (i.e. OVs, which have non-

zero entries in the weight matrix). Common attempts to interpret PCA relate the direction of the principal components to the input variables, and assign them a user-defined meaning. Since it is difficult to do so for more than two or three dimensions, typically only a few dominant LVs are analysed, while LVs explaining only marginally the total variability are neglected. A caveat is also that PCA tends to fail when the number of OVs is very large, or larger than the number of data points (Zou et al., 2006). These challenges are alleviated when using sPCA, where only a subset of the most informative OVs is used to construct

each LV, while the remaining weights are forced to be zero, resulting in a sparse weight matrix. The information, which OVs contribute to each LV, and in particular which do not, greatly simplifies the interpretation of the LVs. Sparse PCA has found notable applications, amongst others, in genomics (Lee et al., 2010), ecology (Gravuer et al., 2008), biology (Li et al., 2017) and neuroscience (Baden et al., 2016). In all settings, sPCA can be used as a drop-in replacement to standard PCA, leading to decompositions with much-improved interpretability. In this work, we extend the sPCA framework by empirically estimating a

bootstrapped distribution of sPCA weights and corresponding LVs, providing confidence intervals around measures of interest. We apply sPCA to a dataset of 111 OVs from the Antarctic Circumnavigation Expedition and choose to obtain 14 LVs, which connect observations across the disciplines involved and serve as a basis for the exploration of processes described by the data.

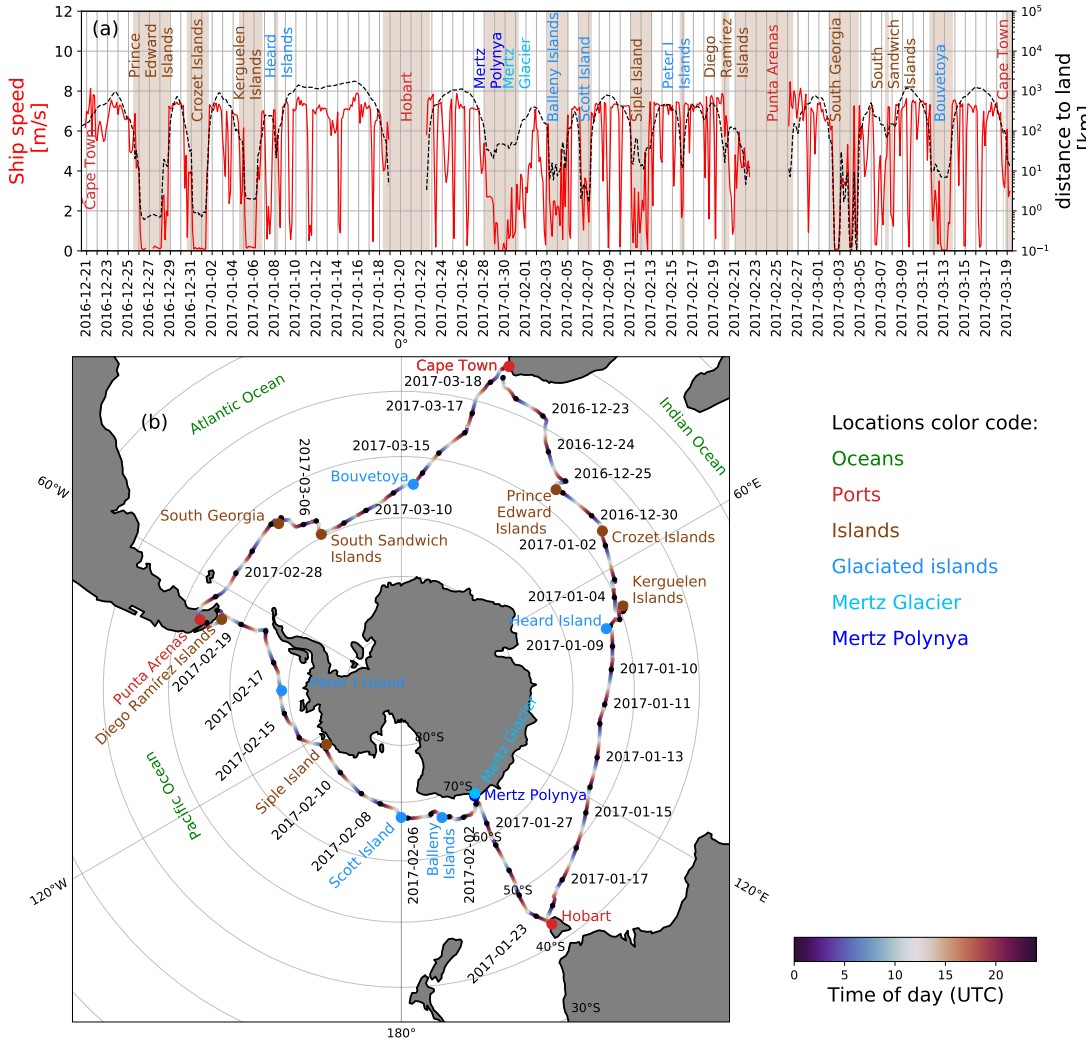

**Figure 2.** (a) Time series of the ship's speed (red solid line, left axis) and distance to land (black dashed line, right axis). Periods where the ship was within 100 km of the nearest land are indicated by brown shading, with the names of the locations provided; (b) Map of the ACE cruise track. Places visited by the ship (ports, islands, Mertz Glacier and polynya) are marked and labelled with coloured bullets and text. The black dots and date ticks refer to the ship's position at midnight of the date specified (UTC) and time of UTC day is further indicated by the colour scale. Overlapping date ticks have been omitted for clarity.





Section 2 briefly describes the datasets used in this study. We elaborate on the sPCA approach in section 3 and introduce the resulting LVs and their weights in section 4. In Section 5, we discuss the individual LVs and relate them to processes. In section 6, we discuss the LVs in conjunction with regional phenomena and geographical "hotspots" and provide a synthesis of the findings. Our conclusions are provided in section 7.





## 2 Data

### 2.1 The Antarctic Circumnavigation Expedition

The Antarctic Circumnavigation Expedition (ACE) (Walton and Thomas, 2018; Schmale et al., 2019) fully circumnavigated
the Antarctic continent between 34°S and 78°S aboard the RV Akademik Tryoshnikov during a single austral summer from 20
December 2016 to 18 March 2017. Figure 2a shows the time series of the ships speed and distance to land and Figure 2b shows
a map of the cruise track. Leg 1 of the expedition started in Cape Town, South Africa, from where the ship travelled through the
Indian Ocean sector of the Southern Ocean to Hobart, Australia. Leg 1 featured open ocean conditions with rough seas and the
highest wind speeds encountered during the expedition. The expedition visited three remote islands: Marion Island (one of the
Prince Edward Islands), the Iles Crozet, and Kerguelen Island during leg 1. During leg 2, from Hobart through the Pacific sector
of the Southern Ocean to Punta Arenas, Chile, the ship stayed mostly close to the Antarctic continent spending several days at
the Mertz Glacier and explored the area which was previously covered by the glacier tongue which had broken off in February
2010 (Campagne et al., 2015). The expedition then visited the Balleny Islands, Scott Island, Siple Island, as well as the Marie
Byrd Land coast, which was possible due to the unusual lack of sea ice in this region, before progressing to Peter I Island.
Sailing northward along the west coast of the Antarctic Peninsula and across the Drake Passage, the expedition visited the
Diego Ramírez Islands before reaching Punta Arenas. During leg 3, the ship returned through the South Atlantic back to Cape
Town and visited South Georgia, the South Sandwich Islands, and Bouvetøya Island. In addition to these numerous sites, the
expedition encompassed all the biogeochemical regimes of the Southern Ocean (Janssen et al., 2020), including the subtropical
and subantarctic zone as well as regions south of the polar front, representing vast regions where primary productivity is known
to be largely limited by the micronutrient iron. Much of the time was also spent along the Southern Ocean storm track between
40°S and 60°S, with frequent passages of cyclones (Simmonds et al., 2003; Papritz et al., 2014). This region of cyclones
is also responsible for a limited influence of anthropogenic air pollutants from the inhabited continents, making the visited
region one of the most pristine atmospheric regimes on Earth (Hamilton et al., 2014; Schmale et al., 2019). The expedition also
offered the possibility of studying the effect of weather systems on sea spray aerosol and trace gas concentrations in one of the
fiercest environments on Earth (Hanley et al., 2010; Landwehr et al., 2020). Altogether, ACE provides a unique picture of the
heterogenous environmental conditions that characterize the Southern Ocean.

### 2.2 Original variables and categories

In this section, we provide a short overview of the observed and derived variables, which are referred to as original variables
(OVs). To present the OVs and to facilitate interpretation of the results, we assigned the variables to eight summary categories
based on processes and phenomena they are associated with. The categories were defined by consensus among the authors
and are intended to orient the reader, and are used for a more generalized presentation of the correlation structures in the ACE
datasets, which have been identified by the sPCA analysis. To facilitate this, a colour is assigned to each category. Table 1 lists
the categories and provides the symbols of the variables contained in each category. The OVs themselves are summarised in
the Appendix tables Table A3 to Table A11, which provide the OV-ID, the variable symbol, SI units, and a description of the



variable as well as the DOI of the published dataset or a reference to the supplementary information section S1, which provides

details on measurement methods and variable derivations. Further, we provide a glossary for each OV, i.e. a brief description

for a quick look-up, in the Appendix section E. We assigned an ID to each of the 111 variables, e.g., wind speed ($u_{10N}$) is the

original variable 1 (OV1). Both symbols and OV-IDs are used in the figures.





**Table 1.** Categories used to classify the original variables. The columns provide the name of the category and the abbreviation, the colour which is used in the figures, and a list of the OVs which were assigned to the category. The OV symbols are explained in Appendix tables A1 to A11 (see Appendix section A).

| Category name (Abbreviation) | Colour | Variables in the category |
|---|---|---|
| Atmospheric dynamics and thermodynamics (Atm. dyn.) | | $u_{10N}$; $T_{air}$; $S_{in}$; $P_{air}$; PAR; mask$_{CW}$; mask$_{cyc}$ |
| Atmospheric side of the hydrological cycle (Atm. hydro.) | | visibility; CL; SC; RH; $\delta^{18}O_{vap}$; $\delta^2H_{vap}$; $dexc_{vap}$; $w$; RR; SR; HHF |
| Atmospheric chemistry (Atm. chem.) | | $CH_4$; $CO_2$; CO; $O_3$; $N_{CCN,0.15}$; $N_{CCN,0.30}$; $N_{CCN,1.00}$; $\kappa_{CCN,0.15}$; $\kappa_{CCN,0.30}$; $\kappa_{CCN,1.00}$; $H_2SO_4$; $HIO_3$; MSA; $SO_4^{2-}$; $Cl^-$; $N_{INP,LV,-8}$; $N_{INP,LV,-20}$; $N_{INP,HV,-8}$; $N_{INP,HV,-20}$; $r_{fluo,fine3\sigma}$; $r_{fluo,coarse3\sigma}$; $N_{Aitken}$; $N_{accumulation}$; $N_{seaspray}$; $N_{nitrate,PM10}$; $N_{oxalate,PM10}$; $N_{bromide,PM10}$; $N_{MSA,PM10}$; $N_{sodium,PM10}$; $N_{ammonium,PM10}$; $N_{potassium,PM10}$; $N_{magnesium,PM10}$; $N_{calcium,PM10}$; Isoprene$_{air}$ |
| Oceanic dynamics and thermodynamics (O. dyn.) | | $I_g$; $H_s$; $T_{m-1,1}$; $T_{sw}$; $\sigma_{0,sw}$; SSH; $U_g$; MLD |
| Oceanic side of the hydrological cycle (O. hydro.) | | $S_{sw}$; $C_i$; $\delta^{18}O_{sw}$ |
| Ocean microbial community (O. microb.) | | Chlide $a$; Phaeob $a$; Phaeophy $a$; PSD$_{slope}$; $F_VF_M$; $\Phi_{PSII}{}'$; $\sigma_{PSII}$; $\sigma_{PSII}{}'$; TChl $a$; Chloro; Crypto; Cyano; DiatA; DiatB; Dino; Hapto8; Hapto67; Prasino; Pelago; Chl $a_{fluo}$; $N_{totalbacteria}$; $N_{synechococcus}$; $N_{nanoeukaryotes}$; $N_{picoeukaryotes}$ |
| Ocean biogeochemistry (O. biogeochem.) | | POC; PON; C:N; Nitrate; Nitrite; Phosphate; Silicate; $a_{nap}/a_p$; $a_{napslope}$; $a_{CDOM}$; DMS; CSO; $CS_2$; Isoprene$_{sea}$; $CHBr_3$; $CH_2Br_2$; acrylate; DMSP; TEP; CSP; Ammonium; NCP |
| Topography (Topo.) | | $d_{land}$; $d_{water}$ |



## 3 Methods

### 3.1 Sparse Principal Component Analysis (sPCA)

Sparse PCA was used to allow an easier interpretation of PCA solutions by encouraging weights (also known as principal directions in the PCA literature) to take values of exactly zero (Zou et al., 2006), while optimally summarizing data variance of the original data into a predefined number of components. Maximizing data variance in a finite set of components guarantees minimal information loss in the solutions.

The standard PCA problem is formulated as (Hotelling, 1933):

$$\hat{\mathbf{U}}_{PCA}, \hat{\mathbf{V}}_{PCA} = \min_{U,V} \frac{1}{2}\|\mathbf{X} - \mathbf{U}\mathbf{V}\|_2^2 \ \text{ subject to} \|\mathbf{V}\mathbf{V}^\top\| = \mathbf{I}, \tag{1}$$

which decomposes the original data matrix $\mathbf{X}$ into a set of mutually orthogonal LVs (principal components) $\mathbf{U}$. LVs are a linear combination of the original OVs and the assigned weights $\mathbf{V}$ as $\mathbf{U} = \mathbf{X}\mathbf{V}^\top$, which are usually non-zero. It is therefore hard to assign unequivocally the importance of an OV to the composition of an LV. Note that $\mathbf{X}$ is transformed into standard scores, i.e. centred to 0 mean and reduced to unit standard deviation, before optimization.

Sparse PCA aims at solving a very similar problem but adds an additional constraint favouring some weights for each $i$-th component ($\mathbf{V}_i$) to be exactly zero. Therefore, only a subset of the OVs contributes to the estimation of the corresponding LV $\mathbf{U}_i$. In these terms, the sPCA solution is obtained by:

$$\hat{\mathbf{U}}_{sPCA}, \hat{\mathbf{V}}_{sPCA} = \min_{U,V} \frac{1}{2}\|\mathbf{X} - \mathbf{U}\mathbf{V}\|_2^2 \ + \lambda\|\mathbf{V}\|_1 \text{ subject to} \|\mathbf{U}_i\|_2 = 1 \ \forall i \tag{2}$$

Due to the addition of the $\ell_1$ norm ($\|\mathbf{V}\|_1$; i.e. the sum of the absolute of the components of V) to the optimization objective, the optimum can only be achieved when many of the weights of $\mathbf{V}_i$ are zero, hence promoting sparsity. We employ the efficient solver presented in (Mairal et al., 2009) as implemented in the scikit-learn sparse PCA module (Pedregosa, 2011).

The model hyperparameter $\lambda$ in Eq. (2), whose value is chosen by the user, controls how much the $\ell_1$ norm participates in the definition of the optimum, i.e. it controls how sparse the solutions are. The other hyperparameter to be set is the number of principal components $c$. In contrast to the standard PCA, this number has to be set beforehand and influences the amount of information extracted for each component. In this work, we set $\lambda$ to the default value of 1, which is a good choice for normalized input data. We set $c$ to a value trimming low variance components. Alternatively, one can use data withheld from the main analysis to optimize over the explained variance, but we opted to include all the data in the analysis. An estimate of the explained variance is obtained by following the approach presented in section 3.4 of Zou et al. (2006).

### 3.2 Estimation of statistical uncertainty using bootstrapping

Bootstrapping provides a robust approximation of the statistics by resampling the available data and computing measures of interest several times from the subsamples. We use this strategy to combine different sPCA solutions, which are based on subsamples of the dataset, and obtain an empirical distribution over the sPCA weights ($\mathbf{V}$) and corresponding LVs ($\mathbf{U}$). From





the empirical distributions, we extract median, mean, median absolute deviation, and standard deviation to assess the robustness

of the solutions with respect to noise.

When running the sPCA optimization, i.e. solving Eq. (2), on a given random subset of data points, the solutions differ typically in the order of the components, the magnitude of weights, and the explained variance. Also, the sign of the weights can be arbitrarily flipped for each solution: with the sign of the LVs flipped accordingly, however, the explained variance does not change. To compute statistics from the bootstrap runs, one must first align the different solutions in a meaningful way

without losing the intrinsic variability brought by the bootstraps. Note that, for every bootstrap, the value distribution of each OV varies, and so does the influence of noise, so that one cannot expect that components across different sPCA bootstraps to be aligned directly, for instance according to their order or the amount of explained variance, as this value can vary.

For this work, we propose an approach based on alignment and thresholding. We first ran the sPCA on the full dataset, which we named "master" run. This first sPCA decomposition provides an ordered set (ordered by explained variance) of

LVs and corresponding weights, which was used as reference for the alignment of the sPCA bootstraps. Next, we computed as many sPCA models as there are bootstraps, where each one of them relies on a random subset of the original data points. Each solution is then aligned to the master run components by a two-fold strategy. First, we tested whether it was necessary to invert the signs of the sPCA runs. To do so, we computed the correlation between the master sPCA run and each component from each bootstrap. All bootstraps that were clearly negatively correlated with any of the master run LVs were flipped in sign

(remember that sign is arbitrary). Subsequently, for each LV from the master run, we computed a similarity score defined by a radial basis function (RBF):

$$s(\mathbf{v}_i, \mathbf{v}_j) = \exp \frac{-d(\mathbf{v}_i, \mathbf{v}_j)}{2 * \bar{\mathbf{u}}^2}, \tag{3}$$

where $\bar{\mathbf{u}}$ is the median Euclidean distance, for all the pairs of weights considered. Then, the component being the most similar and showing a similarity score above 0.5 was taken from each bootstrap and assigned to the ID of the master LV (note that

$s(\mathbf{v}_i, \mathbf{v}_j) \in [0, 1]$, where 1 implies $\mathbf{v}_i = \mathbf{v}_j$). Bootstrapped statistics are obtained from these alignments. We report mean and standard deviations of weights and LVs, but also the mean and standard deviation of the explained variance scores coming from each bootstrap. Note that some LVs might be derived from a number of aligned bootstraps lower than the total number of bootstraps. In that case statistics might be less robust, although we did not observe significant changes in statistical moments when using more bootstraps. Therefore, we also report the number of aligned bootstraps for each component.

This heuristic approach has three main benefits: firstly, the master run is only used as an alignment basis, and it does not influence bootstrap statistics. We made the assumption that noise does not strongly influence the master solution, at least in the first, most informative, LVs. If components were driven by noise, the bootstrapped solutions would be dissimilar enough to prevent meaningful alignment. Secondly, the alignment to a master component avoids testing all the possible pairings, simplifying from an order of $O(b^2 c^2)$ to a $O(bc)$, where $b$ is the number of bootstraps and $c$ the number of components, while

also greatly reducing the choice of matches. Finally, note that the set of LVs for every bootstrap is almost orthogonal. The orthogonality simplifies the selection of components based on the RBF similarity measure, as for every master component, at most one component from every bootstrap will be selected with $s(\mathbf{v}_i, \mathbf{v}_j) > 0.5$. However, as a result, the final set of





averaged LVs might show higher correlations than the LVs of the individual bootstrap runs. The highest correlations between the averaged LVs was between LV1 and LV14 ($R^2 = 0.73$, Figure B1a in Appendix B). A permutation test finds a p-value

of 0.0002 for that specific correlation (Figure B1b in Appendix B). We used a nonparametric permutation test since the LVs are not normally distributed, as underlined by a normality test. However, note that none of the weight vectors corresponding to each LV shows significant correlations, as expected. Correlation between LVs can be expected, as different and potentially independent causes can show activations at similar positions in space and time.

### 3.3    Data preprocessing and model setup

Observations were recorded at various time resolutions, ranging from seconds in the case of wind speed, to several hours for water samples taken from the underway sampling line (approximately every 6 hours) and the 8- and 24-hour aerosol filter samples (for some chemical compounds and ice nucleating particles). We chose to resample all observations to a 3-hour resolution to obtain a uniform spatio-temporal grid along the cruise track. The raw data were first resampled to 3-hour non-overlapping averages, regardless of the number of data points present in each 3-hour window. This averaging smooths the

data temporally, removing some high frequency noise and fills missing observations on shorter timespans. Measurements and samples, which are available at a lower frequency, are assigned to the nearest 3-hour window, leaving those windows which did not include an observation, empty. The combined observations provide a data matrix $\mathbf{X} \in \mathbb{R}^{N \times d}$, with $d = 111$ columns for the original variables and $N = 710$ rows per time interval (at 3-hour resampling).

Since we deal with very heterogeneous data, running the sPCA on the covariance between raw OVs would provide a decom-

position heavily influenced by OVs showing wider dynamic ranges. In order to reduce the effect and harmonize the contribution of each OV, we renormalised the data to a common range (by computing standard scores) and a common probability distribution, to uniformize influence of heavy tails and outliers. Note that using standardised data corresponds to performing the sPCA on the correlation matrix rather than on the covariance. The distributions of the OV may be approximated with different distribution types: for example, aerosol number concentrations are often described with a log-normal distribution (Schmale

et al., 2017); also some observations are better fitted by gamma distributions (Li et al., 2015); the wind speed is known to approximate a Weibull distribution (Hennessey, 1977); and rainfall may be best described with an exponential distribution (Woolhiser and Roldán, 1982). However, some variables show more complex multimodal distributions, e.g. water depth. Here, we classify the OVs based on their distribution during ACE as either approximately normally or approximately log-normally distributed, and apply a log-transform in the latter case to obtain an approximately normal distribution (see fifth column in the

tables Table A3 to Table A11 in Appendix A). Subsequently, we separately normalised each OV to standard scores by mean centring and reduction to unit standard deviations. For some OVs, e.g. precipitation types (rain, snow, horizontal hydrometeor flux), zero represents a valid observation, while at the same time they are better approximated by a log-normal than by a linear distribution. Therefore, the actual zero values cannot be represented exactly by our preprocessing method. To avoid the loss of zero observations, we decided to replace these with a lower limit value (see Table C1 in Appendix C). Where available, we

used the detection limit of the measurement device.





The model is optimized by extracting $c = 14$ principal components and using $\lambda = 1$. The $c$ and $\lambda$ values have been tuned by maximizing the total explained variance, while maintaining the level of sparsity above $\frac{1}{3}$ of the number of OVs. This rule of thumb led to interpretable results and has been kept. We ran five imputation iterations (see section 3.4), and tested different input-data resampling time intervals: 20, 180, and 720 minutes. Those intervals represent very short, medium and long time

frames as compared to the sampling intervals of the OVs. We used 30 bootstraps to estimate the variations, and observe that more do not lead to different estimates. Each bootstrap randomly samples 75% of the available data with replacement. We did not tune this value but we expect its influence to be negligible if the number of bootstraps is set accordingly.

### 3.4    Missing data and imputation

Most of the OVs have gaps and missing data. To overcome the caveats of data gaps, we iteratively estimated missing data

by inverting the sPCA model at the current iteration $t$, $X^t \approx U^{t-1}V^{t-1}$. Using a derivation of the expectation-maximization method (Grung and Manne, 1998) we computed a first solution of the sPCA by replacing missing values with the sample mean, corresponding to 0 after data standardization as described above. By inverting the decomposition, we obtained a first guess reconstruction of the missing data, which is used to fill the gaps in the time series for the next iteration. This process is commonly referred to as data imputation. Note that the real observations are not altered in this process. The reconstruction

error is guaranteed to decrease (Grung and Manne, 1998), and we obtain the final decomposition after five imputation iterations. More iterations reduce the reconstruction error only marginally. The sparse decomposition affects the ability to reconstruct the variations over all the OVs in the sense that OVs associated with a 0 weight over all LVs cannot be accurately reconstructed.

### 3.5    Model Limitations

The main limitation of the proposed framework is that there is not an explicit underlying temporal model. Most phenomena

show some level of smoothness in time, e.g. two observations acquired at short time intervals are more related than two observations acquired at times that are further apart. In our setting, after temporal resampling, we assume data window is independent from all the others and our model does not provide a statistical representation of the temporal variations jointly to temporal evolution.

A second major limitation is the strict linearity assumption, as we are working on correlations. While this assumption might

seem restrictive, we notice that large patterns in OV relationships can still be approximated by a simple linear function. Furthermore, linear models are much more robust to unquantified noise and ultimately easier to interpret thanks to the sparsity in the linear weights. We also notice that it is difficult to develop a data-driven model evaluation, where numbers could objectively quantify the decomposition accuracy. However, as in most unsupervised learning tasks, there is no clear evaluation protocol. In this work, we rely on the evaluation of the correlations found by the model and how they compare to the current state of

knowledge.

The sparsity constraint on the weight vectors leads to an underrepresentation of certain OVs with sparse time series, for which the potential gain in explained variance is low. The natural process represented by these OVs, e.g. the surface ocean mixed layer depth (MLD), can thus be underrepresented in our interpretation of the LVs. Therefore, the absence of a certain





OV from our interpretation does not necessarily imply its absence in nature, but might be purely related to data scarcity. This

limitation can be somewhat mitigated by running the model at lower temporal resolution (see Appendix section D), but this
would be at the expense of the information content of the denser time series.

## 4 Sparse PCA output description

### 4.1 Introduction to the latent variable naming

Figure 3 shows the time series of the 14 LVs, where the blue dots indicate the average of the principal components of the
bootstrap runs and the shading indicates the 95% confidence interval ($\pm 2$ standard deviations). Here, we provide an overview
of the LV time series and their linkages before discussing them separately in section 5.

The 14 LVs can be related to physical, biological and/or chemical processes, or changes in the environment that influence
the variance of OVs within each LV. We name each LV according to the process or environmental condition, which they
reflect (Table 2). The detailed interpretations are provided in section 5. In brief and in the order as discussed in section 5
climatic zones and large-scale horizontal gradients are represented by LV1 and LV14. Meridional advection of cold or warm
air masses that influence air-sea interaction processes feature in LV3, precipitation events in LV4, and the passing of large-
scale weather systems in LV13. Effects of islands and sea bed topography are discussed in LV5, and the influence of sea ice
in LV9. Atmospheric microphysical processes are addressed in LV2 for the cloud condensation nuclei (CCN) population as
well as in LV12 for wind-driven sea spray production. Three LVs (LV6, LV8 and LV11) describe the relationships between
the composition and productivity of the microbial community, and the availability or scarcity of nutrients and light. Temporal
features are represented by LV10 for diurnal patterns and LV7 for seasonal patterns. Our interpretations of these LVs are based
on the combination of contributing OVs as well as the LV-time series. The sPCA results allow us to separate these different
processes and environmental conditions, and to understand how those vary along the ACE cruise track.

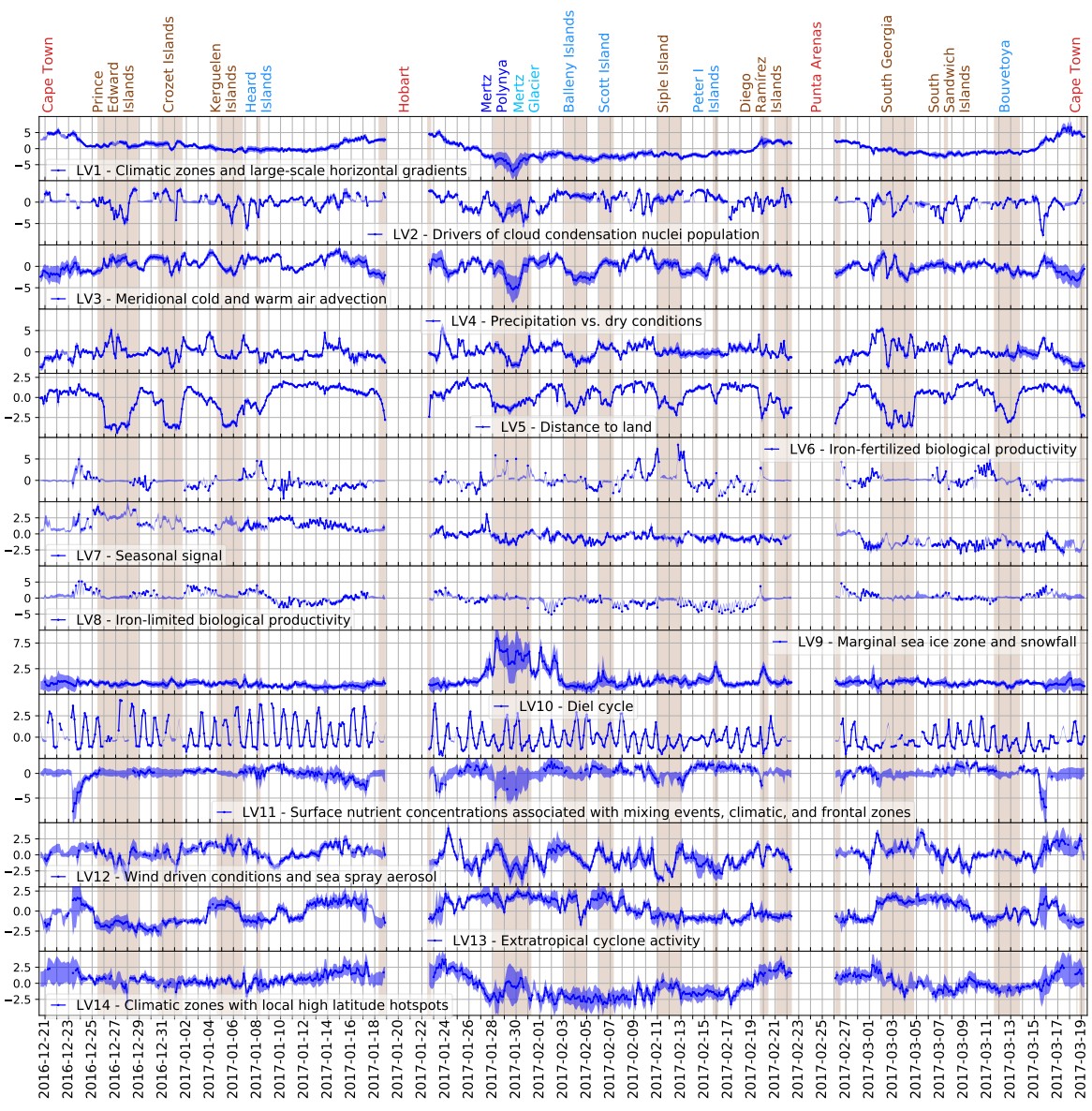

**Figure 3.** Time series of the LV activations. Each time series (blue dots) is calculated as the average of the principle components of the bootstrap runs and the standard deviation (STD) is used to estimate the 95% confidence interval as ±2STD. Mean activation of the LV (blue dots) is only shown if more than 50% of the OVs with the 50% largest weights were observed. Brown shading indicates periods when the ship was within 100 km to the nearest shore. Places visited (ports, islands and the Mertz Glacier and Polynya) are indicated at the top.



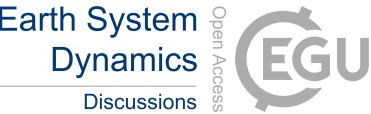

**Table 2.** List of the 14 LVs in the sPCA solution. The columns denote the LV-ID (ranked by total variance explained by the master solution), the LV-title, the number of matching LVs found in the 30 bootstrap runs, the variance explained by the LV, as well as the mean and standard deviation of the variance explained by the bootstrap runs.

| ID | LV-title | Bootstrap matches | Explained var. master run | Explained var. bootstrap runs |
|----|----------|-------------------|---------------------------|-------------------------------|
| LV1 | Climatic zones and large-scale horizontal gradients | 30 | 9.2% | 9.5(±0.6)% |
| LV2 | Drivers of cloud condensation nuclei population | 30 | 4.6% | 4.4(±0.5)% |
| LV3 | Meridional cold and warm air advection | 30 | 4.5% | 4.8(±0.4)% |
| LV4 | Precipitation vs. dry conditions | 30 | 4.4% | 3.9(±0.6)% |
| LV5 | Distance to land | 30 | 4.0% | 4.0(±0.3)% |
| LV6 | Iron-fertilized biological productivity | 30 | 3.7% | 3.8(±0.4)% |
| LV7 | Seasonal signal | 30 | 3.5% | 3.5(±0.2)% |
| LV8 | Iron-limited biological productivity | 30 | 3.4% | 3.5(±0.3)% |
| LV9 | Marginal sea ice zone and snowfall | 29 | 3.2% | 3.4(±0.6)% |
| LV10 | Diel cycle | 30 | 3.1% | 3.2(±0.2)% |
| LV11 | Surface nutrient concentrations associated with mixing events, climatic, and frontal zones | 25 | 3.1% | 2.9(±0.4)% |
| LV12 | Wind driven conditions and sea spray aerosol | 30 | 3.1% | 3.2(±0.4)% |
| LV13 | Extratropical cyclone activity | 25 | 2.7% | 2.6(±0.2)% |
| LV14 | Climatic zones with local high latitude hotspots | 25 | 2.6% | 3.2(±1.4)% |





## 4.2 Latent variable time scales

We characterize the spatio-temporal variabilities captured by the LVs by a frequency analysis (see Figure 4, which shows the periodogram of the LV series). Note, because the ship was mostly moving, spatial variability will appear as temporal variability and this needs to be taken into account when interpreting results. We can classify the LVs based on their peak period or spectral ranges of high activity: the 1-day period of LV10 is most obvious and reveals the relation of this LV to the diurnal cycle (solar radiation). The spectra of LV1 (climatic zones), LV5 (distance to land), and LV14 (Climatic zones with local high-latitude hotspots) have a distinct peak at a period of 30 days, which is about one third of the total duration of the cruise and reflects the repeated travel from north to south and from continental influences to remote open ocean conditions during the three legs. Several LVs (LV3, LV12, LV4, LV11) show enhanced activity at approximately 3- to 10-day periods, which is related to synoptic time scales. For the remaining LVs most variations occur at periods longer than 10 days.

## 4.3 Contribution of the original variables to the latent variables

Single OV contributions to the LVs are controlled by the sPCA weights. We represent them by plotting the explicit distribution obtained through bootstrapping as boxplots. Figures S1 to S4 in the supplementary information shows this representation for every LV. Non-zero median values represent variables, which are active for a specific LV for more than half of the bootstraps. We measured the significance of the weights by the ratio of their magnitude to their variability. In order to obtain robust measures, we used the median of the weights $\tilde{w}$ and the median absolute deviation (MAD) over the bootstrap runs, with $\sigma = 1\,\mathrm{STD} = 1.4826\,\mathrm{MAD}$ for normally distributed data. Significance of the weight assignments is measured in terms of the ratio $\frac{\tilde{w}}{\sigma w}$. Note that the absolute sign of the weight is arbitrary: for the sPCA, opposite sign assignments lead to the same amount of variance. Within each LV a positive sign denotes positive correlation of the OV with the LV time series and a negative sign denotes anticorrelation.

We quantified the dependency of OVs to each LV by computing their covariance. We scaled this value by the corresponding OV-LV entry in the weight matrix, to make the covariance comparable across all OVs.

$$\mathrm{cov}_{\mathrm{scaled}}(X_i, U_j) = \frac{(X_i^\top U_j) \cdot V_{ij}}{N_i} \tag{4}$$

Figure 5 shows the obtained contribution of each LV to explaining the variance of the OVs with the reconstruction. Most OVs co-vary with more than one LV and all LVs are related to OVs from multiple OV-categories as defined in section 2.2 with the colour scheme shown in Fig. 5. As described in section 3.5 we find that the capability of the sPCA reconstruction to explain the variability of the OVs (given by the sum of the scaled covariances) is very low for time series with a large fraction of missing observations.



**Figure 4.** Periodograms of the LV activation time series. The black marker indicates the peak period in days (only shown if smaller than $\frac{1}{2}$ of the sample period).

**Figure 5.** Covariance between the LVs and OVs scaled by the sPCA weights (see Equation 4) as stacked bar plots, with the colour denoting the LV (see legend in the lower panel). OV categories (see section 2.2) are indicated by the background shading (see legend in the upper panel).



## 5 Results

The following subsections provide in-depth analyses of the 14 latent variables in the context of the original variables which contribute to them. A contribution is considered, if the median value of the contributing weight is larger than its single standard deviation from the bootstrap runs. The LVs are grouped based on the six themes (i) climatological background provided by the large-scale ocean and atmosphere circulation (LV1 and LV14); (ii) atmospheric and oceanic advection, cyclones and precipitation (LV3, LV4 and LV13); (iii) the effect of islands, continental land, and sea ice (LV5 and LV9); (iv) aerosol concentrations (LV2 and LV12); (v) ocean microbial dynamics (LV11, LV6, and LV8); and (vi) solar forcing (LV7 and LV10). A further synthesis of the result discussions is provided in section 6.

### 5.1 Climatological background provided by the large-scale ocean and atmospheric circulation

#### 5.1.1 LV1 - Climatic zones and large-scale horizontal gradients

Latent Variable 1 (LV1) explains most of the variance (9.2%) in all the 111 analysed original variables (see Table 2 and Figure 6). It mirrors the climatic background conditions during ACE and ranges from strongly positive LV1 values in the lower latitudes to strongly negative values in the high latitudes. These climatic zones are reflected in the strong horizontal gradients in the air ($T_\mathrm{air}$) and surface ocean temperature ($T_\mathrm{sw}$; Figure 6a), and in the atmospheric water vapour mixing ratio ($w$). These three variables contribute most to LV1 and are correlated positively with LV1 (Figure 6c). The most positive LV1 values occurred during the warmest conditions at the beginning and end of the cruise near Cape Town (Figure 6b). The most negative LV1 values were found during the coldest conditions on 29 January 2017 when the ship was near the Mertz Glacier and under the influence of Antarctic air masses during a cold-air outbreak event (see LV3 and LV9 discussions). In contrast with the most positive LV1 values, which are governed by both high atmospheric and ocean temperatures, the most negative LV1 values are largely caused by low atmospheric temperatures alone. This difference can be explained by the absolute lower bound of the ocean temperature being set by the freezing point.

A second major contribution to LV1 is associated with oceanic regions as reflected by the large contributions of sea-surface height (SSH), surface ocean potential density anomalies ($\sigma_{0,sw}$), and surface ocean geostrophic velocity ($U_\mathrm{g}$). This influence of the oceanic conditions can also be depicted by a change of the LV activation when crossing major oceanic fronts (Figure 6b). Different frontal zones are associated with different ocean biogeochemistry, which is typically associated with nutrient concentrations and their ability to limit primary producers as observed during this expedition (Janssen et al., 2020). Yet, the observed macronutrients (N, P, and Si) are not activated in LV1, suggesting a more complex situation across the Southern Ocean. Several zonal deviations associated with ocean currents were observed in LV1 as it changed from positive to negative values while the ship was moving from Patagonia to South Georgia on a similar latitude band (Figure 6b). These decreasing LV1 values across a similar latitude band can be explained by the northward deflection of polar waters with the Antarctic Circumpolar Current (ACC) driven by topography as outlined by the Subantarctic, Polar, and Southern ACC Fronts in this region, leading to a strong decline of $T_\mathrm{sw}$, SSH, and surface ocean salinity ($S_\mathrm{sw}$), and an increase in $\sigma_{0,sw}$ along this cruise transect. This climatological position of ocean currents leads to relatively cooler ocean surfaces compared to near-surface air



**Figure 6.** (a) time series of the activation of LV1 (left axis) and of the air and water temperatures (right axis). Crossings of the oceanic fronts, which were derived from observations during ACE, are indicated by vertical lines (see supplementary information section S1 for details): from north to south as Subtropical Front (STF; dotted), Subantarctic Front (SAF; dashed), and Polar Front (PF; solid), and Southern ACC Front (SACCF; dash-dotted); (b) map of the ship track coloured by the activation of LV1. Climatological frontal positions (Orsi et al., 1995) are shown as black lines, denoted as in (a). Crosses indicate the ACE-data derived frontal positions; (c) box and whisker plots of the activated weights from the bootstrap runs (only weights with $\overline{w} > 1\sigma$ are shown). The colors indicate the OV categories (see Table 1).





temperature and is likely responsible for the positive weight of the air advection mask ($\mathrm{mask_{cw}}$) in LV1 (see also discussion of
      LV3). The opposite, i.e. a signature from the warm West and East Australian boundary currents, is visible south of Tasmania
      and east of South Africa with the overall most positive LV1 activations. Moreover, zonal gradients can also be observed in the
      southernmost parts (about $65\,°\mathrm{S}$ to about $74\,°\mathrm{S}$) of the cruise in the Pacific sector, where the lowest LV1 values of the entire
      cruise occur in the south-western Pacific and slightly higher LV1 values occur in the central and eastern parts of the transect

(Figure 6b).

      The interpretation that LV1 describes climatological background conditions rather than variations on the timescales of e.g.
      weather (LV12 and LV4), season (LV9 and LV7), or the diel cycle (LV10), is supported by its low frequency (Figure 4) and
      numerous OVs with a known strong climatological influence. LV1 has the highest contribution to the variance in the stable
      water isotopic composition of water vapour ($\delta^2 H_\mathrm{vap}$, $\delta^{18} O_\mathrm{vap}$) and seawater ($\delta^{18} O_\mathrm{sw}$; Figure 5). Such a large influence of the

climatic zones on the meridional distribution of the atmospheric and oceanic stable isotopic composition during ACE has been
      noticed and described in detail by Thurnherr et al. (2020b) and Haumann et al. (2020b), respectively. The overall southward
      decline in these isotope ratios and $S_\mathrm{sw}$ are caused by a loss of moisture from the atmosphere to the ocean. LV1 also has the
      second highest contribution to the atmospheric pressure ($P_\mathrm{air}$ in Figure 5) reflecting the transition from climatologically high-
      pressure subtropical regions to the circumpolar trough surrounding Antarctica, characterized by high cyclone activity (Papritz

et al., 2014).

      LV1 additionally includes information about atmospheric chemistry and microphysics. Gas phase concentrations of sulfuric
      and methanesulfonic acid (MSA) are higher at higher latitudes. These higher concentrations are consistent with emissions of
      DMS which result from higher summertime microbial activity (phytoplankton) near the Antarctic coast. DMS is converted into
      the aforementioned gaseous acids in the atmosphere, which promote new particle formation and aerosol growth (Lana et al.,

2011; Chen et al., 2018; Schmale et al., 2019) and are consequently found as sulfate and MSA in the aerosol particles. In
      general during ACE, ice nucleating particles (INP) at $-20°\mathrm{C}$ ($N_{\mathrm{INP},-20}$) decreased with increasing latitude. Given that at this
      temperature, i.e. - 20°C, the dominant INP is mineral dust, the observed relationship is plausible. At lower latitudes $N_{\mathrm{INP},-20}$
      increases, where dust emissions from Africa, Australia and South America potentially contribute (Welti et al., 2020).

      **In summary**, the largest variance in the 111 analysed variables during ACE (9.2%) results from the climatic zones along

the cruise track, ranging from warm and humid subtropical conditions in the most northern regions during the cruise (positive
      LV1 values) to cold and dry polar conditions at southern high latitudes (negative LV1 values). These conditions are reflected in
      both the surface atmosphere and ocean variables.

### 5.1.2   LV14 - Climatic zones with local high latitude hotspots

      The time series of LV14 is largely similar to LV1 with a correlation coefficient of 0.73 and a p-value of 0.02 (see Figure B1 in

Appendix B). However, the amplitude of LV14 is smaller and it explains only $3.2(\pm 1.4)\%$ of the total variance. The general
      difference between LV14 and LV1 arises from the opposite sign in LV14 in some high latitude regions, where suddenly some
      variables go against the large-scale gradient. Examples are near the Mertz Glacier (January 28, 12:00 UTC until January 30,
      00:00 UTC), near Siple Island (February 11, 00:00 UTC until February 13, 00:00 UTC), on the northern side of South Georgia





(March 1, 00:00 UTC until March 2, 00:00 UTC), and before approaching Bouvetøya (March 8, 00:00 UTC until March 11,
00:00 UTC), where especially the biological activity tracers chlorophyll $a$ and net community production (Chl $a$, NCP) as
well as the Aitken mode particles show variations against the large-scale meridional trend. Further the change in the amplitude
of LV14 north of the SAF is weaker compared to the variability south of the SAF when comparing to LV1. Air and water
temperature as well as the water vapour mixing ratio ($w$) and sea-surface height (SSH), which are all dominant in LV1 are also
active in LV14. However, they have much lower importance compared to the weights of other OVs in LV14 such as rain rate
(RR) and snowfall rate (SR), which have opposing trends and indicate a transition from rainfall in lower latitudes to preferential
snowfall near Antarctica.

   The OV with the largest weight in LV14 is the number concentration of particles in the Aitken mode ($N_{\mathrm{Aitken}}$). Particles in
the Aitken mode are typically smaller than 80 nm in diameter (the median and interquartile range for the Aitken mode diameter
during ACE were 38 nm and 24 - 47 nm, Schmale et al. 2019) and are mostly considered to be formed from gas to particle
conversion. In the context of the spatial distribution of LV14's activation, the positive weight of $N_{\mathrm{Aitken}}$ indicates that they
increase with proximity to the non-glaciated continents and decrease close to Antarctica, with the exception of two new particle
formation (NPF) events near the Mertz Glacier (January 29, January 30/31) related to the special environmental conditions
encountered there (low temperature, low concentration of larger aerosol and relatively high solar irradiance). LV14 also explains
a considerable fraction of the variance of the number of particles acting as cloud condensation nuclei at a supersaturation of
1% ($N_{\mathrm{CCN,1.00}}$) further north. At such high water vapour saturation the smaller Aitken mode particles become competitive
to the larger accumulation mode particles, which generally dominate the cloud condensation nuclei (CCN) population (see
section 5.4.1). Also the hygroscopicity parameter of particles acting as CCN at 1% supersaturation ($\kappa_{\mathrm{CCN,1.00}}$) contributes to
LV14, underlining the role of such small particles as cloud seeds where LV14 has a strong positive activation.

   Several oceanic OVs included in LV14 show clear latitudinal variations. The surface geostrophic current velocity ($U_{\mathrm{g}}$)
was generally high in regions of positive LV14 activation, especially during times spent on the northern flank of the ACC.
During these times, we also notice an increased significant wave height ($H_s$), and expected general northward increases in
the sea-surface height (SSH) and sea surface temperature ($T_{\mathrm{sw}}$). Surface ocean silicate concentrations (Silicate) increase with
latitude (Sarmiento et al., 2004; Freeman et al., 2018), as shown by the negative correlation with LV14. Its gradient largely
coincides with the latitudinal gradients in the physical quantities and the frontal positions due to its consumption by diatoms
that favourably grow in higher latitude waters (Freeman et al., 2018). Diatoms are a large sized species of phytoplankton,
that specifically require this macronutrient. The broad latitudinal trends in the distribution of Silicate are driven by physical
processes, e.g., the lateral supply from the southerly upwelling of the Circumpolar Deep Water source. The phytoplankton
communities south of the Polar Front are typically larger in size relative to lower latitudes. This is supported by the positive
loading of the cryptophyte (Crypto) variable which is driven by an isolated peak in the biomass of cryptophytes, a larger
sized phytoplankton. In addition, $\sigma_{PSII}$ and $\sigma^{`}_{PSII}$, which are the functional absorption (light harvesting) cross-sections of
photosystem II of phytoplankton in the day and night respectively, have positive weights, because the absorption cross section
co-varies with phytoplankton taxa and size (Moore et al., 2005; Suggett et al., 2009). They generally decrease with increasing
cell size, hence underlining that large phytoplankton resides further south. In contrast, net community production (NCP) and





fluorescence of in-water chlorophyll $a$ (Chl $a_{\mathrm{fluo}}$) show both a positive weight, due to the lower concentrations at higher

latitudes.

**In summary** we interpret LV14 as a second group of OVs whose variability is largely affected by the climatic zones and large-scale latitudinal gradients (as in LV1), but with distinct local signals near the Mertz Glacier as well as near some of the islands, and a more moderate latitudinal change north of the SAF.

## 5.2 Atmospheric and oceanic advection, cyclones and precipitation

### 5.2.1 LV3 - Meridional cold and warm air advection

LV3 represents the effects of atmospheric large- to meso-scale and oceanic large-scale advection on the atmospheric composition and stability and explains $4.8(\pm 0.4)\%$ of the total variance. The highest contributions to LV3 stem from relative humidity (RH) and deuterium excess in water vapour ($dexc_{\mathrm{vap}}$, see Figure 8), reflecting the effect of the differing strengths of ocean evaporation in different RH environments on the isotopic composition of water vapour in the marine boundary layer. Strong

ocean evaporation at low RH leads to high $dexc_{\mathrm{vap}}$ in the evaporated water vapour due to strong non-equilibrium effects during isotopic fractionation. This anticorrelation of RH and $dexc_{\mathrm{vap}}$ has been observed and discussed in previous studies (Uemura et al., 2008; Gat, 2008; Pfahl and Sodemann, 2014; Aemisegger and Sjolte, 2018) .

The correlation of the cold and warm temperature advection mask ($mask_{\mathrm{CW}}$) with RH shows the importance of the large-scale, meridional horizontal advection in shaping the RH environment (Thurnherr et al., 2020a). Due to the increased saturation-

specific humidity at high temperature, RH decreases if cold air masses are advected towards the equator over a relatively warm ocean surface, and vice versa for warm air advection over a relatively cold ocean surface. Furthermore, low abundances of heavy water molecules ($\delta^2 H_{\mathrm{vap}}$ and $\delta^{18} O_{\mathrm{vap}}$) and low water vapour mixing ratios ($w$), which are normally observed for atmospheric water vapour at high latitudes (see also LV1), are seen with low RH indicating meridional large-scale advection from the south. The different flow patterns of the air masses for cold and warm temperature advection can be seen with 2-day

backward trajectories in Figure 8b, which often show an equatorward flow for cold and a poleward flow for warm temperature advection, respectively.

While the $mask_{\mathrm{CW}}$ has a large weight in LV3, the sea water temperature ($T_{sw}$) contributes only weakly and the air temperature ($T_{air}$) does not have any median contribution to this LV. This absence of significant contributions by $T_{sw}$ and $T_{air}$ illustrates the importance of the difference between the ocean and atmospheric temperatures rather than their absolute values

in shaping the RH environment. Therefore, while LV1 represents the large-scale horizontal gradients in the atmosphere and ocean, LV3 is mostly associated with vertical gradients between the atmosphere and the ocean due to large-scale meridional advection. The negative weights of $T_{sw}$ show that regional $T_{sw}$ anomalies due to oceanic currents can contribute to the air-sea temperature differences. For example, in the region of the warm Agulhas current around $30°E$, where relatively warm Indian Ocean waters are advected under a relatively cool atmosphere, a negative activation of LV3 is seen (see Figure 8b).

Next to the dominant OVs, RH, $dexc_{\mathrm{vap}}$ and $mask_{\mathrm{CW}}$ in LV3, weaker weights are seen for several OVs, which further characterize the different RH environments. We observe a tendency of higher cloud level (CL) and higher atmospheric ozone





**Figure 7.** (a) time series of the activation of LV14 "Climatic zones with local high latitude hotspots" (left axis), of the Silicate concentration (top half of the right axis), and the Aitken Mode particle number concentration (bottom half of the right axis); (b) map of the ship track (circles) coloured by the activation of LV14; (c) box and whisker plots of the activated weights. See caption of Figure 6 for details on the oceanic fronts and frontal crossings.

($O_3$) concentrations during dry conditions (low RH). This might indicate enhanced vertical mixing in the marine boundary layer during cold air advection, which might lead to the entrainment of free tropospheric air masses with higher $O_3$ concentration into





**Figure 8.** (a) time series of the activation of LV3 "Meridional cold and warm air advection" (left axis) and difference of the air- and seawater temperature (right axis); (b) map of the ship track (circles) coloured by the activation of LV3. Median track of the air mass backward trajectories "released" at the measurement location and calculated up to 48 hours prior to the release are shown as thin lines coloured by the activation of LV3; (c) box and whisker plots of the activated weights.

the marine boundary layer (see Figure 8c). An occurrence of higher surface wind speeds ($u_{10N}$) during the advection of cold air

(i.e. negative activation of LV3) can be seen, especially for the Pacific sector where high wind speeds are expected during cold





air outbreaks near the Antarctic continent (see e.g. Kolstad, 2017). High surface wind speed in extratropical cyclones is often associated with the cold front, but can occur in both the warm and cold sector of the cyclone and shows a complex mesoscale structure (see e.g. Browning, 2004). It is therefore not straightforward to associate high wind speed with areas of cold and warm air advection during the passage of extratropical cyclones, which is also represented by the weaker correspondence of

the surface wind speed with LV3 activation for the Indian Ocean and Atlantic sector (not shown). On the synoptic time scale, i.e. 3 to 6 days, which is the dominant frequency of LV3 (Figure 4), rainfall occurs more often during the advection of cold air masses. Conversely, comparing specifically strong cold and warm advection, which occurred in regions with an air-sea temperature difference of at least $1.0°C$, it has been shown that warm advection is associated with stronger rainfall than cold advection (Thurnherr et al., 2020a). Rainfall in the Southern Ocean is mostly related to atmospheric fronts (Catto et al., 2012),

which mark the boundaries of cold and warm air advection. In this study, cold and warm air advection is defined to include these frontal zones between the regions of cold and warm air advection. If frontal rainfall more strongly impacts regions of cold advection, it could be the reason for the difference in rainfall occurrence during cold and warm air advection to former studies. The contributions of LV3 to the total covariance of rainfall is, however, small compared to LV4 (Figure 5).

A further contribution to LV3 is given by the accumulation mode aerosol number concentration ($N_\mathrm{accumulation}$) with high

concentrations during the advection of cold air. Accumulation mode particles are washed out during rainfall and are thus expected to show lower concentrations with increased rainfall, which is in apparent contradiction with the higher rainfall during cold, compared to warm air advection. The low concentration of $N_\mathrm{accumulation}$ during warm air advection might be explained by precipitation before the air arrives at the measurement location. Backward trajectories show more precipitation occurring for the five days prior to arrival for warm air advection than for cold air advection (not shown). Therefore, the low

concentration of $N_\mathrm{accumulation}$ during warm air advection could be explained by the occurrence of rainfall before arrival at the measurement location. The same logic can be applied to cloud condensation nuclei number concentrations at a supersaturation of $0.3\%$ ($N_\mathrm{CCN,0.30}$), which also exhibit a negative loading. Accumulation mode particles constitute the largest fraction of this type of CCN.

**In summary**, the large-scale advection of air masses or ocean currents leads to near-surface vertical temperature and humid-

ity gradients in the atmosphere. These situations and the induced processes are reflected by LV3 and best represented by the synoptic time scale variability of atmospheric relative humidity and the isotopic composition of the atmospheric water vapour.

### 5.2.2 LV4 - Precipitation vs. dry conditions

LV4 represents the direct effects of single precipitation events, which occur on the time scale of hours (see Figure 9), on humidity and visibility in the marine boundary layer and explains 4.4% of the total variance in the data. LV4 shows high

contribution from rainfall (RR) and relatively smaller contribution from snowfall (SR). Both RR and SR positively correlate with cloud cover (sky cover; SC), relative humidity (RH) and horizontal hydrometeor flux (HHF), and anticorrelated with horizontal visibility (visibility), lowest cloud level height (CL) and atmospheric pressure ($P_\mathrm{air}$). Thus LV4 features almost exclusively OVs from the category *Atmospheric hydrological cycle* (with the exception of $P_\mathrm{air}$).



**Figure 9.** (a) time series of the activation of LV4 "Precipitation vs. dry conditions" (left axis), of the sky cover in octants (top half of the right axis), and the three-hourly average precipitation rates as shaded areas (rain rate in blue and snowfall rate in pink) (bottom half of the right axis); (b) map of the ship track (circles) coloured by the activation of LV4; (c) box and whisker plots of the activated weights.

Total precipitation is mostly represented by rainfall, with a small contribution of snowfall. Approximately 70% of all pre-

cipitation over the Southern Ocean is related to atmospheric fronts (Catto et al., 2012) and, thus, LV4 is representative of the

time scale of mesoscale atmospheric dynamics, which is embedded in to the large-scale atmospheric flow as also seen in the





dominant period of several days for LV4 (Figure 4). Furthermore, no contribution from the cold and warm advection mask is present, which shows that LV4 reflects the dominance of the processes other than the large-scale advection, highlighting single precipitation events of several hours with a recurrence period of several days. These events are embedded in the large-scale

advection of cold and warm air (LV3), which is reflected in several similarities in the time series variations of LV4 and LV3 (see Figure 3) and a moderate correlation of $-0.54$ between the two LV time series (see Figure B1 in Appendix B).

**In summary**, LV4 illustrates the effect of mesoscale precipitation events on the horizontal visibility and cloud level in the marine boundary layer. These events are embedded in the large-scale advection of cold and warm air and lead to a clear increase in relative humidity and cloud coverage.

### 5.2.3    LV13 - Extratropical cyclone activity

LV13 explains $2.6(\pm0.2)\%$ of the variance and features predominantly low frequency variations. The two most important OVs contributing to LV13 are the atmospheric pressure ($P_{\mathrm{air}}$) and the cyclone mask ($\mathrm{mask_{cyc}}$), which indicates the occurrence of extratropical cyclones along the ACE cruise track. $P_{\mathrm{air}}$ and $\mathrm{mask_{cyc}}$ are anticorrelated as extratropical cyclones are identified by a minimum in $P_{\mathrm{air}}$ (Figure 10). LV13 explains the largest part of the variability in $P_{\mathrm{air}}$ during ACE (Figure 5) and illustrates

that the passage of extratropical cyclones is the main reason for pressure variations over the Southern Ocean.

Extratropical cyclones are generally associated with precipitation, high wind speed and enhanced cloud cover (Field and Wood, 2007). The absence of precipitation and snowfall rates (RR and SR) and wind speed ($u_{\mathrm{10N}}$) and the anticorrelation of sky cover (SC) with $\mathrm{mask_{cyc}}$ in LV13 do not agree with the general picture of extratropical cyclones, and might be related to the different dominant spatial and temporal scales of these variables compared to extratropical cyclones. While extratropical

cyclones cover a relatively large area and were often identified over a period of several days up to a week along the ACE track, the aforementioned variables show strong variations on the time scale from hours to a few days. Furthermore, RR, SR, $u_{\mathrm{10N}}$ and SC are more strongly associated with specific features within extratropical cyclones, such as the cyclone's sectors and fronts (Hobbs, 1978; Browning, 1990), than with the entire cyclone as also shown with LV3 and LV4 (Figure 8 and Figure 9). The organisation of clouds along frontal structures could also explain the anticorrelation of SC and $\mathrm{mask_{cyc}}$ as areas of frequent

cyclone occurrence do not coincide with the areas of high front frequencies, which are located equatorward of the cyclone maxima in the Southern Ocean (Papritz et al., 2014).

The increase in surface ocean seawater oxygen isotopic fraction ($\delta^{18}\mathrm{O}_{sw}$) in areas with low atmospheric pressure could indicate an increased upwelling of subsurface waters that are characterised by higher $\delta^{18}\mathrm{O}_{sw}$. The upwelling could occur either through increased surface mixing related to the passing storm or increased vertical advection associated with a surface

divergence of the seawater.

The anticorrelation of some of the microbial biomass production indicators, namely the passive chlorophyll $a$ fluorescence (Chl $a_{\mathrm{fluo}}$), Cryptophyte (Crypto), and Diatom (DiatA) type contribution to chlorophyll biomass with the cyclone activity may be due to light limitation introduced by the stronger wind-driven mixing in the water column. However, the enhanced mixing can also lead to upwelling or lateral advection of iron (Ellwood et al., 2015), which would benefit phytoplankton growth. Note

that the contribution of LV13 to the variability of these biomass precursors is small (see Figure 5).



**Figure 10.** (a) time series of the activation of LV13 "Extratropical cyclone activity" (left axis) and of the atmospheric pressure (right axis); (b) map of the ship track (circles) coloured by the activation of LV13; (c) box and whisker plots of the activated weights.

There is no apparent explanation for the inclusion of carbon monoxide (CO), the mass concentration of sulfate in non-refractory particulate matter ($SO_4^{2-}$), and the atmospheric isoprene concentration (Isoprene$_{air}$), and further analysis is beyond the scope of this work.





**In summary**, LV13 mainly represents the variations in atmospheric pressure due to the passage of extratropical cyclones,
which are the main drivers of pressure variations over the Southern Ocean.

### 5.3 The effect of islands, continental land, and sea ice

#### 5.3.1 LV5 - Distance to land

LV5 represents the ship's position relative to the closest landmass and associated terrestrial influences with positive values
occurring close to land and negative ones for the open ocean sections (see Figure 11a). It captures $4.0(\pm 0.3)\%$ of the variability
in the dataset.

As shallower waters are found closer to land, water depth ($d_{\mathrm{water}}$) and distance to land ($d_{\mathrm{land}}$) each exhibit a positive loading.
Both variables represent the highest contributions to LV5. The geostrophic velocity ($U_{\mathrm{g}}$) is generally low in the shallower
continental shelf waters where large-scale oceanic horizontal pressure gradients are absent. In conjunction with a positive
loading for $U_{\mathrm{g}}$, this effect is represented by high LV5 values close to land. The available light within the surface mixed layer
of the water column ($I_g$) also appeared to decrease at locations closer to landmasses. The reduction of $I_g$ is likely driven by
increased light attenuation from higher particulate matter concentrations in the surface ocean (Tilzer et al., 1994; Perissinotto
et al., 1992; Blain et al., 2001) or changes to the depth of the surface mixed layer (by definition, $I_g$ is the median light intensity
within the mixed layer, see section S1.9 'Microbial, biogeochemical and optical properties' in the supplementary information
and (Behrenfeld et al., 2005)). This is due to variations in the bathymetry, horizontal water flows or introduction of fresh water
sources, for example (Dong et al., 2008).

A connection between marine and atmospheric quantities is made through the cold and warm temperature advection mask
($\mathrm{mask_{CW}}$). Here, the negative loading could indicate advection of (terrestrial) warm air near land, rather than glacier-covered
islands, in the Indian Ocean and Atlantic Ocean sectors. Warm air advection is underlined by a negative loading for air tem-
perature ($T_{\mathrm{air}}$) and positive loading for relative humidity (RH), indicating drier and warmer air close to land. Further, negative
loading for the rain rate (RR) and the wind speed at 10 metres above mean sea level ($u_{10\mathrm{N}}$) can be seen, which might be related
to their orographic enhancement near land. During ACE, atmospheric frontal systems often passed while the ship was close
to subantarctic islands. Therefore, a negative loading for $u_{10\mathrm{N}}$, $\mathrm{mask_{CW}}$, and RR might be due to a coincidental overlap of
island visits and the passage of frontal systems. For the height of the lowermost cloud level (CL), a negative loading was found,
indicating that the cloud deck was lower over the open ocean where moist air (negative loading of RH) leads to a lower cloud
condensation level.

Negative weights for the ratio of coarse fluorescent to total aerosol particles ($r_{\mathrm{fluo,coarse3}\sigma}$) and ice nucleating particle con-
centrations at both $-8°\mathrm{C}$ ($N_{\mathrm{INP,LV},-8}$) and $-20°\mathrm{C}$ ($N_{\mathrm{INP,LV},-20}$) show an increase closer to land. Particles acting as INP at
different temperatures are associated with different particle sources ($-8°\mathrm{C}$: biogenic origin; $-20°\mathrm{C}$: mineral origin). Fluores-
cent particles can be emitted from the ocean during sea spray generation (Wilson et al., 2015), but can also originate from land,
in which case they would correspond to pollen, fungal spores, and plant or animal detritus (Després et al., 2012; Fröhlich-
Nowoisky et al., 2016). Such types of biological particles are known to act as INP efficiently at higher temperatures (Kanji



**Figure 11.** (a) time series of the activation of LV5 (left axis) and of the distance to the nearest shore line ($d_\mathrm{land}$, right axis); (b) map of the ship track coloured by the activation of LV5; (c) box and whisker plots of the activated weights.

et al., 2017). Therefore, the grouping of fluorescent aerosol particles with independently measured INP is consistent with expectations. The fact that both types show elevated concentrations near land points towards coastal regions with increased oceanic biological activity and vegetated land being the source (Mason et al., 2015; McCluskey et al., 2018). In addition,

accumulation mode particles ($N_\mathrm{Accumulation}$) exhibit a weak negative loading. It is conceivable that more condensable trace





gases are emitted from land and productive coastal areas that contribute to accumulation mode particle formation. On the continents, human activities lead to emissions, and on subantarctic islands, animal colonies are responsible for ammonia emissions (Schmale et al., 2013), a precursor to secondary aerosol mass.

**In summary**, LV5 demonstrates terrestrial influences on measurements when the ship's track is close to land. Visits to
smaller islands and the larger continental ports coincided with increased concentrations of ice-nucleating and fluorescent particles, as well as frontal systems which influenced air temperature and humidity as well as the altitude at which clouds formed.

### 5.3.2   LV9 - Marginal sea ice zone and snowfall

LV9 has a very distinct regional signal that is mostly active during leg 2 of the cruise, with a clear peak between 27 January and 2 February 2017 when the ship was going through sea ice while approaching and leaving the Mertz region (Figure 12a
and b). Despite its variance being limited to a relatively short part of the expedition, LV9 explains $3.4(\pm 0.6)\%$ of the variance of all 111 variables (Table 2). The largest contribution to this LV comes from the sea ice concentration ($C_i$), i.e. fraction of surface area covered by sea ice (see Figure 12c and weights table for LV9 in the supplementary information). Generally, sea ice cover in the Southern Ocean was unusually low during the austral summer season 2016/2017 (Schlosser et al., 2018) so much that the cruise could go further south than originally planned during large parts of leg 2 (Walton and Thomas, 2018). The only
other region where the ship went through broken sea ice was around Peter I Island, where LV9 has a much smaller and shorter secondary peak on 15 and 16 February 2017 (Figure 12a and b). According to the satellite-derived sea ice concentration, the ship only went through partial ice cover, with a maximum of 48% ice cover in the Mertz region on 31 January 2017.

A second set of prominent signals associated with the positive LV9 periods are a low surface ocean salinity ($S_{\text{sw}}$) and density ($\sigma_{0,\text{sw}}$; Figure 12c). Relatively fresh and light surface waters in conjunction with a partial sea ice cover suggest a stable surface
ocean stratification associated with recently melted sea ice, since sea ice is the dominant driver of the surface ocean salinity and stratification in the high-latitude Southern Ocean (Haumann et al., 2016). While other surface freshwater fluxes such as snow and glacial melt could have been responsible for the low salinity surface ocean in this region, the absence of a low $\delta^{18}O_{sw}$ in LV9 suggests no significant contribution of these fluxes to this LV. This interpretation is confirmed by the dominant sea ice contribution to the surface ocean freshwater flux balance in the Mertz region during ACE (Haumann et al., 2020b). The low
surface ocean salinity and density associated with LV9 might be the cause for other, smaller peaks in the LV9 time series in the absence of sea ice (Figure 12a).

An interesting observation is the large contribution of the wave period ($T_{m-1,1}$) to LV9 (Figure 12c), with a longer surface wave period in the partially ice covered regions. Ice floes in the marginal ice zone dissipate wave energy (Squire, 2020; Ardhuin et al., 2020) with a faster rate for short-wave components of the spectrum (Meylan et al., 2018). The sPCA confirms
these results, with a significantly longer wave period in the partially ice covered region when LV9 is positive, but no substantial effects are noticed for the surface significant wave height.

Snowfall (SR) has the fourth largest contribution to LV9 and occurred both during the time in the sea ice in the Mertz area and in regions outside the sea ice cover in the south Pacific sector (Figure 12a) under varying atmospheric conditions. While a higher contribution of snowfall compared to rainfall is expected near the Antarctic coast in summer, it is unclear if there is a link

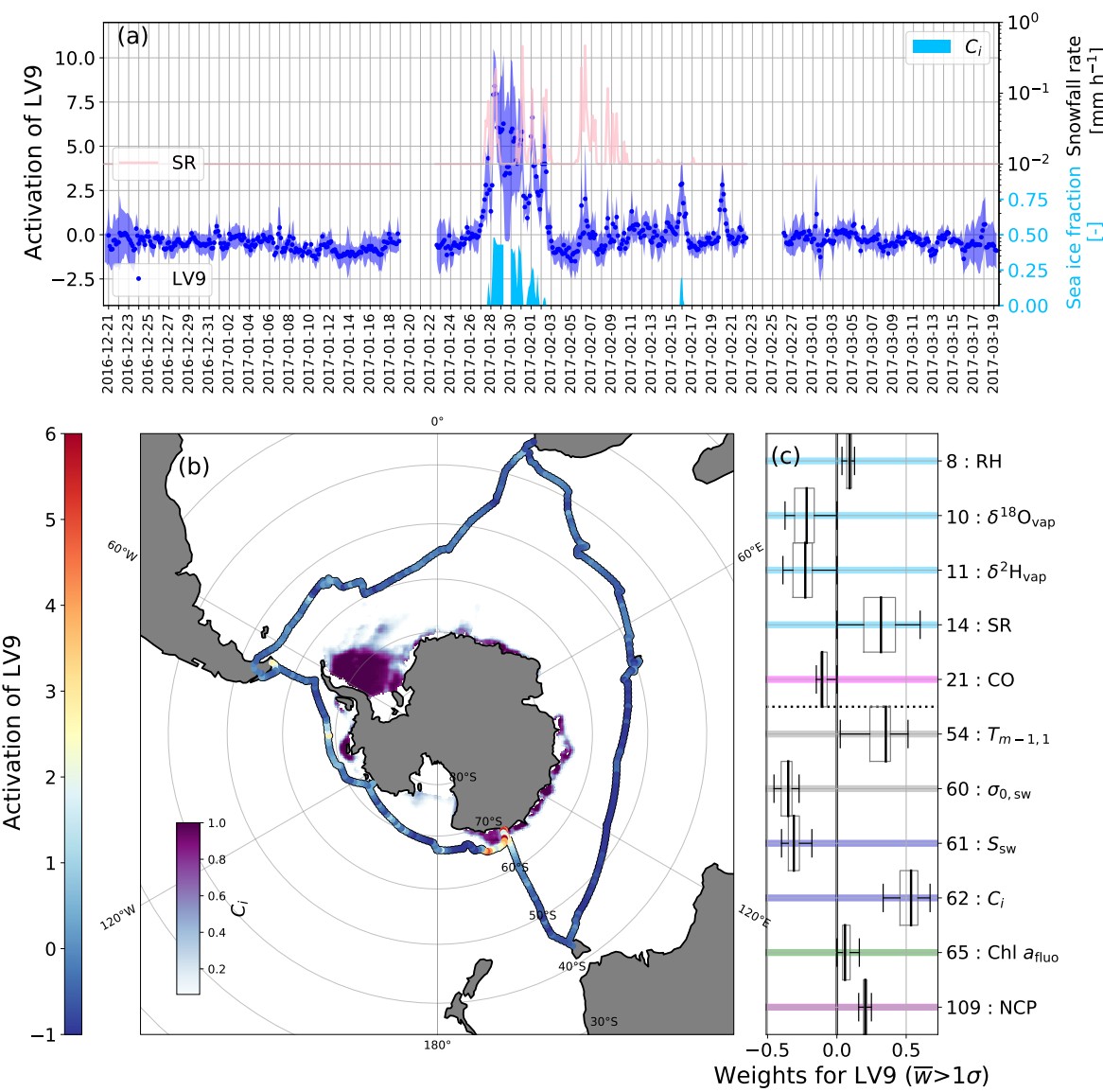

**Figure 12.** (a) time series of the activation of LV9 "Marginal sea ice zone and snowfall" (left axis), sea ice fraction (lower right axis), snowfall rate (upper right axis); (b) the ship track coloured by the activation of LV9 and the sea ice cover during ACE (Peng et al., 2013); (c) box and whisker plots of the activated weights.

between snowfall and the presence of sea ice in LV9. In the Mertz region, where LV9 is active, the two-day back trajectories initiated near the surface are found predominantly over the continent with some contribution from the oceanic regions (see supplementary information section S4). The snowfall occurring during this period is due to the advection of maritime air



masses associated with transient low pressure systems near the Adélie Land coast observed during ACE, as typically found in this area (Servettaz et al., 2020). ACE radiosonde profiles showed a tendency for the RH increasing at upper levels prior to snowfall events, while during snowfall the entire boundary layer was near or at water vapour saturation. This confirms the back trajectory analysis (see supplementary information section S4) showing that the maritime air advection happens in the upper marine boundary layer, while near-surface layers are dominated by a concurrent advection of continental air. The latter vertical atmospheric structure has been observed at the Adélie Coast during the advection of moist air at upper levels in the warm sector of extratropical cyclones (Vignon et al., 2019; Jullien et al., 2020).

The very low abundances of heavy water molecules ($\delta^2$H$_\mathrm{vap}$ and $\delta^{18}$O$_\mathrm{vap}$) of atmospheric water vapour ($w$) in LV9 are likely dominated by low isotopic values observed near the Mertz Glacier and are characteristic for Antarctic air masses. Due to the close proximity to the continent and the presence of sea ice acting as a barrier between the ocean and the atmosphere, it is possible that the continental air mass is only marginally affected by surface fluxes before it reaches the open ocean, which has, for example, been observed in airborne measurements over the Labrador Sea (Renfrew and Moore, 1999). A dominant signal of Antarctic air masses near the surface over the sea ice near the Mertz Glacier would also explain the low carbon monoxide (CO) concentrations in the air, since Antarctic air would be expected to contain less CO. Interestingly, near-surface relative humidity (RH) is slightly higher during positive activations of LV9, which might be caused by (i) moist advection from the ocean during snowfall events, (ii) evaporation from the ocean surface in fractured sea ice zones during advection of continental air and, possibly, (iii) the sublimation of deposited or falling snow. Moreover, the low air temperature above the sea ice can further increase RH due to the temperature dependency of the saturation vapour pressure. More detailed analyses, which go beyond the scope of this study, are needed to assess the relative importance of these processes for the situation represented by LV9.

Net community production (NCP) and phytoplankton biomass (as estimated by the fluorescent chlorophyll $a$ concentration Chl $a_\mathrm{fluo}$) are both positively correlated with LV9. Cassar et al. (2011) showed that NCP in the Southern Ocean was consistently low when mixed layers are deeper than about 45 m, regardless of iron availability, suggesting light limitation to be the driving factor. For shallower mixed layers, NCP correlated with iron sufficiency as estimated by variable fluorescence. In this regard, sea ice melt could influence water column productivity through 1) Fe fertilization (Lannuzel et al., 2008, 2016), and/or 2) enhanced water column stratification, relieving light limitation (Vernet et al., 2008; Cassar et al., 2011; Eveleth et al., 2017). Mixed-layer depths in 32 profiles in the Mertz region were generally shallower than 40 m (average 24 m±11 m). Hence, relief from light or iron limitation due to ice melt could explain the LV9 NCP pattern.

**In summary**, LV9 provides interesting new insights into processes associated with melting sea ice in the marginal ice zone and snowfall during the austral summer. These processes include a salinity driven stable surface ocean stratification, an associated phytoplankton bloom, dissipation of surface waves, and an atmospheric boundary layer that is dominated by Antarctic continental air masses near the surface with moist and warm advection aloft producing snowfall at times.





### 5.4  Atmospheric microphysical and chemical processes

#### 5.4.1  LV2 - Drivers of the cloud condensation nuclei population

LV2 explains the second largest fraction of variance of all 111 OVs ($4.4(\pm0.5)\%$). The activation of LV2 positively correlates with the concentration of cloud condensation nuclei at a supersaturation of 0.15, 0.3 and 1.0% ($N_{\mathrm{CCN},0.15}$, $N_{\mathrm{CCN},0.30}$, and $N_{\mathrm{CCN},1.00}$) very closely (Figure 13a). The most positive activations of LV2 (corresponding to high $N_{\mathrm{CCN}}$) are near the continents, the islands in the Indian Ocean and along the coast of Antarctica in the Pacific sector (Figure 13b). The same locations were identified by Schmale et al. (2019) for $N_{\mathrm{CCN},0.20}$ in a study focused on the ACE aerosol measurements.

The number of available CCN is strongly connected to the particle number in the accumulation mode ($N_{\mathrm{accumulation}}$), particles between roughly $80\,\mathrm{nm}$ and $1\,\mathrm{\mu m}$ in diameter, which is also strongly positively correlated to LV2. Moreover, particulate sulfate ($SO_4^{2-}$) follows the same pattern. It is known to contribute a large mass fraction to the accumulation mode over the Southern Ocean (e.g. Raes et al., 2000; Quinn et al., 2017). There, one important source of particulate sulfate is the pathway via photooxidation of dimethyl sulfide (DMS) emitted by marine microbial activity (Quinn and Bates, 2011; Weller et al., 2011). In parallel to sulfate, methanesulfonic (MSA) is produced and also contributes to accumulation mode particle mass. Schmale et al. (2019) observed a connection between MSA concentration as measured by 24-hour filter samples ($N_{\mathrm{MSA,PM10}}$) and the CCN number concentration. This link is not reflected in LV2. The absence might be due to the low MSA sampling frequency (daily samples), which is a limitation to the here presented analysis (see section 3.5).

Particulate chloride ($Cl^-$), a surrogate for sea spray, and the number concentration of coarse particles, a proxy for sea spray ($N_{\mathrm{seaspray}}$), correlate positively with LV2, indicating their contribution to the CCN budget. Their variability contributes less than those of sulfate and the accumulation mode. While sea spray particles are extremely good CCN, the smaller contribution to LV2 can be expected, because their absolute contribution to the CCN budget is smaller than that of the accumulation mode (Schmale et al., 2019). Large sea spray particles defined here as $> 700\,\mathrm{nm}$ only occur in low number concentrations.

The CCN population is linked to the hygroscopicity of the particle ensemble, which is expressed as the hygroscopicity parameter kappa at different supersaturations (Petters and Kreidenweis, 2007) ($\kappa_{0.15}$, $\kappa_{0.3}$). More in-depth analysis shows that the hygroscopicity decreases with decreasing CCN number Schmale et al. (2019). The positive relation might be caused by the air mass history, i.e. washout of more hygroscopic particles is consistent with the decrease in availability of CCN. The sPCA does not directly account for air mass history because upstream measurements are not available. To check our hypothesis concerning rainout, we investigated the precipitation rate along the backward trajectories for the previous three days (see Figure 14). The 72 h integrated surface precipitation along the trajectories shows high precipitation in situations with low CCN number and sulfate concentrations. This correlation agrees with our hypothesis of a simultaneous decrease in CCN number concentration and the particles' hygroscopicity due to the washout of hygroscopic particles during precipitation. The positive correlation of the weights of cold and warm temperature advection mask ($\mathrm{mask}_{\mathrm{CW}}$) and CCN number concentrations in LV2 seems to be in contradiction to this hypothesis because enhanced precipitation is expected during warm air advection (see also LV3), which would lead to low CCN number concentrations. Furthermore, an opposite correlation of $\mathrm{mask}_{\mathrm{cw}}$ and accumulation mode aerosols in LV2 and LV3 is seen, which could indicate that different processes during warm air advection are important

**Figure 13.** (a) time series of the activation of LV2 "Drivers of CCN population" (left axis) and of the CCN number concentration at 0.15% super saturation ($N_{CCN,0.15}$, right axis); (b) map of the ship track coloured by the activation of LV2; (c) box and whisker plots of the activated weights.

in these two LVs. While LV3 mainly represents the contrasting properties of cold and warm air advection, which includes

the rainout of accumulation mode aerosols during warm air advection, $mask_{cw}$ in LV2 might represent the relevance of cloud processes for CCN number concentrations. The frequent formation and dissipation of clouds during warm air advection might



be conducive to the formation of accumulation mode aerosol and CCN through heterogeneous sulfate chemistry, i.e. addition of sulfate to particles through processes in cloud droplets (Hoppel and Frick, 1990; Schmale et al., 2019). This may lead to a positive correlation of CCNs and the mask$_{\mathrm{cw}}$. Further investigations are needed to clearly identify the relevant processes during

precipitation and in clouds for the CCN number concentrations represented by LV2.

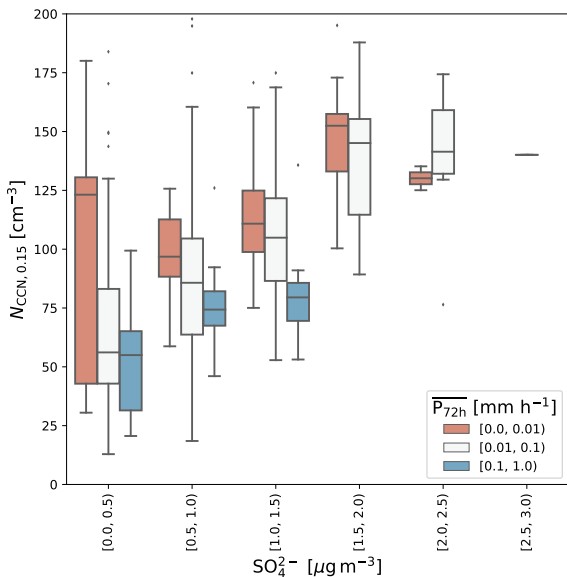

**Figure 14.** Box and whisker plots of $N_{\mathrm{CCN,0.15}}$ as function of the particulate $\mathrm{SO_4^{2-}}$ concentration and the average ERA5 total precipitation rate along the air mass back trajectories over 72 hours before arrival.

**In summary**, LV2 represents the particles and chemical constituents able to act as CCN and the processes leading to variability in CCN number concentrations. For ACE, the CCN number concentrations are strongly related to the accumulation mode and sulfate concentrations. The main process leading to variability in CCN number concentrations most likely is the washout of hygroscopic particles due to precipitation upwind of the measurement site.

## 5.5   LV12 - Wind driven conditions and sea spray aerosol

LV12 groups the OVs related to sea spray aerosols (SSAs) along with some meteorological (wind speed, cyclone flag) and physical oceanographic (significant wave height) variables (see Figure 15). This grouping is consistent with the known mechanisms of sea spray aerosol production (Lewis and Schwartz, 2004): surface winds generate breaking waves and resulting bubble clouds known as whitecaps. When the submerged bubbles rise to the ocean surface and burst, they eject droplets of seawater

into the atmosphere (Monahan et al., 1986). These droplets dry out and form so-called sea spray aerosol. The strong correlation of wind speed ($u_{\mathrm{10N}}$), significant wave height ($H_s$), and SSA number concentrations ($N_{\mathrm{sea\ spray}}$) to LV12 are entirely consis-





tent with these physical processes, as well as the weaker correlation of the cyclone flag variable ($\mathrm{mask_{cyc}}$), which is associated to stronger winds.

**Figure 15.** (a) time series of the activation of LV12 "Wind driven conditions and sea spray aerosol" (left axis) and the ten metre neutral wind speed ($u_{10\mathrm{N}}$, right axis); (b) map of the ship track coloured by the activation of LV12; (c) box and whisker plots of the activated weights.

SSA particles are composed of complex mixtures of sea salt compounds, organic compounds, and marine biological material
(Quinn et al., 2015). Therefore, the mass concentrations of sea salt-related elements measured in aerosol samples also contribute


positively in LV12 (i.e., chloride, Cl$^-$; sodium, $N_{\mathrm{sodium,PM10}}$; magnesium $N_{\mathrm{magnesium,PM10}}$; potassium, $N_{\mathrm{potassium,PM10}}$; sulfate, SO$_4^{2-}$; bromide, $N_{\mathrm{bromide,PM10}}$; calcium, $N_{\mathrm{calcium,PM10}}$, where the latter four have median weights smaller than one standard deviation and are therefore not shown in Figure 15c). This grouping is consistent with previous studies that have observed moderate to strong correlations between the number of SSA particles and the aerosol mass concentrations of elements

comprising sea salt (Modini et al., 2015; Quinn et al., 2017).

In terms of aerosol size distribution modes smaller than sea spray, the concentration of accumulation mode aerosol particles makes a strong negative contribution to LV12 (i.e., it is anticorrelated to wind speed and SSA concentrations). Accumulation mode particles likely contain large fractions of particulate sulfate, produced secondarily, including by aqueous-phase cloud processes (see LV2 section 5.4.1). They are also efficient CCN. The anticorrelation between the concentrations of SSA and

accumulation mode particles in this LV could have potential physical explanations. For example, the onset of windy and stormy conditions conducive to very strong SSA production and net enhancement of SSA concentrations may also result in less photochemical or cloud-processed sulfate production (Fossum et al., 2020), or increased losses of accumulation mode particles through deposition on the ocean surface (Landwehr et al., 2020) or by precipitation scavenging (see discussion of LV3 and LV4 in sections 5.2.1 and 5.2.2), which would all tend to suppress accumulation mode particle concentrations. However,

the observed anticorrelation between SSA and accumulation mode aerosol concentrations could potentially also be explained by a methodological, misclassification artefact related to the mode fitting analysis procedure that was used to separate these two size distribution modes (see supplementary information Section S1).

The relatively large size of airborne SSA droplets and particles means that they are effective at scattering solar radiation, and thereby reducing visibility through the atmosphere. The strong negative weight of visibility in LV12 (i.e. anticorrelation

between visibility and SSA concentrations) is consistent with this physical reasoning. The median light intensity in the mixed layer below the ocean surface ($I_{\mathrm{g}}$) also makes a strong negative contribution to LV12 (i.e., it anticorrelates with SSA concentrations). This OV is derived from measurements of the photosynthetically active radiation available at the ocean surface and the oceanic mixed layer depth (see supplementary information Section S1 and glossary entry for $I_{\mathrm{g}}$ in the appendix section E). Since the measured photoreactive radiation does not contribute to LV12, the negative contribution of $I_{\mathrm{g}}$ to the LV is more likely

a result of a deepening of the mixed layer depth during stormy conditions.

The time series of LV12 activation (Figure 15a) is consistent with a weather-driven process. LV12 was activated episodically as synoptic-scale weather systems passed over the research vessel every 3 to 6 days. This is consistent with previous marine aerosol studies that suggested that SSA production is an episodic process occurring during higher wind speed conditions (e.g. Modini et al., 2015). The map of LV12 activation (Figure 15b) clearly shows that LV12 was most consistently positively

activated during the open ocean legs 1 and 3, where weather systems with strong winds were encountered (see also Schmale et al., 2019). During leg 2 close to sea ice and the Antarctic continent, the baseline of LV12 is shifted and positive activations of LV12 appear for shorter and sharper periods, which may have been related to shifting regional wind directions between the ocean and the continent.

**In summary**, LV 3 describes the production of sea spray due to stormy conditions and the absence of accumulation mode

aerosols in these instances.





### 5.6 Ocean microbial dynamics

Latent variables LV6, LV8 and LV11 work in concert to capture the major physical and chemical features that dictate the dynamics of phytoplankton and bacteria in the sunlit layer of the Southern Ocean, and their biogeochemical activity. Collectively they explain 10.2% of the variance in the 111 OVs (see Table 2), with 59 OVs across the three LVs having weights $\overline{w} > 1\sigma$.

In the Southern Ocean, critical resources for phytoplankton and bacteria include macronutrients such as silicate (captured by LV11) and micronutrients such as iron (Fe; LV6 and LV8 for Fe fertilized and Fe limited productivity, respectively), as well as light availability. However, the distribution and interplay of these resources is spatially and temporally heterogeneous (Boyd, 2002). Both the suboptimally low $F_V F_M$ (Figure 18) and persistent low ratio of dissolved iron:nitrate measured from Cape Town to Punta Arenas (Janssen et al., 2020) suggested generally Fe limited productivity during ACE, as previously reported in the Southern Ocean (Moore et al., 2013).

#### 5.6.1 LV11 - Surface nutrient concentrations associated with mixing events, climatic, and frontal zones

LV11 (Figure 16) activation corresponds primarily to the dissolved concentrations of nitrate (Nitrate) and phosphate (Phosphate) which are positively correlated with the LV activation. The distribution of these nutrients is controlled by physical processes, biological consumption, and respiration/remineralisation (Sarmiento et al., 2004; Freeman et al., 2018). The strongest negative activation of LV11 occurs south of Africa near the Subantarctic Front (SAF) in the Atlantic Sector and near the Subtropical Front (STF) in the Indian Sector. The major ocean fronts in the Southern Ocean physically divide the surface ocean, resulting in distinctly different surface seawater properties in the zones between the fronts. Moving southwards, nutrient-rich deep waters are progressively raised closer to the surface and subsequently entrained into the mixed-layer (Pollard et al., 2002; Weber and Deutsch, 2010), resulting in increased nitrate, nitrite (Nitrite) and phosphate concentrations. Silicate (Silicate) has a lower contribution to LV11 compared to other nutrients, as stable high concentrations are only observed south of the Southern Antarctic Circumpolar Current Front (SACCF) in the signal of LV14.

The biological OVs contributing to activation of LV11 (OVs Chl $a_{\text{fluo}}$, POC, PON, $N_{\text{totalbacteria}}$, $r_{\text{fluo,coarse}3\sigma}$, $r_{\text{fluo,fine}3\sigma}$, explained below) are a product of the nutrient supply to the surface ocean and the complex controls on productivity across the Southern Ocean. Despite high concentrations of macronutrients (OVs Nitrate and Phosphate), iron acts as a key limiting nutrient for biological growth as indicated by low dissolved Fe concentrations (Janssen et al., 2020) and by persistently low photosynthetic efficiency throughout the voyage ($F_V F_M < 0.3$; further explored in LV8). Hence much of the Southern Ocean is classified as a High Nutrient Low Chlorophyll (HNLC) region.

This low growth-high (macro-)nutrients versus high growth-low (macro-)nutrient scenario within HNLC areas drives the anticorrelation observed between nutrient concentrations and the overall biomass 'indicators' chlorophyll $a$ (from fluorescence; Chl $a_{\text{fluo}}$), particulate organic carbon (POC), particulate organic nitrogen (PON), and bacterial abundance ($N_{\text{totalbacteria}}$) in LV11. Chl $a$, POC and PON are the major organic pools synthesised by phytoplankton for use in, or as a result of, their photosynthetic process (Cullen, 1982). Bacterial abundance has a relatively high negative contribution to LV11 (see Figure 5),



**Figure 16.** (a) time series of the activation of LV11 and the concentrations of Nitrate (right axis); (b) map of the activation of LV11; (c) weights of the OVs contribution to LV11 for which the bootstrap median was larger than $1\sigma$. See caption of Figure 6 for details on the oceanic fronts and frontal crossings.

as bacterial concentrations are linked to the availability of dissolved organic matter (a product of particulate organic matter including POC and PON) and nutrients (Church et al., 2000; Kirchman et al., 2009).





The ratio of coarse and fine fluorescent aerosol particles to total aerosol particle numbers in the atmosphere ($r_{\text{fluo,coarse}3\sigma}$; $r_{\text{fluo,fine}3\sigma}$), which are negatively correlated with LV11 activation, are also linked with bacterial concentrations in the Southern Ocean (Moallemi et al., 2021). The likely source of the fluorescent particles is sea spray enriched with organic matter, specifically bacteria (Moallemi et al., 2021).

  The complex interactions between microbial biological OVs and biogeochemical OVs drives the remaining LV11 activation
not described above. Negative activation of LV11 near Mertz and Siple Island was due to enhanced growth of phytoplankton, which also increased biological consumption of nitrate and phosphate, as observed by Janssen et al. (2020) at Mertz. A subtle signature of negative LV11 activation is also observed at locations near subantarctic islands and the Antarctic landmass where increased phytoplankton productivity and growth are indicated by Chl $a_{\text{fluo}}$. The influence of the islands on phytoplankton productivity is explored further in LV6. Conversely, the strongest positive activation of LV11 occurs in the open ocean regions
away from oceanic frontal and landmass influences, particularly evident south of the Polar Front (PF) in the Pacific and Atlantic sectors. These positive activations share similarities with the biological responses captured in LV8 in relation to severe Fe-limitation (captured by $F_V F_M < 0.3$).

  **In summary**, LV11 captures the availability of dissolved macronutrients in the Southern Ocean and the complex relationships between dissolved nutrient availability and the growth and productivity of phytoplankton and bacteria.

### 5.6.2 LV6 - Iron-fertilized biological productivity

Activation of LV6 (Figure 17a) closely follows the patterns in POC, PON and Chl $a$ concentration (TChl $a$ and Chl $a_{\text{fluo}}$; Figure 17b), where increases are attributed to an increase in larger sized phytoplankton. These larger sized phytoplankton include diatoms (DiatA, DiatB) and haptophytes (Hapto67), the latter likely also contributing to the nanoeukaryote population ($N_{\text{nanoeukaryotes}}$). As a result, presence of these taxa increased the overall particle size ($< 100\,\mu\text{m}$) demonstrated by a flattening
(or reduction) in the particle size slope (PSDslope). Diatoms and *Phaeocystis sp.* (a type of haptophyte) are well-known opportunistic bloom forming species in the Southern Ocean, which can rapidly respond to iron replenishment and are widely linked to increases in Chl $a$ and carbon production (Schoemann et al., 2005; Arrigo et al., 2010).

  The strongest activation of LV6 (Figure 17) occurred at biological "hotspots" within the Southern Ocean. These include subantarctic islands and glaciers, specifically Siple Island and Mertz Glacier, where alleviation from iron-limitation is known
to occur due to terrestrial-marine interactions attributed to the island mass effect (IME) (Doty and Oguri, 1956; Blain et al., 2007), sea ice and glacial melting, as well as changes in currents (satellite derived geostrophic flow, $U_g$; Blain et al., 2001; Mongin et al., 2008; Hawkings et al., 2014). Such localized Fe enrichment was measured at Mertz, Balleny and Kerguelen islands during ACE (Janssen et al., 2020). Iron enrichments in the surface waters from the Ross Sea and the Atlantic sector could be related to sedimentary and sea ice inputs (Coale et al., 2004; Lannuzel et al., 2010), while IME and atmospheric dust
deposition could be invoked in the Atlantic sector (Jickells et al., 2005; Cassar et al., 2007). In these Fe enriched "hotspots", the highest concentrations of Chl $a$ and marine particulate organic matter, as well as the highest abundances of diatoms and haptophytes were observed. In addition, the ratio of Chl $a$ degradation pigments to Chl $a$, including pheophorbide $a$ (Pheob$a$) and pheophytin (Phaophy$a$), increased significantly, suggesting increased consumption by zooplankton (Shuman and Lorenzen,



**Figure 17.** (a) time series of the activation of LV6 (left axis) and the pigment biomass attributable to diatom type phytoplankton (DiatB, right axis); (b) the activation of LV6 over a satellite-derived map of monthly average Chl $a$ concentrations (18 December to 16 January for the Indian ocean sector, 17 January to 17 February for the Pacific sector and 26 February to 21 March for the Atlantic sector); (c) weights of the OV contributions to LV6 for which the bootstrap median was larger than $1\sigma$. See caption of Figure 6 for details on the oceanic fronts and frontal crossings.



1975; Ingalls et al., 2006), which can intensify the Fe recycling and iron ability to support net community production (NCP).
In contrast, the ratio of the Chl *a* degradation pigment chlorophyllide *a* (Chlide *a*; an indicator of cell death) to Chl *a* decreased
significantly (Wright et al., 2010).

The vigorous biological productivity associated with LV6 is further supported by activation of LV6 following NCP patterns.
NCP, the balance between photosynthetic carbon fixation by phytoplankton and community carbon consumption by respi-
ration, was generally consistent with trends in Chl *a* and marine particulate organic matter. This is to be expected as NCP
is equal to net primary production (NPP) minus heterotrophic respiration, and Chl *a* generally correlates with NPP. Strong
correlations between NCP and Chl *a* have been reported in earlier studies (Cassar et al., 2011). Exceptions to this appear on
transit between the Balleny and Scott Islands (4 to 5 February 2017) and on transit northward through the Antarctic and Polar
Frontal Zones in the Atlantic Sector (14 to 16 March 2017) where NCP is not correlated with Chl *a* and is also abnormally low
($< -10\,\mathrm{Mmol\,O_2\,m^{-2}day^{-1}}$). Apparent deep mixing in these areas likely results in light limitation leading to a reduction in
NCP as has been observed previously (Cassar et al., 2011).

Activation of LV6 is also driven by seawater transparent exopolymer particles (TEP), coomassie stainable particles (CSP)
and, to lesser extent, acrylate and dissolved isoprene ($C_5H_8$). In addition to synthesising POC and PON, phytoplankton also
produce and release gel-like organics such as polysaccharide-rich TEP (Zamanillo et al., 2019) and protein-rich CSP (Engel
et al., 2020), as well as the biogenic trace gas isoprene, and the organic compound acrylate, which is a non-volatile byproduct
of the production cycle of the trace gas dimethyl sulfide (DMS). As documented in other publications, each of these sec-
ondary compounds are positively correlated with phytoplankton biomass indicators (Chl *a*, POC, PON; Zamanillo et al., 2019;
Rodríguez-Ros et al., 2020), abundances of diatoms (DiatA, DiatB; Zamanillo et al., 2019; Rodríguez-Ros et al., 2020), and
haptophytes (Hapto67; Zamanillo et al., 2019; Rodríguez-Ros et al., 2020; Kinsey et al., 2016), all of which show important
contributions to LV6. Even though the contribution of seawater DMS was very low, there was a considerable positive contri-
bution by one of its atmospheric oxidation products, gaseous methanesulfonic acid (MSA). However, this latter contribution
appears to be mostly driven by a period of high atmospheric MSA concentrations in the vicinity of Bouvetøya Island, coincid-
ing with increased ocean microbial activity. Oxidation of DMS is the only known source of MSA, therefore, it is difficult to
explain a direct causal link between the higher MSA concentration and the enhanced microbial activity without a correspond-
ingly higher DMS concentration, which was not measured in this case. In general, it is rare to observe a direct connection
between microbial activity in the ocean and DMS oxidation products such as MSA and sulfuric acid when using linear corre-
lation analysis. This is due for example to the relatively long lifetime of atmospheric DMS (2 to 5 days in the Southern Ocean)
(Chen et al., 2018) and the influence of various environmental conditions on the different oxidation product yields (Barnes
et al., 2006).

**In summary**, LV6 depicts "hotspots" of biological productivity by phytoplankton and bacteria, which are driven largely by
the island mass effect.





### 5.6.3 LV8 - Iron-limited biological productivity

Many OVs contribute to LV8, which in essence appears to represent the typical HNLC waters of the Subantarctic and Polar Frontal Zones of the Southern Ocean, depicting the resource limited open ocean environment, where biomass accumulation (e.g. total chlorophyll $a$; TChl $a$) and productivity is commonly co-limited by light and iron availability (Boyd, 2002; Boyd

et al., 2007) (Figure 18b), and also potentially by silicate availability north of the SACCF.

In HNLC waters, the microbial community is more diverse than in higher productivity waters (Ishikawa et al., 2002; Wright et al., 2010; Wolf et al., 2013; Cassar et al., 2015; Eriksen et al., 2018), indicated by the inclusion of many different OVs representing different taxa, including prasinophytes (Prasino), haptophytes (Hapto8 and Hapto67), chlorophytes (Chloro), cyanobacteria (Cyano) and Synechococcus ($N_{synechococcus}$), cryptophytes (Crypto), picoeukaryotes ($N_{picoeukaryotes}$), pelago-

phytes (Pelago), and dinoflagellates (Dino). Overall, the abundance of taxa and the concentrations of biogenic compounds such as dimehtylsulphoniopropionate (DMSP), carbon disulfide ($CS_2$) and dissolved isoprene ($Isoprene_{sea}$; Rodríguez-Ros et al., 2020; Rodríguez-Ros et al., 2020), are closely and positively related to TChl $a$ and photosynthetic efficiency ($F_V F_M$ and $\Phi'_{PSII}$). Their positive contributions of LV8 activation suggests that phytoplankton and bacterial growth and productivity in the open ocean are triggered by deeper mixed layers. Indeed, the ocean's physical (significant wave height, $H_s$; wave period,

$T_{m-1,1}$) and dynamical properties (mixed layer depth, MLD; sea-surface height, SSH) are distinctly different in the positive and negative activation of LV8. These OVs suggest that favourable conditions for biological productivity in open ocean iron-deficient waters correspond to a deepening of the mixed layer, perhaps driven by increased wind ($u_{10N}$) (Carranza and Gille, 2015) and wave induced mixing. Although this results in a reduction of median light intensity available within the mixed layer ($I_g$), it is likely an important mechanism for facilitating mixed layer entrainment of deep dissolved iron stores (Janssen et al.,

2020; Carranza and Gille, 2015) that are otherwise inaccessible at locations away from the influences of landmass and sea ice. Increased salinity ($S_{sw}$) and an increase in phytoplankton produced detrital material ($a_{nap}/a_p$; slope of detrital absorption, $a_{napslope}$) with positive activation are additional indicators that the productivity regime depicted by LV8 is focused on the open ocean areas during ACE, in contrast to LV6 which highlighted the IME.

The strong negative activations of LV8 during leg 2, when the ship was in open waters south of the SACCF, corresponded

with the lowest measurements of photosynthetic efficiency ($F_V F_M$ and $\Phi'_{PSII}$), further highlighting the severe iron-limitation that has been documented in waters south of the SACCF.

Interestingly aerosol sulfate ($SO_4^{2-}$) was negatively correlated to LV8 and other atmospheric variables e.g. $CO_2$, $CH_4$, $CO$ and $O_3$. Its inclusion appears to be driven by enhanced values near the coast of Antarctica, likely due to the availability of atmospheric DMS oxidation products and their conversion to particulate mass in cloud droplets (Schmale et al., 2019). The

appearance of the other atmospheric variables is likely coincidental rather than reflecting a causal relationship. LV8 activates positively at lower latitudes, where these trace gases are also more abundant.

**In summary**, LV8 highlights the microbiological regimes associated with the classical HNLC open ocean environments in the Southern Ocean, and emphasizes the importance of water column mixing for accessing deep-stores of essential nutrients for stimulating biological growth.





**Figure 18.** (a) time series of the activation of LV8 (left axis) and of $F_V F_M$ (right axis); (b) map of the activation of LV8 and monthly average Chl $a$ concentrations (18 December to 16 January for the Indian ocean sector, 17 January to 17 February for the Pacific sector and 26 February to 21 March for the Atlantic sector); (c) weights of the OVs contribution to LV8 for which the bootstrap median was larger than $1\sigma$. See caption of Figure 6 for details on the oceanic fronts and frontal crossings.





## 5.7 Solar forcing

### 5.7.1 LV7 - Seasonal signal

LV7 decreases steadily throughout the cruise and appears to be driven by the seasonal trends of a number of original variables, including OVs related to incoming solar radiation, some atmospheric and oceanic trace gases, and nutrients in the water. The strongest contributions arise from atmospheric methane ($CH_4$), ozone ($O_3$), carbon monoxide (CO), median light intensity in the ocean mixed layer ($I_g$) and atmospheric isoprene ($Isoprene_{air}$). LV7 explains 3.5% of the variance.

The decreasing seasonal trend typically observed for $CH_4$ (Dlugokencky et al., 2019), CO (Petron et al., 2019) and $I_g$ over the duration of the cruise (December to March) is mirrored by the overall trend of LV7. Conversely, atmospheric ozone increases over the same timescale (McClure-Begley et al., 2013; Boylan et al., 2015) as captured by the negative weight attributed to $O_3$. The trends of atmospheric trace gases over the timescales of the ACE cruise are primarily driven by the seasonality of atmospheric photochemistry and primary emissions. Minimum abundances of $CH_4$ and CO are observed in the late summer, and appear to lag 1 to 2 months behind the seasonal peak in abundance of the hydroxyl radical (OH), which is the primary sink for both species (Khalil and Rasmussen, 1983, 1990). This lag is attributed to the complex interplay between oxidation reaction rates (driven by OH concentrations and temperature) and seasonality of emissions (e.g., biomass burning peaks in the late winter/early spring in the Southern Hemisphere). CO reaches its minimum during austral summer, as described by Duncan et al. (2003); Té et al. (2016). The seasonal cycle of $O_3$ is anticorrelated with those of CO and methane primarily as a result of the decreasing photolysis frequency of ozone due to decreasing light intensity (Wilson, 2015).

Isoprene dissolved in the ocean ($Isoprene_{sea}$), originating from microbial activity, follows the overall decreasing trend of LV7. This is in line with trends of brominated trace gases of biological origin ($CH_2Br_2$ and $CHBr_3$), which decrease throughout the season as a result of reduced phytoplankton activity (Stemmler et al., 2015). $Isoprene_{air}$ is attributed a negative weight, indicating an increase in the mixing ratio with time, which is in contrast to the trend in $Isoprene_{sea}$. One potential explanation for these opposing trends may lie in the balance between isoprene sources and sinks: while marine isoprene emissions decrease throughout the ACE cruise, as they are driven by incoming solar radiation, phytoplankton biomass, and sea surface temperature (Rodríguez-Ros et al., 2020), the main sink for atmospheric isoprene (reaction with the OH radical) also decreases, potentially masking the effect of a decreasing marine source. The smaller weight attributed to dissolved isoprene might indicate that its decreasing trend is indeed weaker than that in OH abundance. Increasing mixing ratios at these latitudes over similar timescales have also been observed and modelled for other short-chain hydrocarbons (Pozzer et al., 2010). It is also worth noting that the observed trend in atmospheric isoprene may be influenced by terrestrial sources, as the lifetime of isoprene increases with increasing latitude (e.g., from about 2 h at $40°$S to more than 10 h south of $55°$S) based on numerical simulations by Ferracci et al. (2018) and, coupled with high wind speeds, would enable long-range transport of isoprene from landmasses. The entanglement of marine and land sources might also explain why $Isoprene_{air}$ does not occur in LV5 (distance to land) despite the fact that there are known terrestrial sources of isoprene on many of the visited islands and continents.

Trends in the oceanic variables particulate organic carbon (POC) and the ratio of POC to particulate organic nitrogen (PON; C:N) as well as the slope of the marine particle size distribution (2 to 20 μm; PSDslope), which follow the LV activation

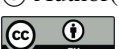

**Figure 19.** (a) time series of the activation of LV7; (b) map of the ship track coloured by the activation of LV7; (c) box and whisker plots of the activated weights for which the bootstrap median was larger than $1\sigma$.

pattern, i.e., decreasing over time, signal the end of the productive phytoplankton season. POC and C:N decline to a minimum at the end of the summer season as surface macro- and micro-nutrients are progressively drawn-down and exhausted, and the available light decreases (Llort et al., 2015). The decline in PSDslope over time indicates that the proportion of smaller and larger particles in the surface ocean have become more equal, representative of seasonal succession in phytoplankton where





community taxonomic diversity and size heterogeneity increase after the productive peaktime in phytoplankton biomass where the community is generally dominated by a few taxa (Lourey and Trull, 2001; Eriksen et al., 2018). Interestingly, dissolved
silicate (Silicate) appears to increase throughout the duration of the ACE cruise, despite documented declines in surface silicate concentration approaching autumn (Freeman et al., 2018). This increase may be due to reduced biological drawdown of silicate, which is controlled by diatom activity or may be due to increased time spent south of the Southern ACC Front in the Pacific and Atlantic sectors, where surface waters are more replete in silicate upwelled from deep waters. Net community production (NCP) also contributes to this seasonal LV with a decreasing trend over time but it is not captured by the diel cycle LV10,
which is likely due to the long integration time of oxygen and hence a lack of a pattern at short timescales.

The sea level pressure at high latitudes in the Southern hemisphere decreases from December to March (Walland and Simmonds, 01 Dec. 1999), which is reflected in the seasonal pattern of LV7. The seasonal decrease in the temperature difference between air and sea surface ($\mathrm{mask_{cw}}$) indicates that the air temperature becomes cooler relative to the sea surface temperature as the summer progresses into autumn. This change of the temperature gradient towards negative values is thought to arise
from an increase in cold air advection during late summer/early autumn, which could stem from either an increase in cyclone frequency (leading to more atmospheric meridional large-scale advection), or an increase in the vertical intrusion of dry air masses north of 40°S.

The hygroscopicity of CCN activating at a supersaturation of 1.0% ($\kappa_{\mathrm{CNN,1.00}}$) increases slightly over time, meaning that atmospheric Aitken mode particles might have changed in composition. Due to the inherent difficulty, the chemical composition
of these particles was not measured. Information about the chemical composition would point towards a source of these particles, which in turn would allow for an interpretation. Hence, based on the available data no explanation for the occurrence of this OV in LV7 can be given.

**In summary**, LV7 confirms our understanding of the seasonal trends of a number of variables, including atmospheric and dissolved trace gases, incoming solar radiation, atmospheric pressure and both oceanic and atmospheric particulate matter. The
seasonal behaviour of atmospheric isoprene and dissolved silicate, while initially somewhat counterintuitive, might point at the complex seasonal and marine-terrestrial interactions driving some of the OVs.

### 5.7.2 LV10 - Diel cycle

LV10 describes the diurnal cycle. It peaks during the day and decreases during night following solar radiation, which was measured across visible and UV wavelengths in $S_{\mathrm{in}}$ and for photosynthetically active wavelengths only (PAR, 400 to 700 nm),
which are the two main contributors to LV10. Moreover, the amplitude of LV10 decreases over the course of the expedition as a result of summer progressing into autumn and the consequent reduction of the amplitude of the daily cycle, which is in line with the reduction of solar radiation reaching the Southern Hemisphere. Active ($\Phi'_{\mathrm{PSII}}$) and passive (Chl $a_{\mathrm{fluo}}$) chlorophyll fluorescence respond to the diurnal light cycle predictably, decreasing through the day as irradiance increases (Yentsch and Ryther, 1957; Falkowski and Kolber, 1995), hence the opposite signs of $\Phi'_{\mathrm{PSII}}$ and Chl $a_{\mathrm{fluo}}$ compared to the solar radiation
OVs. Under increased irradiance phytoplankton dissipate excess light energy through non-photochemical quenching (Krause and Jahns, 2004; Browning et al., 2014), reducing their passive fluorescence signal and photosynthetic efficiency. The physi-





ological response can remain for hours after the peak irradiance period, resulting in a slight offset in the diurnal cycle to the solar radiation cycle and lower contribution to the LV.

**Figure 20.** (a) time series of the activation of LV10 "Diel cycle" (left axis) and of the incoming solar radiation ($S_{in}$, right axis); (b) map of the ship track coloured by the activation of LV10; (c) box and whisker plots of the activated weights for which the bootstrap median was larger than $1\sigma$.





Sulfuric acid and iodic acid ($H_2SO_4$ and $HIO_3$) also contribute to LV10, as they are both photochemically produced and
have a short lifetime (minutes to hours). In the remote marine boundary layer, sulfuric acid is photochemically produced from
DMS oxidation products (*i.e.* the reaction of $SO_2$ with OH or the thermal decay of the $CH_3SO_3$ radical). Observations of the
sulfuric acid diurnal cycle have been reported both over the Southern Ocean and on the Antarctic plateau (Lucas and Prinn,
2002; Jefferson et al., 1998). The formation pathway of iodic acid in the atmosphere is still not resolved and observations are
scarce (Sipilä et al., 2016; Baccarini et al., 2020; He et al., 2021). However, iodic acid would not form without the iodine
radical, which can be produced by photolysis of different precursors (e.g. $I_2$ or $CH_2I_2$) (Saiz-Lopez et al., 2012) and explains
the contribution of iodic acid to LV10.

Other minor contributors to LV10 are the cold and warm temperature advection mask ($mask_{cw}$) and the sky cover (SC). The
contribution of $mask_{cw}$ in this case does not indicate any large-scale advection but might be the result of a stronger diurnal
cycle in the air than in the ocean surface temperature, which can leave a signal of more warm advection during the day than
during the night. The contribution of SC has an opposite sign compared to solar irradiance because a higher cloud coverage
would decrease the amount of solar radiation that can reach the surface.

**In summary**, LV10 clearly represents the diurnal cycle driven by solar radiation. OVs such as atmospheric trace gases and
marine microbial activity that depend on solar radiation feature strongly in this LV.

### 5.8 Short summary of all latent variables

The sPCA solution describes 55% of the variability of the 111 OVs with 14 LVs. The largest signal by far originates from
the large-scale horizontal temperature and pressure gradients that exist between the low and high latitudes. The effect of these
gradients on physical properties of the surface ocean and its activity are mostly captured in the two climatic zone signals (LV1
and LV14). The meridional distribution of the nutrient availability and its effect on the productivity is further highlighted in
LV11, LV6 and LV8. The meridional temperature and pressure gradients give rise to the meridional advection of cold and warm
air (LV3) with implications on cyclone activity (LV13) and the freshwater cycle with the intermittent character of precipitation
events (LV4). The sPCA solution also clearly highlights aerosol sources (especially for INP and fluorescent aerosol) on or in
the proximity of islands and continents (LV5) and the positive effect of sea ice on microbial productivity (LV9), as well as
the effect that both land and sea ice have on precipitation patterns. We observe a clear link between wind speed and sea state
and the concentration of large sea spray aerosol (LV12), tying them to the most wind-driven regions of the Southern Ocean.
In contrast to that, the smaller accumulation mode particles (LV2) are ubiquitous, because of their long lifetime and various
source processes contributing to their abundance. The sPCA successfully decouples the high spatial and temporal variability of
iron-limited (LV8) and iron-fertilized blooms (LV6) and their dependence on nutrient availability (LV11), helping to identify
the factors responsible for changes to the biogeochemistry and microbial community structure. The method further highlights
the effects of diurnal variability of solar forcing on phytoplankton photosynthetic efficiency and trace gas oxidation (LV10)
as well as that of the seasonal variation of the solar forcing on dissolved as well as atmospheric trace gas concentrations and
seasonal cycle in microbial productivity (LV7).





## 6   Discussion

In the previous sections, the individual LVs were discussed following the interpretation of the OVs' spatio-temporal varia-
tions. We discussed them in specific themes following the order of large-scale circulation, atmospheric and oceanic advection,
geographical effects, atmospheric chemical processes, marine microbial dynamics and solar forcing.

Here, we highlight features identified by the sPCA, which occur across several LVs and OVs: (i) particular geographical
locations ("hotspots"), where many LVs responded, (ii) LVs which give insight into atmosphere-ocean interactions, and (iii)
OVs which contribute to variability on many spatial and temporal time scales, i.e. they appear in many LVs. We also discuss the
limitations of the sPCA method and how these might have influenced the outcome. The three main drawbacks of the method
are (a) the absence of an underlying temporal model, which favors direct correlations in time and space, (b) the linearity
assumption, which cannot reveal nonlinear processes, and (c) OVs represented by sparse data that might not feature in LVs,
which does not imply that they are not part of a process.

### 6.1   Hotspots of latent variable activation

The dimensionality reduction achieved by the sPCA allows for visual inspection of the joint variabilities of variable groups that
are provided in the LV-time series. Periods where several of these groups show large coinciding variabilities are of particular
interest as they may indicate local "hotspots" of biological activity or events that fall outside the "normal" variability. In
Figure 21, "hotspots" are indicated along the cruise track during which a minimum of four LVs strongly responded. We
grouped "hotspots" into three types.

**Aerosols and precipitation.** The first "hotspot" (P1), which we call "strong precipitation even", coincides with the visit
to the Prince Edward Islands (26 to 28 December 2016). Heavy and prolonged rainfalls (LV4) coincide with a reduction
of Aitken (LV14), accumulation (LV2) and coarse mode aerosol concentrations (LV12). The observed decrease in aerosol
concentrations are lagging up to 12 hours behind the observed precipitation rates. This time lag is likely due to the fact that
most particles are not removed through interception with falling rain, but rather through activation in the cloud layer (Seinfeld
and Pandis, 1998) and vertical mixing is required before the depleted air can be observed near the ground. Therefore, a time lag
in the order of a few hours is conceivable (Lewis and Schwartz, 2004). However, due to the heterogeneity of the precipitation
patterns, the depletion in the aerosol concentrations may have originated from rainfall events other than the ones observed.
A similar, but much shorter "hotspot" (P2), with activations of the same LVs occurred near the Kerguelen Islands (3 to 4
January 2017). Nine further strong rain and snowfall events (P3 to P8 and S1 to S5 in Figure 21b) with precipitation rates
$>0.1\,\mathrm{mm\,h^{-1}}$ are less clearly reflected in the three aerosol related LVs. There are also three occurrences where LV4 shows
strong negative activation (driven by low visibility) and LV12 and LV2 strong negative activations (few particles) during rather
weak precipitation events (X1 to X3 in Figure 21b). In general, the time series of LV2 and LV12 show stronger resemblance in
periods with low aerosol concentrations than for high concentrations. We interpret this as a relatively stronger similarity in the
sink processes of accumulation and coarse mode aerosols than in their sources.


**Advection of Antarctic air.** Hotspot A1 (Figure 21c) was observed near Mertz Glacier, where the ship stayed from 27

January until 2 February 2017 (see section 5.3.2). The advection of cold Antarctic air masses (12:00 UTC on 28 January until

12:00 UTC on 30 January 2017) led to a sudden drop of the air temperature to the lowest value encountered during ACE

$(-10\,°C)$. This drop in temperature is reflected in the lowest values of LV1 indicating the greatest polar influence observed and

LV3 indicating the strongest air-sea temperature gradient during the cruise. At the same time, conditions were dry as indicated

by the negative activation of LV4. During the event, a strong increase in the Aitken mode particle number concentration

$(N_{Aitken})$ over the otherwise low concentrations in the Pacific sector was observed (see Figure 7). In fact, this event constitutes

the most pronounced difference in the time series of LV1 (climatic zones and large-scale horizontal gradients) and LV14

(hotspot-driven climatic zones).

**Hotspots of ocean productivity.** There are a number of well-known ocean productivity "hotspots" in the Southern Ocean

near Subantarctic islands and the Antarctic continent. Two "hotspot" locations were observed in LV6 and LV14, at Siple Island

(B1), and near the Mertz Glacier (B2) in Figure 21d. These "hotspots" are fueled by local iron enrichment due to the effects

of topography and sea ice melt as described in section 5.6.2 and section 5.3.2. The result is increased productivity, microbial

biomass, and other secondary products such as gel-like organics, protein-rich particles and trace gases.

## 6.2 Atmosphere-ocean interactions

Figure 22 shows how many categories were activated in each LV. For example LV4 contains only OVs from the two categories

*Atmospheric dynamics and thermodynamics* and *Atmospheric side of the hydrological cycle* (Figure 22d), while LV1 contains

OVs from all categories except *Topography* (Figure 22a). The categories contain different numbers of OVs. In order to make

activations comparable they are shown as ratios of the activated OVs per category over the total number of OVs per category.

In most LVs, we find a coinciding activation of variables in the *Atmospheric dynamics and thermodynamics* and in the

*Oceanic dynamics and thermodynamics* category, which are related to local coupling of wind and waves, larger-scale variations

of air and water temperature, and characteristics of the ocean currents. Such a relation between the atmosphere and ocean is,

however, missing for precipitation (LV4; Figure 22d), incoming solar radiation (LV10; Figure 22j), and cyclone activity (LV13;

Figure 22m). These LVs only activate OVs from the *Atmospheric dynamics and thermodynamics* category, but not from the

*Oceanic dynamics and thermodynamics*. One possible explanation for the absence of a clear influence on the ocean is that

the precipitation (LV4) and the diurnal cycle (LV10) represent strong variation of atmospheric OVs on time scales of less

than a day, which might be too short to trigger considerable oceanic variability of detectable strength. Likewise, the effects

of mesoscale air-sea interaction processes (varying wind speed and direction) on the variations in the sea state OVs are much

stronger than the effects of the large scale synoptic (cyclonic) features, which are represented by LV13.

For the *Oceanic* and *Atmospheric hydrological cycles*, we find a similar pattern. Links between ocean and atmosphere are

visible for LVs with a strong low-frequency ($> 1\,month$) component like the climatic zones (LV1; Figure 22a), the seasonal

signal (LV7; Figure 22g), and intermediate frequencies (in the order of days) such as sea ice cover (LV9; Figure 22i), and cy-

clone activity (LV13; Figure 22m). LVs which happen on short time scales, for example strong precipitation related variations

of LV4, trigger only a weak ($\overline{w} < 1\sigma$) marine reaction, e.g. in the local surface water salinity (see Figure 9), which predom-

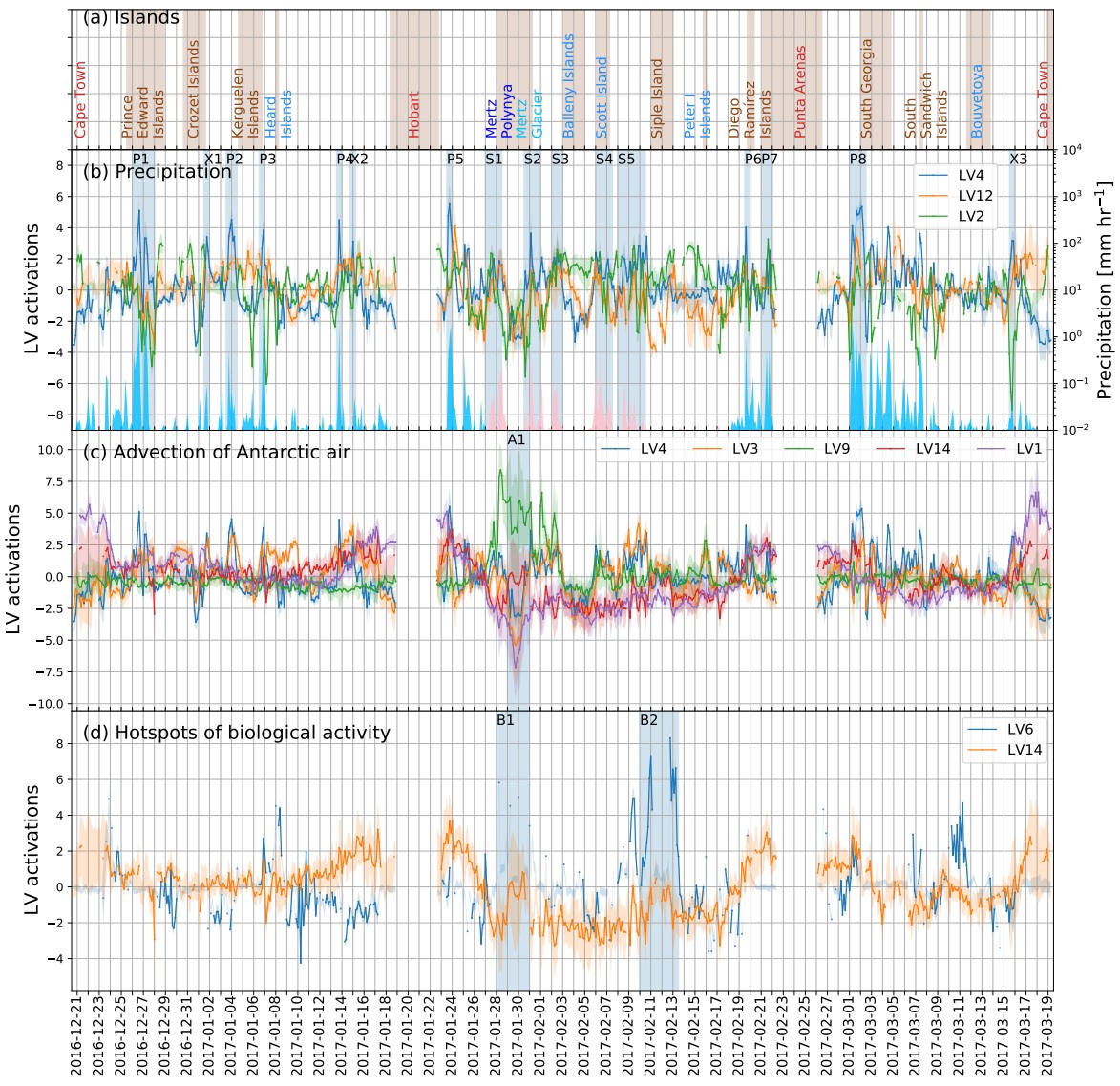

**Figure 21.** (a) indicators for the islands; (b) time series of LV4, LV12 and LV2 with precipitation "hotspots" (rainfall P1 to P8, snowfall S1 to S5) as well as situations of reduced visibility (X1 to X3); (c) time series of LV4, LV3, and LV9 with the advection of Antarctic air (A1); (d) LV6 and L13 with the two biological "hotspots" (B1 and B2).

inantly varies over larger spatio-temporal scales due to mixing processes, the cumulative effects of rainfall and evaporation (Dong et al., 2009; Ren et al., 2011), sea ice formation and melting (Haumann et al., 2016), and glacial meltwater (Jacobs, 2002).




Coupling of OVs from the *Oceanic biogeochemistry* and *Oceanic microbial community* categories with OVs from the *Atmospheric chemistry* category is rare. The relationship between DMS-emitting microbial communities, trace gases, and aerosol chemical composition is long known. However, a direct correlation, as would be discovered by the sPCA, cannot be expected, due to the timescales involved. Atmospheric DMS oxidation in the Southern Ocean is estimated to take 2-5 days (Chen et al., 2018) and during this time the air mass will move away from the microbial activity area. Hence, a direct correlation is not observed. A hint of the connection between microbial activity and atmospheric DMS concentrations is given by the higher MSA concentrations at higher latitudes (see LV1 section 5.1.1) and by the positive weight of MSA in particulate matter smaller than 10 $\mu$m in LV6, which describes the occurrence of iron-fertilized plankton blooms (see section 5.6.2).

The above observations show that our analysis targets processes that manifest themselves in rather local correlations, such as the established link between wind speed and sea state or correlations based on smooth variations over time and space, such as the large-scale horizontal gradients in the air and sea water temperature and the hydrological cycle. To include processes occurring with a time lag or those affected by transport across larger scales, the coupling with air mass back trajectory analysis provides a valuable extension to infer potential relations of the observed signals with up-wind conditions and air mass history, for example the advection of cold or warm air (see section 5.2.1), or the removal of accumulation mode aerosols during successive precipitation events (see section 5.4.1). Note that these findings would not change fundamentally if we were to choose a coarser categorisation of the OVs by merging *Atmospheric/Ocean dynamics and thermodynamics* and *Oceanic/Atmospheric hydrological cycles categories*.

## 6.3 Key original variables

Figure 5 shows the scaled covariances ($\text{cov}_{\text{scaled}}$; Eq. 4) between the OVs and LVs as stacked barplot for each of the OVs. The $\text{cov}_{\text{scaled}}$ provides a measure of the contributions of the LVs to the reconstruction of an individual OV. We observe that most OVs are related to more than one LV. The inclusion of an OV in multiple LVs can occur for several reasons: (i) the OV may be important for or affected by a number of processes and can thus be seen as key variable in the dataset; (ii) derived variables are more likely to occur in several LVs as they are strongly correlated to the multiple observed variables from which they were constructed; (iii) coincident correlations: while they cannot be ruled out completely, they are not very likely due to the three-month long observation periods. Table 3 shows OVs that occur in four or more LVs with weights that satisfy $\overline{w} > 2\sigma$, i.e., where the model assigns a high level of significance to the correlation between OV and LV. Here, we discuss these top seven OVs.

The ranking is led by the cold and warm temperature advection mask ($\text{mask}_{\text{CW}}$), which features in five of the 14 LVs. The frequent occurrence of $\text{mask}_{\text{CW}}$ in the LVs highlights the importance of the air-sea temperature difference, which describes the thermal dis-equilibrium between the ocean and the atmosphere. The $\text{mask}_{\text{CW}}$ correlates strongest ($\text{cov}_{\text{scaled}} = 0.33$) with LV3, which relates to the meridional advection of cold or warm air masses (see section 5.2.1), and secondly ($\text{cov}_{\text{scaled}} = 0.17$) with LV5, which relates to the ship's location relative to the nearest land (see section 5.3.1). The sPCA results also isolate a weak seasonal trend (LV7) ($\text{cov}_{\text{scaled}} = 0.06$) and diel variations ($\text{cov}_{\text{scaled}} = 0.02$) of $\text{mask}_{\text{CW}}$ (LV10).

**Figure 22.** (a) Radar charts of the activations of the OV-categories in LV1. The angular component and marker colour denote the categories and the radial component denotes the number of active OVs ($\overline{w} > 1\sigma$) per category divided by the total number of OVs per category (see (o) as legend). The absence of a marker denotes zero OVs of the respective category were activated; (b)–(n) same as (a) but for the remaining LVs; (o) provides the category labels and the axis label of the radial axis.

Like $\mathrm{mask_{CW}}$, the derived variable, median light intensity within the ocean mixed layer ($I_\mathrm{g}$) shows ($> 2\sigma$) contributions to

five LVs, which is due to the inclusion of other OVs in the calculation of $I_\mathrm{g}$ (see supplementary information section S1.9 for





**Table 3.** OVs which occur in 4 or more LVs with ($\overline{w} > 2\sigma$).

| Symbol | ID | List of LVs with $\overline{w} > 2\sigma$ | No. LVs |
|--------|-----|------------------------------------|---------|
| $\text{mask}_{\text{CW}}$ | OV4 | LV3, LV5, LV6, LV7, LV10 | 5 |
| $I_{\text{g}}$ | OV59 | LV5, LV2, LV6, LV7 | 5 |
| $u_{10\text{N}}$ | OV1 | LV3, LV5, LV8, LV12 | 4 |
| RH | OV8 | LV3, LV4, LV5, LV9 | 4 |
| $P_{\text{air}}$ | OV2 | LV1, LV4, LV7, LV13 | 4 |
| $U_{\text{g}}$ | OV56 | LV1, LV5, LV6, LV14 | 4 |
| SSH | OV55 | LV1, LV7, LV8, LV14 | 4 |

the calculation). For example, the strong link with the seasonal signal (LV7) reflects the decreasing solar radiation intensity and period over the progression of ACE. The anticorrelation of $I_{\text{g}}$ with the wind driven conditions (LV12) likely results from the deepening of the oceanic mixed layer in stormy conditions. Also the decrease of $I_{\text{g}}$ closer to land (LV5) is likely caused by the increased light attenuation of the higher particulate matter concentrations in the surface ocean.

Surface wind speed ($u_{10\text{N}}$) is strongly linked to four LVs. In LV12, it drives the sea spray aerosol concentration ($N_{\text{seaspray}}$). LV8 (iron-limited biological productivity) reveals a positive correlation of wind speed, significant wave height ($H_s$) and period ($T_{m-1,1}$) with biological productivity indicators, because the microbial community profits from an increased nutrient supply due to the stronger vertical mixing in rougher sea conditions (Carranza and Gille, 2015). The sPCA resolves the higher probability of high wind speeds during cold air advection compared to warm air advection (see LV3). As discussed in section 5.3.1

the weak anticorrelation of wind speed and the distance to land ($d_{\text{land}}$) may be either due to orographic enhancement of wind near land or the coincidence of some island visits with the passage of storms.

Due to their meridional pattern, the satellite-derived sea-surface height (SSH) and surface ocean geostrophic velocity ($U_{\text{g}}$) both occur in LV1 and LV14, which relate to the climatic zones and large-scale horizontal gradients. $U_{\text{g}}$ is also affected by the sea bed topography (LV5), while a seasonal change in SSH is related to the change in water temperature (LV7). In addition,

the two OVs are linked to microbial activity. $U_{\text{g}}$ shows relations to the observed patterns in iron-fertilized productivity as represented by LV6 with high productivity occurring close to land masses where currents are weaker and productivity is stimulated by the island mass effect (Doty and Oguri, 1956; Blain et al., 2007). The activation of SSH in LV8 shows that distinctive changes to the ocean dynamics and thermodynamics in open waters during ACE, represented by SSH and other OVs in that category, appear to be important for facilitating the re-supply of much-needed dissolved iron to the iron-starved

surface microbial community.

The appearance of atmospheric pressure ($P_{\text{air}}$) in several LVs (LV1, LV4, LV7 and LV13) reflects the importance of atmospheric pressure, and especially atmospheric pressure gradients, in shaping variations in large-scale atmospheric dynamics. The overall meridional pressure gradient (LV1) is modulated by synoptic-scale variability due to the life cycle of cyclones and





anticyclones over the Southern Ocean. Measurements of atmospheric pressure are, thus, indicative of cyclone activity (LV13)
and the passage of the cyclones' cold and warm front and related precipitation events (LV4). Seasonal variations in surface
pressure are further described by LV7.

In summary, we find that the sPCA is not only capable of resolving many of the complex connections between the OVs
but also to provide estimates of their relative importance for the observed variability of each OV. We find that Earth system
state variables such as the air-sea temperature difference, wind speed, sea-surface height, surface ocean geostrophic velocity,
atmospheric pressure and median light intensity in the ocean mixed layer are critical for the identification of Southern Ocean
processes and they relate to chemical and biological processes in the atmosphere and ocean. We suggest that these variables
should be given high importance in the planning and execution of future large-scale research campaigns, in long-term obser-
vational networks, and satellite-based monitoring, such that numerical models can be assessed in their capability of accurately
describing these key variables.

## 7   Conclusions

We applied the unsupervised machine learning method sparse Principal Component Analysis to a heterogenous dataset of 111
original variables (OVs) from the Southern Ocean, which were measured during the three-month long Antarctic Circumnav-
igation Expedition. These variables describe the physical, chemical, and biological state of the surface ocean and the lower
atmosphere during the 2016/2017 austral summer season. Scientific interpretations are given for the 14 latent variables resulting
from the sPCA. Together they explain 55% of the total variance of the 111 OVs.

The resulting 14 latent variables offer a new statistical perspective on relationships between physical, chemical, and biolog-
ical processes, as well as air-sea interactions over the Southern Ocean, in line with existing knowledge. Our results describe
processes in the following domains: large-scale circulation (LV1, LV14), atmospheric and oceanic advection (LV3, LV4, LV13),
geographical effects (LV5, LV9), atmospheric chemical processes (LV12, LV2), marine microbial dynamics (LV6, LV8, LV11)
and solar forcing (LV7, LV10). We classified the OVs into 8 categories compromising *oceanic-* and *atmospheric dynamics and
thermodynamics*, the *oceanic-* and *atmospheric side of the hydrological cycle*, *atmospheric chemistry* and *ocean biogeochem-
istry*, as well as the *oceanic microbial community* and *topography*. Most of the LVs include oceanic and atmospheric OVs from
multiple categories, which supports the notion of the Southern Ocean as a heavily interconnected system.

Our large survey of the Southern Ocean and sPCA analysis reaffirmed the important role of the oceanic circulations and
frontal zones in shaping the nutrient availability, which controls biological community composition and productivity (LV6,
LV8, LV11, LV14). We identified a strong regional impact of sea ice on sea water salinity, on the dampening of surface
waves, and on increased phytoplankton growth and net community productivity (LV9). This strong control of the sea ice on
the ocean points towards important impacts that possible future sea ice changes in the region could have on the physical and
ecological system. Various atmospheric chemical regimes were identified. For example, LV12 establishes the link between
large sea spray particle concentrations and heavy sea state and hence the region of the westerly wind belt. LV2 describes the
dominant and ubiquitous role of accumulation mode aerosols for cloud seeding, while LV14 illustrates the negative latitudinal


gradient of the Aitken mode particles, with modulations by local "hotspot" sites near coastal Antarctica and certain islands. A number of further "hotspots" were identified across several LVs. These represent specific features such as strong precipitation, cold air mass outbreaks, and the presence of sea ice and islands, all with implications for atmospheric and marine processes.

While it is beyond the scope of this work to analyze these events in more detail than provided in the discussion section, it appears that a better understanding of the types, timescales and implications of processes at the "hotspots" is needed. The identification of "hotspots" demonstrates the ability of sPCA to highlight outstanding features across the Southern Ocean. Seven OVs contributed to four or five LVs and were hence interpreted as key variables. They include the air-sea temperature difference, upper ocean light intensity, wind speed, relative humidity, atmospheric pressure, oceanic geostrophic velocity and

sea surface height. We suggest that these variables should be given high importance in future research campaigns, long-term observational networks, and satellite-based monitoring, such that they can be used to evaluate numerical models in their capability of accurately describing Southern Ocean processes.

     The interpretation of the results requires the combination of expert knowledge on the various original datasets and the components of the environmental system that they describe. At the same time, the sPCA results provide an ideal basis for the

interdisciplinary exploration of multivariate datasets, because they can be used to visualise the complex relations between the OVs in an accessible way. The linearity, which may be seen as a strong limitation of the method, is a strength in this context, because it warrants full traceability of the decomposition weights and LV activations back to the original variables.

     Our extension of the sPCA method to estimate uncertainty with the bootstrap approach reduces the influence of spurious correlations caused by measurement errors or extreme events, which cannot be properly accounted for within the linear frame-

work of the method. The uncertainty estimates proved to be valuable information for the interpretation process, as they allowed the separation of robust and spurious correlations. In combination with the iterative imputation of missing observations, the uncertainty analysis makes the method particularly suited for real-world data with gaps and outliers. We therefore recommend this method for further application to environmental datasets.

     We find that sPCA is capable of resolving many of the complex connections between the OVs and of providing estimates of

their relative importance for the observed variability of each OV. On the one hand the sPCA can be used to find relationships between observed quantities which appear jointly in a latent variable. On the other hand, one can also analyse how important environmental processes, i.e. latent variables, are for the variability of the observed variables, by reconstruction of the OVs from the LVs. Examples are the effects of meridional variations (LV1, LV14), enhanced biological production near melting sea ice (LV9), and the island mass effect (LV6) on net community production (see Figure 5). In combination, these two

steps make sPCA a powerful tool for the exploratory analysis of multivariate datasets. At the same time we note that, while many relationships can be identified by sPCA, some cannot be resolved, because they do not establish themselves as direct correlations between the observed variables due to time lags or insufficient data availability. For example, the contribution of biogenic trace gas emissions to the chemical aerosol composition remains unidentified due to the timescales of the related atmospheric chemical reactions and the coarse time resolution of some of the data.

There is no explicit underlying spatial or temporal model in sPCA, such that the method is capable of resolving multiple important co-existing regimes within the temporal and spatial dimension. For example, within the spatial component there are



a number of separate regimes simultaneously influencing microbial dynamics and biogeochemistry, such as large meridional changes in nutrient supply from upwelling versus local advective and other input processes near land masses or melting sea ice. To date, multivariate models, which can cope with spatial and temporal components at the same time and provide such

simple statistical output for interpretation are rare. Therefore, we believe that the approach presented here extends the use of data science in environmental disciplines by providing enhanced interpretation of connected processes (Blair et al., 2019).

*Code and data availability.*   The python code which was used for the analysis and to create the plots is available at https://renkulab.io/gitlab/ACE-ASAID/spca-decomposition and as a Renku project https://renkulab.io/projects/ACE-ASAID/spca-decomposition. For the availability of the data used in this study please refer to tables A1 to A11 in the Appendix section A.

**Appendix A:  Original variables**

This section provides tables A1 to A11 that list the OVs sorted into the 8 categories. Each table provides the original variable IDs, the symbols, the SI units, the full name and a short description of the variable, the input normalisation used for the sPCA analysis (see section 3.3), as well as reference to the published dataset or to the methods section S1 in the supplementary information.





**Table A1.** Original variables used in this study, which fall in the category *Atmospheric dynamics and thermodynamics*. The columns provide the original variable ID, the symbol, the SI units, a short description of the variable, the input normalisation used for the sPCA analysis, as well as a reference to the published dataset or to the methods section.

| OV-ID | Symbol | Unit | Description | Norm | Reference |
|---|---|---|---|---|---|
| OV1 | $u_{10N}$ | $\mathrm{m\,s^{-1}}$ | Ten meter neutral wind speed derived from the flow distortion corrected in situ measurements | linear | 10.5281/zenodo.3836439 |
| OV2 | $P_{air}$ | hPa | Atmospheric pressure recorded 20 meter above sea level | linear | 10.5281/zenodo.3379590 |
| OV3 | $\mathrm{mask_{cyc}}$ | – | Surface cyclone mask | linear | 10.5281/zenodo.3974312 |
| OV4 | $\mathrm{mask_{CW}}$ | – | Cold and warm temperature advection mask | linear | 10.5281/zenodo.3989318 |
| OV5 | $T_{air}$ | °C | Air temperature measured 23.7 meter above sea level | linear | 10.5281/zenodo.3379590 |
| OV6 | $S_{in}$ | $\mathrm{W\,m^{-2}}$ | Solar radiation | linear | 10.5281/zenodo.3379590 |
| OV7 | PAR | $\mathrm{\mu mol\,photons\,m^{-2}\,s^{-1}}$ | Photosynthetically active radiation (PAR), sky irradiance over PAR wavelengths (400 to 700 nm) | linear | 10.5281/zenodo.3859836 |

**Table A2.** Original variables used in this study, which fall in the category *Atmospheric side of the hydrological cycle*. The columns provide the original variable ID, the symbol, the SI units, a short description of the variable, the input normalisation used for the sPCA analysis, as well as a reference to the published dataset or to the methods section.

| OV-ID | Symbol | Unit | Description | Norm | Reference |
|---|---|---|---|---|---|
| OV8 | RH | % | Relative humidity | linear | 10.5281/zenodo.3379590 |
| OV9 | $w$ | ppmv | Water vapour mixing ratio | log | 10.5281/zenodo.3250790 |
| OV10 | $\delta^{18}O_{vap}$ | ‰ | $\delta^{18}O$ of atmospheric water vapour | linear | 10.5281/zenodo.3250790 |
| OV11 | $\delta^{2}H_{vap}$ | ‰ | $\delta^{2}H$ of atmospheric water vapour | linear | 10.5281/zenodo.3250790 |
| OV12 | $dexc_{vap}$ | ‰ | Deuterium excess of atmospheric water vapour | linear | 10.5281/zenodo.3250790 |
| OV13 | RR | $\mathrm{mm\,h^{-1}}$ | Rainfall rate at 100 to 200 m a.s.l. | log | 10.5281/zenodo.3367284 |
| OV14 | SR | $\mathrm{mm\,h^{-1}}$ | Snowfall rate | log | 10.5281/zenodo.3367284 |
| OV15 | HHF | $\mathrm{m^{2}\,s^{-1}}$ | Horizontal hydrometeor flux | log | 10.5281/zenodo.4446616 |
| OV16 | visibility | m | Horizontal Visibility | log | 10.5281/zenodo.3379590 |
| OV17 | CL | m | Lowest cloud level estimated with a Ceilometer | log | 10.5281/zenodo.3379590 |
| OV18 | SC | octants | Scy cover at the lowest cloud level | linear | 10.5281/zenodo.3379590 |





**Table A3.** Original variables used in this study, which fall in the category *Atmospheric chemistry*. The columns provide the original variable ID, the symbol, the SI units, a short description of the variable, the input normalisation used for the sPCA analysis, as well as a reference to the published dataset or to the methods section.

| OV-ID | Symbol | Unit | Description | Norm | Reference |
|---|---|---|---|---|---|
| OV19 | $CO_2$ | ppm | Dry mixing ratio of carbon dioxide in ambient air | linear | 10.5281/zenodo.4028749 |
| OV20 | $CH_4$ | ppm | Dry mixing ratio of Methane in ambient air | linear | 10.5281/zenodo.4028749 |
| OV21 | CO | ppm | Mixing ratio of carbon monoxide in ambient air | linear | 10.5281/zenodo.4028749 |
| OV22 | $O_3$ | ppb | Mixing ratio of ozone in ambient air | linear | 10.5281/zenodo.2636779 |
| OV23 | $Isoprene_{air}$ | ppb | Mixing ratio of isoprene in ambient air | log | See SI for details |
| OV24 | $N_{Aitken}$ | $cm^{-3}$ | Particle number concentration in the Aitken mode | log | See SI for details |
| OV25 | $N_{accumulation}$ | $cm^{-3}$ | Particle number concentration in the accumulation mode | log | See SI for details |
| OV26 | $N_{seaspray}$ | $cm^{-3}$ | Particle number concentration in the sea spray mode | log | See SI for details |
| OV27 | $N_{oxalate,PM10}$ | $\mu g\,m^{-3}$ | Mass concentration of oxalate in PM10 dry aerosol particles from off-line high-volume filter sampling | log | 10.5281/zenodo.3922147 |
| OV28 | $N_{bromide,PM10}$ | $\mu g\,m^{-3}$ | Mass concentration of bromide in PM10 dry aerosol particles from off-line high-volume filter sampling | log | 10.5281/zenodo.3922147 |
| OV29 | $N_{MSA,PM10}$ | $\mu g\,m^{-3}$ | Mass concentration of MSA in PM10 dry aerosol particles from off-line high-volume filter sampling | log | 10.5281/zenodo.3922147 |
| OV30 | $N_{sodium,PM10}$ | $\mu g\,m^{-3}$ | Mass concentration of sodium in PM10 dry aerosol particles from off-line high-volume filter sampling | log | 10.5281/zenodo.3922147 |
| OV31 | $N_{ammonium,PM10}$ | $\mu g\,m^{-3}$ | Mass concentration of ammonium in PM10 dry aerosol particles from off-line high-volume filter sampling | log | 10.5281/zenodo.3922147 |
| OV32 | $N_{potassium,PM10}$ | $\mu g\,m^{-3}$ | Mass concentration of potassium in PM10 dry aerosol particles from off-line high-volume filter sampling | log | 10.5281/zenodo.3922147 |
| OV33 | $N_{magnesium,PM10}$ | $\mu g\,m^{-3}$ | Mass concentration of magnesium in PM10 dry aerosol particles from off-line high-volume filter sampling | log | 10.5281/zenodo.3922147 |
| OV34 | $N_{calcium,PM10}$ | $\mu g\,m^{-3}$ | Mass concentration of calcium in PM10 dry aerosol particles from off-line high-volume filter sampling | log | 10.5281/zenodo.3922147 |
| OV35 | $N_{nitrate,PM10}$ | $\mu g\,m^{-3}$ | Mass concentration of nitrate in PM10 dry aerosol particles from off-line high-volume filter sampling | log | 10.5281/zenodo.3922147 |
| OV36 | $SO_4^{2-}$ | $\mu g\,m^{-3}$ | Mass concentration of Sulfate in non-refractory particulate matter (PM1) | log | 10.5281/zenodo.3559982 |





**Table A4.** Extension of Table A3

| OV-ID | Symbol | Unit | Description | Norm | Reference |
|---|---|---|---|---|---|
| OV37 | $Cl^-$ | $\mu g\,m^{-3}$ | Mass concentration of Chloride in non-refractory particulate matter (PM1) (incomplete since Chloride is refractory) | log | 10.5281/zenodo.3559982 |
| OV38 | $H_2SO_4$ | $molec\,cm^{-3}$ | Concentration of gaseous sulfuric acid | log | 10.5281/zenodo.3265832 |
| OV39 | $HIO_3$ | $molec\,cm^{-3}$ | Concentration of gaseous iodic acid | log | See SI for details |
| OV40 | MSA | $molec\,cm^{-3}$ | Concentration of gaseous methanesulfonic acid | log | 10.5281/zenodo.2636771 |
| OV41 | $N_{CCN,0.15}$ | $cm^{-3}$ | Particle number concentration acting as CCN at 0.15% supersaturation | log | 10.5281/zenodo.4415495 |
| OV42 | $N_{CCN,0.30}$ | $cm^{-3}$ | Particle number concentration acting as CCN at 0.3% supersaturation | log | 10.5281/zenodo.4415495 |
| OV43 | $N_{CCN,1.00}$ | $cm^{-3}$ | Particle number concentration acting as CCN at 1.0% supersaturation | log | 10.5281/zenodo.4415495 |
| OV44 | $\kappa_{CCN,0.15}$ | – | Hygroscopicity parameter of particles acting as CCN at 0.15% superaturation | log | 10.5281/zenodo.4415495 |
| OV45 | $\kappa_{CCN,0.30}$ | – | Hygroscopicity parameter of particles acting as CCN at 0.5% superaturation | log | 10.5281/zenodo.4415495 |
| OV46 | $\kappa_{CCN,1.00}$ | – | Hygroscopicity parameter of particles acting as CCN at 1.0% superaturation | log | 10.5281/zenodo.4415495 |
| OV47 | $N_{INP,LV,-8}$ | $dm^{-3}$ | INP number concentration at $-8\,°C$ from off-line low-volume PM10 filter sampling | log | 10.5281/zenodo.4311665 |
| OV48 | $N_{INP,LV,-20}$ | $dm^{-3}$ | INP number concentration at $-20\,°C$ from off-line low-volume PM10 filter sampling | log | 10.5281/zenodo.4311665 |
| OV49 | $N_{INP,HV,-8}$ | $dm^{-3}$ | INP number concentration at $-8\,°C$ from off-line high-volume PM10 filter sampling | log | See SI for details |
| OV50 | $N_{INP,HV,-20}$ | $dm^{-3}$ | INP number concentration at $-20\,°C$ from off-line high-volume PM10 filter sampling | log | See SI for details |
| OV51 | $r_{fluo,fine3\sigma}$ | – | Ratio of fluorescent to total aerosol particles (for particles with optical diameter smaller than $1\,\mu m$) | log | See SI for details |
| OV52 | $r_{fluo,coarse3\sigma}$ | – | Ratio of fluorescent to total aerosol particles (for particles with optical diameter greater than $1\,\mu m$) | log | See SI for details |



**Table A5.** Original variables used in this study, which fall in the category *Oceanic dynamics and thermodynamics*. The columns provide the original variable ID, the symbol, the SI units, a short description of the variable, the input normalisation used for the sPCA analysis, as well as a reference to the published dataset or to the methods section.

| OV-ID | Symbol | Unit | Description | Norm | Reference |
|---|---|---|---|---|---|
| OV53 | $H_s$ | m | Significant wave height | log | 10.5281/zenodo.4541564 |
| OV54 | $T_{m-1,1}$ | s | Spectral mean wave energy period | log | 10.5281/zenodo.4541564 |
| OV55 | SSH | m | Sea-surface height (satellite absolute dynamic topography) | linear | 10.5281/zenodo.3660852 |
| OV56 | $U_g$ | $\mathrm{m\,s^{-1}}$ | Surface ocean geostrophic velocity (satellite) | log | 10.5281/zenodo.3660852 |
| OV57 | $T_{sw}$ | $^\circ\mathrm{C}$ | Surface ocean temperature of seawater | linear | 10.5281/zenodo.3660852 |
| OV58 | MLD | m | Surface ocean mixed layer depth | linear | 10.5281/zenodo.3836648 |
| OV59 | $I_g$ | $\mathrm{\mu mol\,photons\,m^{-2}\,s^{-1}}$ | Median light level over 24 hours within the surface ocean mixed layer | log | 10.5281/zenodo.3859836 |
| OV60 | $\sigma_{0,sw}$ | $\mathrm{kg\,m^{-3}}$ | Surface ocean potential density anomaly of seawater | linear | 10.5281/zenodo.3660852 |

**Table A6.** Original variables used in this study, which fall in the category *Oceanic side of the hydrological cycle*. The columns provide the original variable ID, the symbol, the SI units, a short description of the variable, the input normalisation used for the sPCA analysis, as well as a reference to the published dataset or to the methods section.

| OV-ID | Symbol | Unit | Description | Norm | Reference |
|---|---|---|---|---|---|
| OV61 | $S_{sw}$ | PSU | Surface ocean salinity of seawater | linear | 10.5281/zenodo.3660852 |
| OV62 | $C_i$ | – | Sea-ice concentration (satellite) | linear | 10.5281/zenodo.3660852 |
| OV63 | $\delta^{18}O_{sw}$ | ‰ | Surface ocean $\delta^{18}O$ of seawater | linear | 10.5281/zenodo.1494915 |



**Table A7.** Original variables used in this study, which fall in the category *Ocean microbial community*. The columns provide the original variable ID, the symbol, the SI units, a short description of the variable, the input normalisation used for the sPCA analysis, as well as a reference to the published dataset or to the methods section.

| OV-ID | Symbol | Unit | Description | Norm | Reference |
|---|---|---|---|---|---|
| OV64 | TChl $a$ | $\mathrm{mg\,m^{-3}}$ | Total chlorophyll-a concentration derived from high performance liquid chromatography and particulate absorption line height | log | 10.5281/zenodo.3816726 |
| OV65 | Chl $a_\mathrm{fluo}$ | $\mathrm{\mu g\,l^{-1}}$ | In vivo fluorescence of chlorophyll $a$ | log | See SI for details |
| OV66 | Chlide $a$ | $\mathrm{mg\,m^{-3}}$ | Chlorophyllide $a$ pigment concentration | log | 10.5281/zenodo.3816726 |
| OV67 | Phaeob $a$ | $\mathrm{mg\,m^{-3}}$ | Pheophorbide $a$ pigment concentration | log | 10.5281/zenodo.3816726 |
| OV68 | Phaeophy $a$ | $\mathrm{mg\,m^{-3}}$ | Pheophytin-a pigment concentration | log | 10.5281/zenodo.3816726 |
| OV69 | $\mathrm{PSD_{slope}}$ | – | Slope of the particle size distribution 2–60 μm | linear | See SI for details |
| OV70 | $\mathrm{F_V F_M}$ | – | Photochemical efficiency of photosystem II measured during the night | linear | See SI for details |
| OV71 | $\Phi_\mathrm{PSII}'$ | – | Photochemical efficiency of photosystem II during the day | linear | See SI for details |
| OV72 | $\sigma_\mathrm{PSII}$ | $\mathrm{\mathring{A}^2\,RCII^{-1}}$ | Functional absorption cross section of PSII during the night | linear | See SI for details |
| OV73 | $\sigma_\mathrm{PSII}'$ | $\mathrm{\mathring{A}^2\,RCII^{-1}}$ | Functional absorption cross section of PSII during in the day | linear | See SI for details |
| OV74 | Chloro | $\mathrm{mg\,m^{-3}}$ | Chlorophyte contribution to chlorophyll biomass | log | 10.5281/zenodo.3816726 |
| OV75 | Crypto | $\mathrm{mg\,m^{-3}}$ | Cryptophyte contribution to chlorophyll biomass | log | 10.5281/zenodo.3816726 |
| OV76 | Cyano | $\mathrm{mg\,m^{-3}}$ | Cyanobacteria type 2 contribution to chlorophyll biomass | log | 10.5281/zenodo.3816726 |
| OV77 | DiatA | $\mathrm{mg\,m^{-3}}$ | Diatom type 1 contribution to chlorophyll biomass | log | 10.5281/zenodo.3816726 |
| OV78 | DiatB | $\mathrm{mg\,m^{-3}}$ | Diatom type 2 contribution to chlorophyll biomass | log | 10.5281/zenodo.3816726 |
| OV79 | Dino | $\mathrm{mg\,m^{-3}}$ | Dinoflagellate type 1 contribution to chlorophyll biomass | log | 10.5281/zenodo.3816726 |
| OV80 | Hapto8 | $\mathrm{mg\,m^{-3}}$ | Haptophyte type 8 contribution to chlorophyll biomass | log | 10.5281/zenodo.3816726 |
| OV81 | Hapto67 | $\mathrm{mg\,m^{-3}}$ | Haptophyte type 6&7 contribution to chlorophyll biomass | log | 10.5281/zenodo.3816726 |



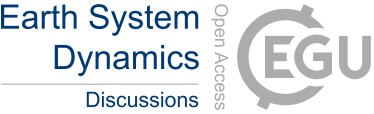

**Table A8.** Extension of Table A7

| OV-ID | Symbol | Unit | Description | Norm | Reference |
|-------|--------|------|-------------|------|-----------|
| OV82 | Prasino | $\mathrm{mg\,m^{-3}}$ | Prasinophyte type 3 contribution to chlorophyll biomass | log | 10.5281/zenodo.3816726 |
| OV83 | Pelago | $\mathrm{mg\,m^{-3}}$ | Pelagophyte contribution to chlorophyll biomass | log | 10.5281/zenodo.3816726 |
| OV84 | $N_{\mathrm{totalbacteria}}$ | $\mathrm{cells\,ml^{-1}}$ | Abundance of phototrophic prokaryotes, (mainly free living bacteria) | log | See SI for details |
| OV85 | $N_{\mathrm{synechococcus}}$ | $\mathrm{cells\,ml^{-1}}$ | Abundance of *Synechococcus* | log | See SI for details |
| OV86 | $N_{\mathrm{nanoeukaryotes}}$ | $\mathrm{cells\,ml^{-1}}$ | Abundance of nano-phytoplankton (roughly $3$–$10\,\mu\mathrm{m}$ size) | log | See SI for details |
| OV87 | $N_{\mathrm{picoeukaryotes}}$ | $\mathrm{cells\,ml^{-1}}$ | Abundance of eukaryotic pico-phytoplankton ($\lesssim 3\,\mu\mathrm{m}$ size) | log | See SI for details |





**Table A9.** Original variables used in this study, which fall in the category *Ocean biogeochemistry*. The columns provide the original variable ID, the symbol, the SI units, a short description of the variable, the input normalisation used for the sPCA analysis, as well as a reference to the published dataset or to the methods section.

| OV-ID | Symbol | Unit | Description | Norm | Reference |
|---|---|---|---|---|---|
| OV88 | Nitrate | µM | Dissolved nitrate ($NO_3^-$) concentration | log | 10.5281/zenodo.3903134 |
| OV89 | Nitrite | µM | Dissolved nitrite ($NO_2^-$) concentration | log | 10.5281/zenodo.3903134 |
| OV90 | Phosphate | µM | Dissolved phosphate ($PO_4^{3-}$) concentration | log | 10.5281/zenodo.3903134 |
| OV91 | Silicate | µM | Dissolved silicate ($Si(OH)_4$) concentration | log | 10.5281/zenodo.3903134 |
| OV92 | Ammonium | µM | Dissolved ammonium ($NH_4^+$) concentration | log | 10.5281/zenodo.3751143 |
| OV93 | POC | µM | Particulate organic carbon concentration | log | 10.5281/zenodo.3859515 |
| OV94 | PON | µM | Particulate organic nitrogen concentration | log | 10.5281/zenodo.3859515 |
| OV95 | C:N | – | Particulate organic carbon to nitrogen ratio | log | 10.5281/zenodo.3859515 |
| OV96 | TEP | $\mu g\,XGeq\,L^{-1}$ | Concentration of transparent exopolymeric particles | log | See SI for details |
| OV97 | CSP | $\mu g\,BSAeq\,L^{-1}$ | Concentration of Coomassie stainable particles | log | See SI for details |
| OV98 | $a_{CDOM}$ | $m^{-1}$ | Absorption due to coloured dissolved organic matter at 350 nm | log | See SI for details |
| OV99 | $a_{napslope}$ | $nm^{-1}$ | Spectral slope of detrital absorption 380 to 700 nm (Reference wavelength 380 nm) | linear | See SI for details |
| OV100 | $a_{nap}/a_p$ | – | Ratio of detrital absorption relative to total particulate absorption at 440 nm | linear | See SI for details |
| OV101 | acrylate | nM | Concentration of acrylate | log | See SI for details |
| OV102 | DMSP | nM | Concentration of total (dissolved + particulate) dimethylsulfoniopropionate | log | See SI for details |
| OV103 | DMS | nM | Concentration of dimethyl sulfide | log | See SI for details |
| OV104 | CSO | pM | Concentration of carbonyl sulfide | log | See SI for details |
| OV105 | $CS_2$ | pM | Concentration of carbon disulfide | log | See SI for details |

**Table A10.** Extension of Table A9

| OV-ID | Symbol | Unit | Description | Norm | Reference |
|---|---|---|---|---|---|
| OV106 | Isoprene$_{sea}$ | pM | Concentration of isoprene ($C_5H_8$) | log | See SI for details |
| OV107 | $CHBr_3$ | pM | Concentration of tribromomethane (bromoform) | log | See SI for details |
| OV108 | $CH_2Br_2$ | pM | Concentration of dibromomethane | log | See SI for details |
| OV109 | NCP | $Mmol\,O_2\,m^{-2}\,day^{-1}$ | Net community production ($O_2$/Ar based) | linear | 10.5281/zenodo.3979091 |



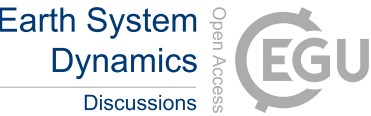

**Table A11.** Original variables used in this study, which fall in the category *Topography*. The columns provide the original variable ID, the symbol, the SI units, a short description of the variable, the input normalisation used for the sPCA analysis, as well as a reference to the published dataset or to the methods section.

| OV-ID | Symbol | Unit | Description | Norm | Reference |
|-------|--------|------|-------------|------|-----------|
| OV110 | $d_{\mathrm{land}}$ | m | Distance to the nearest shore line in meter | log | 10.5281/zenodo.3832045 |
| OV111 | $d_{\mathrm{water}}$ | m | Water depth calculated from the General Bathymetric Chart of the Oceans | log | 10.5281/zenodo.3773102 |





## Appendix B: Correlation between the latent variable time series


Figure B1a shows the correlation matrix between the LV times series. Figure B1b shows the p-value for each correlation for pairs of LV. Shades of blue highlight LV pairs showing a level of correlation significant with a p-value < 0.05. The darker the blue, the more significant. Conversely, yellow to red cells correspond to non-significant correlations, the darker the red, the less significant. The permutation test is computed by taking pairs of LV, and computing a correlation value. Then, the first LV

is shuffled randomly, and a correlation value computed, for 10000 times. The p-value is the fraction of times that the random correlation is larger than the correlation between actual, non-shuffled, LVs. Note the high correlation between LV1 and LV14 (R=0.74) with a p-value of 0.0002.

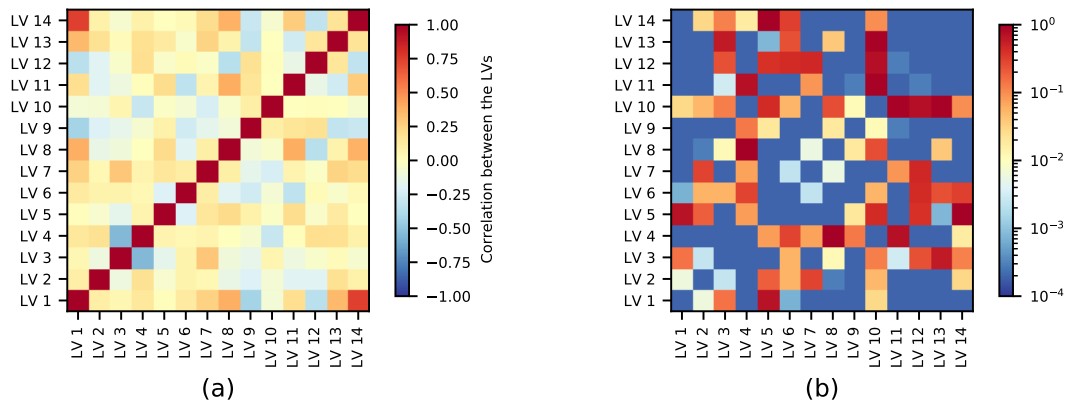

**Figure B1.** (a) Correlation between the LV time series and (b) the corresponding p-value of a two-sample permutation test with 10000 permutations.

## Appendix C: Low value replacement

Zero value observations would be lost when a log-transformation is applied to the log-normal distributed OV time series.

In addition the log-transformation puts larger weight on small variations, which may not necessarily reflect real variability if they are below the instruments detection limit. For this reason we choose to replace low values in some of the OV series prior to applying the log-transformation. Table C1 lists the OVs for which this was the case and provides the values that were chosen for the low value replacement as well as the fraction of replaced data. The low value replacements were chosen based on the instrument's limit of detection or based on the judgement of the data experts. Large fractions of data are replaced

for the precipitation time series RR and SR due to the large fraction of observations without precipitation. Also a relatively large fraction of the Ice nucleation particle number concentration measurements ($N_{\mathrm{INP}}$) and some of the contributions to the chlorophyll biomass derived by the CHEMTAX analysis were below the respective detection limits. For the remaining OVs in table C1, only a small fraction of datapoints (i.e. < 5%) in each timeseries were replaced with the low value replacement value.



**Table C1.** Original variables used in this study, to which a low value replacement was applied. The columns provide the original variable ID, the symbol, the SI units, the low value replacement, and the fraction of replaced data.

| OV-ID | Symbol | Unit | Low value replacement | Data fraction replaced |
|---|---|---|---|---|
| OV13 | RR | $\mathrm{mm\,h^{-1}}$ | 1.00E-02 | 0.639 |
| OV14 | SR | $\mathrm{mm\,h^{-1}}$ | 1.00E-02 | 0.876 |
| OV15 | HHF | $\mathrm{m^2\,s^{-1}}$ | 3.50E+01 | 0.056 |
| OV36 | $SO_4^{2-}$ | $\mathrm{\mu g\,m^{-3}}$ | 1.40E-01 | 0.015 |
| OV37 | $Cl^-$ | $\mathrm{\mu g\,m^{-3}}$ | 3.20E-02 | 0.004 |
| OV47 | $N_{\mathrm{INP,LV},-8}$ | $\mathrm{dm^{-3}}$ | 8.00E-04 | 0.882 |
| OV48 | $N_{\mathrm{INP,LV},-20}$ | $\mathrm{dm^{-3}}$ | 1.15E-01 | 0.599 |
| OV49 | $N_{\mathrm{INP,HV},-8}$ | $\mathrm{dm^{-3}}$ | 1.89E-04 | 0.403 |
| OV50 | $N_{\mathrm{INP,HV},-20}$ | $\mathrm{dm^{-3}}$ | 2.76E-02 | 0.299 |
| OV53 | $H_s$ | m | 3.00E-01 | 0.007 |
| OV56 | $U_{\mathrm{g}}$ | $\mathrm{m\,s^{-1}}$ | 1.00E-02 | 0.018 |
| OV74 | Chloro | $\mathrm{mg\,m^{-3}}$ | 1.00E-05 | 0.412 |
| OV75 | Crypto | $\mathrm{mg\,m^{-3}}$ | 1.00E-05 | 0.264 |
| OV76 | Cyano | $\mathrm{mg\,m^{-3}}$ | 1.00E-05 | 0.389 |
| OV77 | DiatA | $\mathrm{mg\,m^{-3}}$ | 1.00E-05 | 0.097 |
| OV78 | DiatB | $\mathrm{mg\,m^{-3}}$ | 1.00E-05 | 0.125 |
| OV79 | Dino | $\mathrm{mg\,m^{-3}}$ | 1.00E-05 | 0.060 |
| OV80 | Hapto8 | $\mathrm{mg\,m^{-3}}$ | 1.00E-05 | 0.065 |
| OV82 | Prasino | $\mathrm{mg\,m^{-3}}$ | 1.00E-05 | 0.352 |
| OV83 | Pelago | $\mathrm{mg\,m^{-3}}$ | 1.00E-05 | 0.093 |
| OV92 | Ammonium | μM | 2.00E-02 | 0.006 |
| OV96 | TEP | $\mathrm{\mu g\,XGeq\,L^{-1}}$ | 5.00E+00 | 0.024 |
| OV104 | CSO | pM | 1.00E+00 | 0.016 |
| OV110 | $d_{\mathrm{land}}$ | m | 1.00E+02 | 0.005 |
| OV111 | $d_{\mathrm{water}}$ | m | 2.00E+01 | 0.040 |





## Appendix D: Sparse PCA results for different time resolutions

The sPCA analysis requires all OVs to be resampled to the same temporal resolution. Choosing a too high resolution will result in large differences in the sparsity of the OV time series and OVs, which are sampled at low resolution will likely not contribute to the sPCA solution. Conversely, resampling all variables to a low temporal resolution will result in a loss of information about small scale variability. The optimal choice of the averaging time depends on the dataset and on the research questions. Figure D1 shows the correlation between the weight vectors obtained from sparse PCA decompositions of the 180

and 720 minutes time resolutions. Figure D2 shows the corresponding correlation of the time series. Here we label LVs from the 720 minute time resolution run with lower case letters (i.e. lv1 instead of LV1).

     For both time resolutions, the climatic zones and large-scale horizontal gradients (LV1) provide the largest signal. The Drivers of CCN population signal (LV2) explains relatively less variance in the 720 min where it corresponds to lv7. The signals of Meridional cold and warm air advection (LV3) and Precipitation (LV4), which both showed relatively large correlation in

their time series both correlate with lv4. The signals of the ship's distance to land (LV5), the iron-fertilized and iron-limited biological productivity (LV6 and LV8), the seasonal signal (LV7), the diel cycle (LV10), the surface nutrient concentration associated with mixing events, climatic and frontal zones (LV11) as well as the wind-driven conditions and sea spray aerosol (LV12) find very similar resemblance in the sPCA solution found for the 720 min resolution. The LVs describing the marginal sea ice zone and snowfall (LV9), the extratropical cyclone activity (LV13), and the climatic zones with local high-latitude

hotspots (LV14) on the other hand appear to be somewhat redistributed in the 720 min sPCA, as can be seen by the weaker correlation to more than one of the lvs.



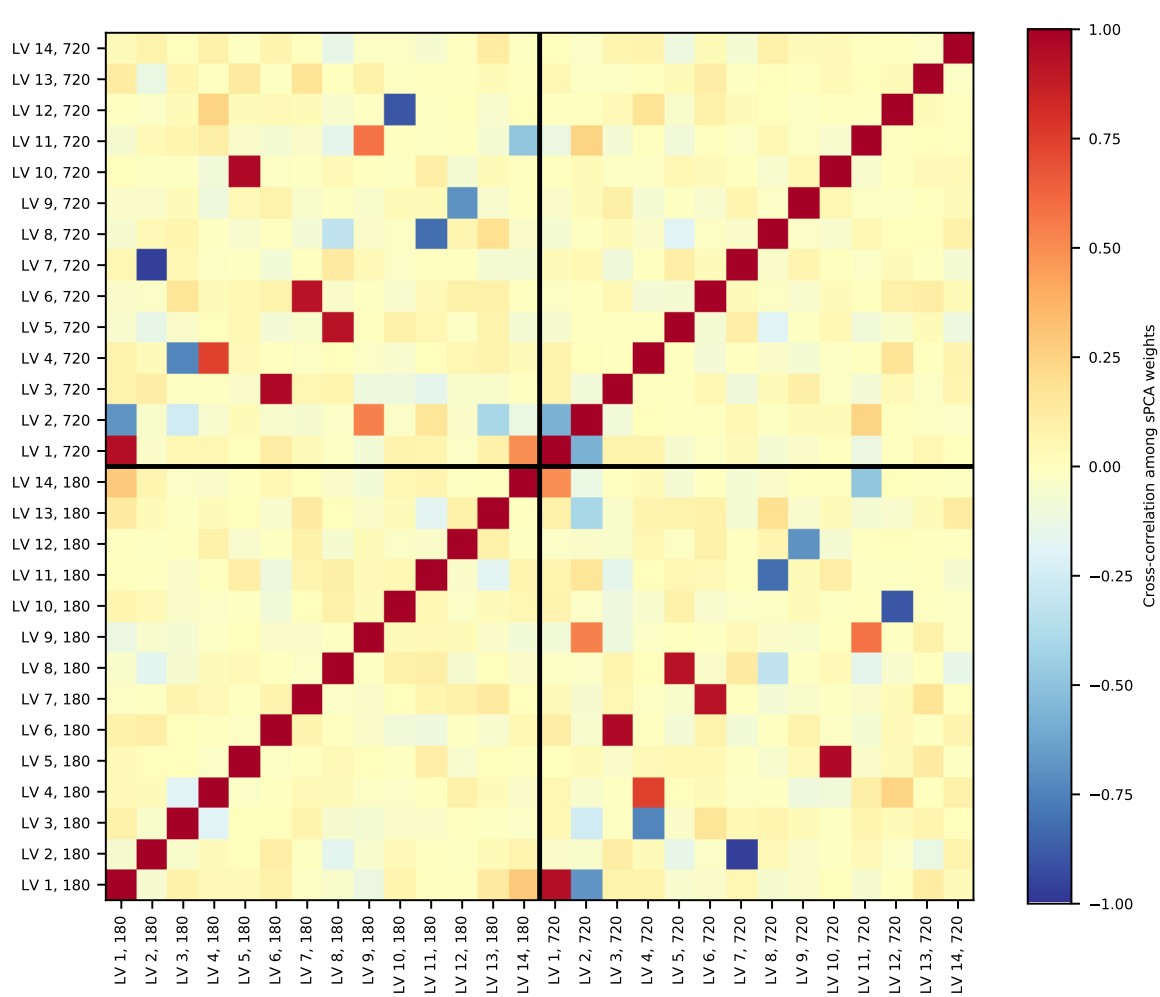

**Figure D1.** Correlation between the weight vectors obtained from sPCA decompositions of the 180 and 720 minute time resolutions.



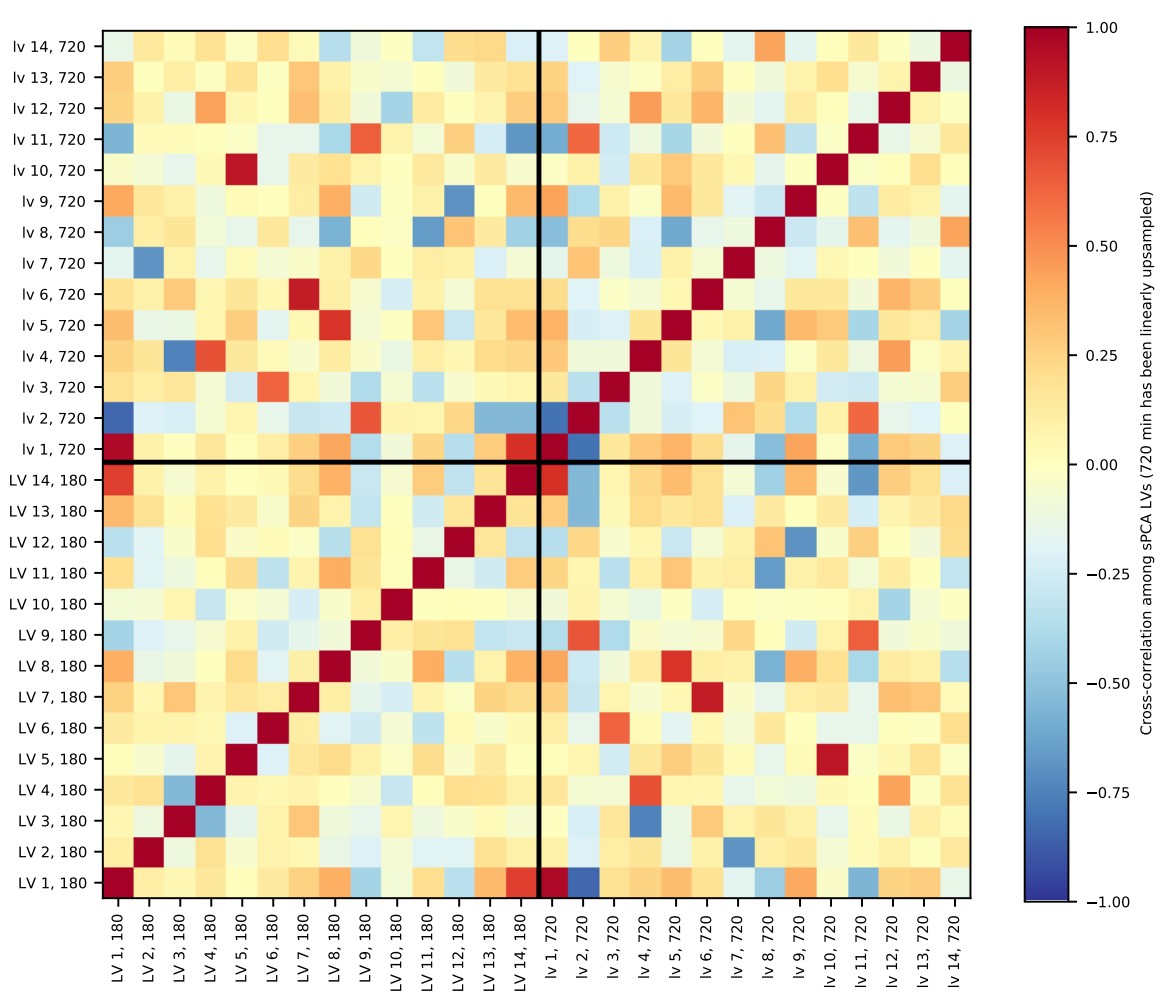

**Figure D2.** Correlation between the LV time series obtained from sPCA decompositions of the 180 and 720 minute time resolutions.



**Appendix E: Variable glossary**



**OV1 : Ten metre neutral wind speed** ($u_{10\mathrm{N}}$)

Ten metre neutral wind speed is a standardised form to report the wind speed that would be measured under the given wind forcing at ten metre height and neutral atmospheric stability and provides a proxy for the wind forcing at the ocean surface and can be directly related to the surface friction velocity via the surface drag coefficient. The ten metre neutral wind speed is calculated from in situ wind speed measurements at a given measurement height under the assumption of a near-logarithmic height-profile using bulk-flux formula and an empirical relation for the wind speed dependence of the neutral drag coefficient.

**OV2 : Atmospheric pressure** ($P_{\mathrm{air}}$)

Atmospheric pressure was measured at a height of approximately 20 m above sea level.

**OV3 : Surface cyclone mask** ($\mathrm{mask}_{\mathrm{cyc}}$)

The surface cyclone mask denotes time periods when a surface cyclone passes the ACE ship track. More specifically, if the ship track lies within a surface cyclone which is defined by the outermost closed sea level pressure contour around a pressure minimum (see supplementary information S1.2), the cyclone mask is active (1 in mask). During the passage of an extratropical cyclone, enhanced wind speed and high variability in surface pressure, air temperature and specific humidity as well as precipitation is expected. The actual environmental changes due to the passage of an extratropical cyclone depends on the measurement position relative to the cyclone center and the cyclone's fronts.

**OV4 : Cold and warm temperature advection mask** ($\mathrm{mask}_{\mathrm{CW}}$)

The cold and warm temperature advection mask denotes time periods of differences between the near-surface air temperature and the sea surface temperature. An air-sea temperature difference larger than $0\,°\mathrm{C}$ (i.e. the atmosphere is warmer than the ocean surface) is defined as warm temperature advection (1 in mask), a difference smaller than $0\,°\mathrm{C}$ as cold temperature advection (-1 in mask). The cold and warm temperature advection mask is an indicator of surface sensible and latent heat fluxes and is strongly influenced by the passage of extratropical cyclones, which lead to the large-scale advection of air masses. Furthermore, strong horizontal gradients in sea surface temperature, for example across oceanic fronts, lead to locally large air-sea temperature differences.

**OV5 : Air temperature** ($T_{\mathrm{air}}$)

Air temperature was measured at approximately 24 m above sea level using a Vaisala MAWS420 meteorological station (sensor model QMH102).

**OV6 : Solar radiation** ($S_{\mathrm{in}}$)

Solar radiation was measured at approximately 24 m above sea level.



**OV7 : Photosynthetically active radiation** (PAR)

Photosynthetically active radiation, or sky irradiance over photosynthetically active wavelengths (400 to 700 nm; $\mu\mathrm{mol\,photons\,m^2\,s^{-1}}$) is an instantaneous measurement of the visible irradiance from the sun made above the surface of the ocean. It provides an estimate of the flux of visible light delivered to the surface ocean which is utilised by photosynthetic organisms, namely phytoplankton, as an energy source for their photosynthesis. Excess light in the visible range can also be harmful to photosynthetic organisms, inhibiting their photosynthesis for hours after exposure and has been linked to the production of secondary metabolites and trace gasses by these organisms.

**OV8 : Relative humidity** (RH)

Relative humidity was measured at approximately 24 m above sea level using a Vaisala MAWS240 meteorological station (sensor model QMH102) and represents the humidity saturation of the atmosphere relative to the saturation vapour pressure. The relative humidity depends on the air temperature and the atmospheric pressure.

**OV9 : Water vapour mixing ratio** ($w$)

The water vapour mixing ratio $w$ is the water vapour volume relative to the dry air volume given in part per million volume (ppmv) and is a measure of humidity in the air. $w$ was measured using a cavity ring-down laser spectrometer and was calibrated with a dew point generator.

**OV10 : $\delta^{18}$O of atmospheric water vapour** ($\delta^{18}\mathrm{O_{vap}}$)

$\delta^{18}\mathrm{O_{vap}}$ gives the abundance of the heavy water molecule $\mathrm{H_2^{18}O}$ relative to the light water molecule $\mathrm{H_2^{16}O}$ in atmospheric water vapour and can be used as a tracer of phase change processes in the atmosphere. Various processes, e.g. ocean evaporation, the large-scale advection of air masses or vertical mixing of air masses influence the temporal and spatial variability of $\delta^{18}\mathrm{O_{vap}}$.

**OV11 : $\delta^2$H of atmospheric water vapour** ($\delta^2\mathrm{H_{vap}}$)

$\delta^2\mathrm{H_{vap}}$ gives the abundance of the heavy water molecule $\mathrm{^1H^2H^{16}O}$ relative to the light water molecule $\mathrm{^1H_2^{16}O}$ in atmospheric water vapour and can be used as a tracer of phase change processes in the atmosphere. Various processes, e.g. ocean evaporation, the large-scale advection of air masses or vertical mixing of air masses influence the temporal and spatial variability of $\delta^2\mathrm{H_{vap}}$.

**OV12 : Deuterium excess of atmospheric water vapour** ($dexc_{\mathrm{vap}}$)

The second-order isotope variable deuterium excess is defined as $d_{\mathrm{vap}} = \delta^2\mathrm{H_{vap}} - 8 \cdot \delta^{18}\mathrm{O_{vap}}$ and is often used as a measure of non-equilibrium processes in the atmospheric branch of the water cycle. For example, during ocean evaporation, $d_{\mathrm{vap}}$ increases in the atmospheric water vapour due to the diffusion of water vapour away from the ocean surface.



**OV13 : Rainfall rate** (RR)

The rainfall rate is derived from the 100 to 200 m a.s.l. bin of measurements using a micro-rain radar (MRR2, manufactured by Metek) using Metek software.

**OV14 : Snowfall rate** (SR)

The snowfall rate (SR) is estimated from radar effective reflectivity (Z) at 400 m a.s.l height derived from a vertically profiling Doppler radar using the algorithm by Maahn and Kollias (2012) for snowfall. Z-S relationship from Grazioli et al. (2017) was used. Influence of microphysical snow properties was considered less important for sPCA analysis compared to the variability due to snowfall rate.


**OV15 : Horizontal hydrometeor flux** (HHF)

Horizontal hydrometeor flux (HHF, $\text{m}^{-2}\,\text{s}^{-1}$) is the total flux of particles with sizes between 0.036 to 2 mm derived from Snow Particle Counter (SPC; manufactured by Niigata) measurements and wind speed measurements (starboard anemometer). HHF includes 1) snowfall and blowing snow during snowfall, 2) drifting/blowing snow lifted from the ground (snow on sea ice, 1300 snow from the ice sheet when the ship was in vicinity), 3) rain drops, 4) possibly sea spray.

**OV16 : Horizontal visibility** (visibility)

Horizontal visibility is defined as the "maximum distance at which an observer can see and identify an object lying close to the horizontal plane on which he or she is standing" (AMS glossary). During ACE, horizontal visibility (or meteorological 1305 optical range, from 10 m to 20 km) is derived from Vaisala's MAWS Present Weather Detector (PWD22) measurements of the extinction in forward scattering by small particles suspended in the air (fog droplets, haze, smoke) and larger particles (rain drops, drizzle, snowflakes, ice grains, etc).

**OV17 : Lowest cloud level** (CL)

The height of the lowest cloud level derived from ceilometer (CL31, pulsed diode laser LIDAR) attenuated backscatter profile. Vaisala's CL31 provides information about up to 3 cloud layers and here the lowest is taken.

**OV18 : Sky cover at the lowest cloud level** (SC)

Sky cover at the lowest cloud level is cloud amount of the lowest cloud layers at the corresponding layer height derived from 1315 the time series of ceilometer data by Vaisala CL31 Sky condition algorithm.

**OV19 : Dry air mixing ratio of carbon dioxide** ($CO_2$)

Carbon dioxide ($CO_2$) dry mixing ratio in ambient air, reported in parts per million. $CO_2$ is a long-lived trace gas which disperses globally and is mainly produced by fossil fuel combustion. The Southern Ocean in summer is thought to be a net $CO_2$





sink. Data were cleaned from the influence of pollution using the same methodology applied to aerosol measurements.

**OV20 : Dry air mixing ratio of methane** ($CH_4$)

Methane ($CH_4$) dry mixing ratio in ambient air, reported in parts per million. $CH_4$ is a long-lived trace gas which disperses globally and can be produced by a variety of different sources (e.g. wetlands, agriculture, biomass burning). The Southern

Ocean is thought to be a net $CH_4$ sink. Data were cleaned from the influence of pollution using the same methodology applied to aerosol measurements.

**OV21 : Air mixing ratio of carbon monoxide** (CO)

Carbon monoxide (CO) mixing ratio in ambient air, reported in parts per billion. CO has a lifetime of few weeks to months and

is produced by combustion processes and oxidation of other hydrocarbons (e.g. methane). The main sink for CO is reaction with the hydroxyl radical. Data were cleaned from the influence of pollution using the same methodology applied to aerosol measurements.

**OV22 : Dry air mixing ratio of ozone** ($O_3$)

Ozone ($O_3$) mixing ratio in ambient air, reported in parts per billion. Ozone is a reactive oxidant gas, it is photochemically produced from the oxidation of CO, $CH_4$ and other volatile organic compounds in the presence of $NO_x$. The remote marine boundary layer is a net sink for ozone, which is removed by photolysis and dry deposition to the ocean surface. Data were cleaned from the influence of pollution using the same methodology applied to aerosol measurements.

**OV23 : Isoprene** ($Isoprene_{air}$)

Molar fraction of isoprene, or 2-methyl-1,3-butadiene ($C_5H_8$) in ambient air, reported in nmol of isoprene per mole of air (ppb).

**OV24 : Number concentrations of Aitken mode particles** ($N_{Aitken}$)

The number concentration of aerosol particles in the Aitken size distribution mode ($N_{Aitken}$) was derived from aerosol number

size distributions measured with a scanning mobility particle sizer (SMPS) and Aerodynamic Particle Sizer (APS) using the procedure following Modini et al. (2015). Co-located, 15-minute median size distributions measured by the SMPS and APS were first joined together by interpolating the measurements to a common diameter scale represented in terms of geometric particle diameter. Electrical mobility diameters measured by the SMPS were simply assumed to represent geometric particle diameters (i.e., shape effects were neglected), and aerodynamic diameters measured by the APS were converted to geometric

particle diameters by dividing by the square root of particle effective density. A constant particle effective density of $1.9\,\mathrm{g\,cm^{-3}}$ was assumed for all distributions. Following this, a sequential lognormal mode-fitting procedure was applied to the joined size distributions. A 'sea spray' size distribution was first fitted to the upper portions of the joined size distributions (as described below for $N_{seaspray}$). Residual size distributions were then obtained by subtracting the fitted sea spray modes from the joined size distributions. Bi-lognormal functions were then fitted to the residual size distributions to represent the combined Aitken and





accumulation size distribution modes. A total of 84% of all joined size distributions were successfully fit with this procedure and retained for further analysis, where success was defined as having a mean absolute error (between the residual and fitted distributions) divided by total integrated particle concentration of less than 15%. The fitted Aitken modes were constrained to have modal diameters between 10 and 100 nm and geometric standard deviations between 1.2 and 1.8. $N_{\mathrm{Aitken}}$ represents the integrated number concentrations in the fitted Aiken modes.

**OV25 : Number concentrations of Accumulation mode particles** ($N_{\mathrm{accumulation}}$)

The number concentration of aerosol particles in the accumulation size distribution mode ($N_{\mathrm{Accumulation}}$) was derived from aerosol number size distributions measured with a scanning mobility particle sizer (SMPS) and Aerodynamic Particle Sizer (APS) using the procedure following Modini et al. (2015). Co-located, 15-minute median size distributions measured by the SMPS and APS were first joined together by interpolating the measurements to a common diameter scale represented in terms of geometric particle diameter, before a sequential lognormal mode-fitting procedure was applied to the joined size distributions, as described above for $N_{\mathrm{Aitken}}$. The fitted accumulation modes were constrained to have modal diameters between 90 and 200 nm and geometric standard deviations between 1.2 and 1.8. $N_{\mathrm{Accumulation}}$ represents the integrated number concentrations in the fitted accumulation modes.

**OV26 : Number concentrations of sea spray mode particles** ($N_{\mathrm{seaspray}}$)

The number concentration of aerosol particles in the 'sea spray' size distribution mode ($N_{\mathrm{seaspray}}$) was derived from aerosol number size distributions measured with a scanning mobility particle sizer (SMPS) and Aerodynamic Particle Sizer (APS) using the procedure following Modini et al. (2015). Co-located, 15-minute median size distributions measured by the SMPS and APS were first joined together by interpolating the measurements to a common diameter scale represented in terms of geometric particle diameter, before a sequential lognormal mode-fitting procedure was applied to the joined size distributions, as described above for $N_{\mathrm{Aitken}}$. The fitted sea spray lognormal modes were constrained to have modal diameters within 20% of 180 nm and geometric standard deviations between 2 and 2.5, consistent with previous measurements of sea spray aerosol size distributions above breaking waves (Prather et al., 2013). $N_{\mathrm{seaspray}}$ represents the integrated number concentrations in the fitted sea spray modes. We estimate that the uncertainty in $N_{\mathrm{seaspray}}$ is $\pm 50\%$ based on a sensitivity analysis to the choice of constrained mode diameter over the range from 140 to 300 nm.

**OV27 : Aerosol mass concentration of oxalate** ($N_{\mathrm{oxalate,PM10}}$)

The mass concentration of oxalate in PM10 dry aerosol particles ($N_{\mathrm{oxalate,PM10}}$) was derived from off-line high-volume sampled filters using ion chromatography. The formula for the oxalate di-anion is $C_2O_4^{2-}$.

**OV28 : Aerosol mass concentration of bromide** ($N_{\mathrm{bromide,PM10}}$)

The mass concentration of bromide in PM10 dry aerosol particles ($N_{\mathrm{bromide,PM10}}$) was derived from off-line high-volume





sampled filters using ion chromatography.


**OV29 : Aerosol mass concentration of methanesulfonic acid** ($N_{\mathrm{MSA,PM10}}$)

The mass concentration of methanesulfonic acid (MSA) in PM10 dry aerosol particles ($N_{\mathrm{MSA,PM10}}$) was derived from off-line high-volume sampled filters using ion chromatography. The formula for MSA is $CH_3SO_3H$.

**OV30 : Aerosol mass concentration of sodium** ($N_{\mathrm{sodium,PM10}}$)

The mass concentration of sodium in PM10 dry aerosol particles ($N_{\mathrm{sodium,PM10}}$) was derived from off-line high-volume sampled filters using ion chromatography.

**OV31 : Aerosol mass concentration of ammonium** ($N_{\mathrm{ammonium,PM10}}$)

The mass concentration of ammonium in PM10 dry aerosol particles ($N_{\mathrm{ammonium,PM10}}$) was derived from off-line high-volume sampled filters using ion chromatography.

**OV32 : Aerosol mass concentration of potassium** ($N_{\mathrm{potassium,PM10}}$)

The mass concentration of potassium in PM10 dry aerosol particles ($N_{\mathrm{potassium,PM10}}$) was derived from off-line high-volume

sampled filters using ion chromatography.

**OV33 : Aerosol mass concentration of magnesium** ($N_{\mathrm{magnesium,PM10}}$)

The mass concentration of magnesium in PM10 dry aerosol particles ($N_{\mathrm{magnesium,PM10}}$) was derived from off-line high-volume sampled filters using ion chromatography.


**OV34 : Aerosol mass concentration of calcium** ($N_{\mathrm{calcium,PM10}}$)

Mass concentration of calcium in PM10 dry aerosol particles from off-line high-volume filter sampling. The mass concentration of calcium in PM10 dry aerosol particles ($N_{\mathrm{calcium,PM10}}$) was derived from off-line high-volume sampled filters using ion chromatography.


**OV35 : Aerosol mass concentration of nitrate** ($N_{\mathrm{nitrate,PM10}}$)

The mass concentration of nitrate in PM10 dry aerosol particles ($N_{\mathrm{nitrate,PM10}}$) was derived from off-line high-volume sampled filters using ion chromatography.

**OV36 : Sulfate** ($SO_4^{2-}$)

Sulfate ($SO_4^{2-}$) is solid or liquid particles including sulfuric acid with a size of few micrometers. In this marine environment, it potentially originates from sea salt, ship exhaust and natural marine emissions of dimethylsulfide.




**OV37 : Chloride** ($Cl^-$)

Chloride ($Cl^-$) is a clear reflection of the sea salt contribution to the aerosol concentrations. However, since the chloride signals were obtained via ToF-ACSM, only non-refractory part of Chloride is included, and the signal needs to be considered as a qualitative measurement.

**OV38 : Gaseous sulfuric acid** ($H_2SO_4$)

Atmospheric concentration of gaseous sulfuric acid ($H_2SO_4$) measured with a nitrate chemical ionization mass spectrometer. In the atmosphere, sulfuric acid is produced by the oxidation of sulfur dioxide, which can have both a natural or anthropogenic origin (e.g. phytoplankton emission or fossil fuel combustion). Data were cleaned from the influence of the ship exhaust, using the same methodology applied to all other aerosol measurements.

**OV39 : Gaseous Iodic acid** ($HIO_3$)

Atmospheric concentration of gaseous iodic acid ($HIO_3$) measured with a nitrate chemical ionization mass spectrometer. In the atmosphere, iodic acid acid is produced by the oxidation of the iodine radical, which can come from a variety of different iodine precursors (e.g. molecular iodine and diiodomethane). In the Southern Ocean the only known sources of iodine are natural, like phytoplankton, sea ice and volcanoes. The ship exhaust was not found to influence the iodic acid concentration,

therefore the pollution mask was not applied to the data.

**OV40 : Gaseous methanesulfonic acid** (MSA)

Atmospheric concentration of gaseous methanesulfonic acid ($CH_3SO_3H$) measured with a nitrate chemical ionization mass spectrometer. In the atmosphere, MSA is produced by the oxidation of dimethylsulfide, which can happen both in the gas and

in the acqueous phase. The ship exhaust was not found to influence the MSA concentration, therefore the pollution mask was not applied to the data.

**OV41 : Particle number concentration acting as CCN at 0.15% supersaturation** ($N_{CCN,0.15}$)

The number concentration particles acting as CCN at $0.15\%$ ($N_mathrmCCN, 0.15$) was derived from in situ measurements

using a CCN counter. In the instrument, the flow of sampled particles is directed through a column where a supersaturation of $0.15\%$ (equal to a relative humidity of $100.15\%$) is upheld for 20 minutes during each measurement cycle. All particles that activate at set supersaturation are counted at the column exit by an optical particle counter. Assuming particle activation being foremost size-dependent Dusek et al. (2006), $N_{CCN,0.15}$ hints on abundance of the particles on the larger end of the sampled particle size range.


**OV42 : Particle number concentration acting as CCN at 0.3% supersaturation** ($N_{CCN,0.30}$)

The number concentration of particles acting as CCN at $0.3\%$ supersaturation ($N_{CCN,0.3}$) was derived from in situ measurements using a CCN counter. In the instrument, the flow of sampled particles is directed through a column where a supersatu-





ration of $0.3\%$ (equal to a relative humidity of $100.3\%$) is upheld for 10 minutes during each measurement cycle. All particles
that activate at set supersaturation are counted at column exit by an optical particle counter. $N_{\mathrm{CCN},0.3}$ is a quantity relevant for
aerosol-cloud interaction studies, as a supersaturation of $0.3\%$ is a typical value for in-cloud conditions.

**OV43 : Particle number concentration acting as CCN at 1.0% supersaturation** $(N_{\mathrm{CCN},1.00})$

The number concentration of particles acting as CCN at $1.0\%$ supersaturation $(N_{\mathrm{CCN},1.0})$ was derived from in situ measure-
ments using a CCN counter. In the instrument, the flow of sampled particles is directed through a column where a supersatu-
ration of $1.0\%$ (equal to a relative humidity of $101\%$) is upheld for 10 minutes during each measurement cycle. All particles
that activate at set supersaturation are counted at column exit by an optical particle counter. Under the assumption that particle
activation is foremost size-dependent Dusek et al. (2006), $N_{\mathrm{CCN},1.0}$ hints towards the abundance of the smaller end of the
sampled particle size range, when compared with $N_{\mathrm{CCN}}$ at a lower supersaturation.


**OV44 : Hygroscopicity parameter of particles acting as CCN at 0.15% supersaturation** $(\kappa_{\mathrm{CCN},0.15})$

The hygroscopicity parameter for particles acting as CCN at $0.15\%$ supersaturation $(\kappa_{0.15})$ attempts to characterise the chemistry-
dependent hygroscopicity of aerosol particles with a single parameter. $\kappa$ was first proposed in Petters and Kreidenweis (2007)
and is derived by integration of the PNSD along decreasing particle diameter, up until the diameter where the integral equals
respective $N_{\mathrm{CCN}}$, the critical diameter $(D_{\mathrm{crit}})$. Typical values are $\kappa = 0.0$ for insoluble material, $\kappa = 0.1$ for water soluble
organics, $\kappa = 0.61$ for ammonium sulfate, and $\kappa = 1.3$ for sodium chloride. Here, under the assumption that particle activation
is foremost size-dependent Dusek et al. (2006), $\kappa_{0.15}$ provides information on the chemical composition of the larger particles
of our sampled size range.

**OV45 : Hygroscopicity parameter of particles acting as CCN at 0.3% supersaturation** $(\kappa_{\mathrm{CCN},0.30})$

The hygroscopicity parameter for particles acting as CCN at $0.3\%$ supersaturation $(\kappa_{0.3})$ tries to describe the chemistry-
dependent hygroscopicity of aerosol particles with a single parameter. $\kappa$ was first proposed in Petters and Kreidenweis (2007)
and is derived by integration of the PNSD along decreasing particle diameter, up until the diameter where the integral equals
respective $N_{\mathrm{CCN}}$, the critical diameter $(D_{\mathrm{crit}})$. Typical values are $\kappa = 0.0$ for insoluble material, $\kappa = 0.1$ for water soluble
organics, $\kappa = 0.61$ for ammonium sulfate, and $\kappa = 1.3$ for sodium chloride. Here, $\kappa_{0.3}$ provides information on the chemical
composition of particles relevant for in-cloud processes, as a supersaturation of $0.3\%$ is a typical value for in-cloud conditions.

**OV46 : Hygroscopicity parameter of particles acting as CCN at 1.0% supersaturation** $(\kappa_{\mathrm{CCN},1.00})$

The hygroscopicity parameter for particles acting as CCN at $1.0\%$ supersaturation $(\kappa_{1.0})$ tries to describe the chemistry-
dependent hygroscopicity of aerosol particles with a single parameter. $\kappa$ was first proposed in Petters and Kreidenweis (2007)
and is derived by integration of the PNSD along decreasing particle diameter, up until the diameter where the integral equals
respective $N_{\mathrm{CCN}}$, the critical diameter $(D_{\mathrm{crit}})$. Typical values are $\kappa = 0.0$ for insoluble material, $\kappa = 0.1$ for water soluble
organics, $\kappa = 0.61$ for ammonium sulfate, and $\kappa = 1.3$ for sodium chloride. Here, under the assumption that particle activation



is foremost size-dependent Dusek et al. (2006), $\kappa_{1.0}$ provides information on the chemical composition of the whole sampled size-ranged, including both particles of larger and smaller diameter.

**OV47 : Ice nuclei number concentration at $-8\,°C$ from off-line low-volume PM10 filter sampling** $(N_{\mathrm{INP,LV,}-8})$

The number concentration of INP at $-8\,°C$ $(N_{\mathrm{INP,}-8,\mathrm{LV}})$ is derived from off-line low-volume filter sampling of PM10 particles. Ice activity at this temperature range is usually associated with biological particles (Kanji et al., 2017; O'Sullivan et al., 2018).

**OV48 : Ice nuclei number concentration at $-20\,°C$ from off-line low-volume PM10 filter sampling** $(N_{\mathrm{INP,LV,}-20})$

The number concentration of INP at $-20\,°C$ $(N_{\mathrm{INP,}-20,\mathrm{LV}})$ is derived from off-line low-volume filter sampling of PM10 particles. Ice activity at this temperature range is usually associated with mineral dust (Murray et al., 2012; Welti et al., 2018).

**OV49 : Ice nuclei number concentration at $-8\,°C$ from off-line high-volume PM10 filter sampling** $(N_{\mathrm{INP,HV,}-8})$

The number concentration of INP at $-8\,°C$ $(N_{\mathrm{INP,}-8,\mathrm{HV}})$ is derived from off-line high-volume filter sampling of PM10 particles. Ice activity at this temperature range is usually associated with signals from biological particles (Kanji et al., 2017; O'Sullivan et al., 2018).

**OV50 : Ice nuclei number concentration at $-20\,°C$ from off-line high-volume PM10 filter sampling** $(N_{\mathrm{INP,HV,}-20})$

The number concentration of INP at $-20\,°C$ $(N_{\mathrm{INP,}-20,\mathrm{HV}})$ is derived from off-line high-volume filter sampling of PM10 particles. Ice activity at this temperature range is usually associated with signals from mineral dust (Welti et al., 2018).

**OV51 : Ratio of fluorescent to total aerosol particle number concentration for fine particles** $(r_{\mathrm{fluo,fine}3\sigma})$

The ratio of fluorescence to total aerosol particles number concentration with optical diameter smaller than $1\,\mu\mathrm{m}$ and larger than $500\,\mathrm{nm}$. The aerosol particles were measured by a wideband integrated bioaerosol sensor.

**OV52 : Ratio of fluorescent to total aerosol particle number concentration for coarse particles** $(r_{\mathrm{fluo,coarse}3\sigma})$

The ratio of fluorescence to total aerosol particles number concentration with optical diameter larger than $1\,\mu\mathrm{m}$ and smaller than 20 um. The aerosol particles were measured by a wideband integrated bioaerosol sensor.

**OV53 : Significant significant wave height** $(H_s)$

Significant significant wave height is defined as four times the standard deviation of the surface elevation or equivalently as four times the square root of the zero-order moment (area) of the wave energy spectrum. Significant significant wave height is the integral measure of the total wave energy of the sea state and represents the average of the highest one third of the waves.





**OV54 : Spectral mean wave energy period** ($T_{m-1,1}$)

Mean Wave period is the characteristic time between two wave crests. It is associated with the wavelength and phase velocity
(i.e. rate at which the wave propagates) via the linear dispersion relation. It provides a proxy of the distance from generation,
i.e. short periods are associated to waves generated by the local wind, while long periods are linked to fully developed swell
waves. Used in conjunction with the significant wave height ($H_s$), defines the shape of waves (steepness) a parameter that char-
acterises that can be determined defining nonlinear wave properties and can be used as a proxy of breaking probability. Here,
the spectral mean wave energy period is used and is obtained using the reciprocal frequency moment of the wave spectrum.


**OV55 : Sea-surface height** (SSH)

Sea-surface height (SSH) is the satellite-derived absolute dynamic topography, which has been interpolated to the cruise track
by Haumann et al. (2020c). Daily fields (0.25° resolution) are from the gridded and merged SSALTO/DUACS Delayed-Time
Level-4 multi-satellite altimetry observations measurements (Altika Drifting Phase, Cryosat-2, Jason-3, OSTM/Jason-2 Inter-
leaved, Sentinel-3A; version 5.9) distributed by CMEMS/Mercator Ocean (http://marine.copernicus.eu) and created on June
14[th], 2018. We use the absolute dynamic topography, which is the sea surface height above geoid. We use the 1-minute GPS
date, latitude, and longitude record (Thomas and Pina Estany, 2019) to find the closest points in space and time in the gridded
satellite product. We interpolate the four spatially closest satellite data points to the ship's location using a distance weighted
mean. This is done for the two closest fields in time, i.e. the daily average (centered at noon UTC) before and after the GPS
record. These two points are then interpolated to the GPS time using a distance weighted mean. Note that while the resolution
of the record is 1-minute along the cruise track, the actual temporal and spatial resolution is determined by the original product.
That means that the dataset for example does not capture any daily cycle as the satellite data consists of daily means.

**OV56 : Surface ocean geostrophic velocity** ($U_g$)

Surface ocean geostrophic velocity ($U_g$) is calculated from the satellite-derived absolute geostrophic velocity zonal and merid-
ional components, which have been interpolated to the cruise track by Haumann et al. (2020c). Daily fields (0.25° resolu-
tion) are from the gridded and merged SSALTO/DUACS Delayed-Time Level-4 multi-satellite altimetry observations mea-
surements (Altika Drifting Phase, Cryosat-2, Jason-3, OSTM/Jason-2 Interleaved, Sentinel-3A; version 5.9) distributed by
CMEMS/Mercator Ocean (http://marine.copernicus.eu) and created on June 14[th], 2018. We use both the zonal and meridional
components of the absolute geostrophic velocity. We use the 1-minute GPS date, latitude, and longitude record (Thomas and
Pina Estany, 2019) to find the closest points in space and time in the gridded satellite product. We interpolate the four spatially
closest satellite data points to the ship's location using a distance weighted mean. This is done for the two closest fields in
time, i.e. the daily average (centered at noon UTC) before and after the GPS record. These two points are then interpolated to
the GPS time using a distance weighted mean. Note that while the resolution of the record is 1-minute along the cruise track,
the actual temporal and spatial resolution is determined by the original product. That means that the dataset for example does
not capture any daily cycle as the satellite data consists of daily means. We then calculate the magnitude of the surface ocean





geostrophic velocity ($U_{\mathrm{g}}$) as the square-root of the sum of the squared velocity components.

**OV57 : Surface ocean temperature of seawater** ($T_{\mathrm{sw}}$)

Surface ocean temperature ($T_{sw}$) is largely derived from the thermosalinograph connected to the underway line (Haumann et al., 2020c), which has been corrected using the surface ocean temperature data from CTD (Henry et al., 2020) and Expendable Bathythermograph (XBT) probe (Haumann et al., 2020d) measurements. Temperature data from the thermosalinograph has been merged with satellite-derived sea-surface temperature (Reynolds et al., 2007). For this purpose, we use satellite-derived, daily (0.25° resolution) NOAA Optimally Interpolated Sea Surface Temperature data (version 2; Reynolds et al.,

2007), which was downloaded March 23$^{\mathrm{rd}}$, 2018. We use the 1-minute GPS date, latitude, and longitude record (Thomas and Pina Estany, 2019) to find the closest points in space and time in the gridded satellite product. We interpolate the four spatially closest satellite data points to the ship's location using a distance weighted mean. This is done for the two closest fields in time, i.e. the daily average (centered at noon UTC) before and after the GPS record. These two points are then interpolated to the GPS time using a distance weighted mean. Note that while the resolution of the record is 1-minute along the cruise track, the

actual temporal and spatial resolution is determined by the original product. That means that the satellite part of the temperature dataset for example does not capture any daily cycle as the satellite data consists of daily means. Missing data is filled with the satellite-derived temperature whenever no thermosalinograph temperature was available within ±0.25 days. Any remaining gaps are interpolated using the Matlab Modified Akima method. Matlab's "Modified Akima cubic Hermite interpolation" is designed to avoid overshoots and is based on a piecewise function of third-order polynomials.


**OV58 : Surface ocean mixed layer depth** (MLD)

Surface ocean mixed-layer depth (MLD) is estimated from both the CTD (Henry et al., 2020) and XBT vertical temperature profiles using the temperature threshold criterion (de Boyer Montégut et al., 2004) and is distributed as part of the XBT data publication (Haumann et al., 2020d). Note that due to the limited number of vertical temperature profiles, the spatio-temporal

availability of MLD along the cruise track is very limited.

**OV59 : Median light intensity within the mixed layer** ($I_{\mathrm{g}}$)

The median light intensity within the mixed layer ($I_{g}$; $\mu\mathrm{mol\,photons}\,m^{-2}\,s^{-1}$) is a derived parameter which aims to consolidate the vertical and diurnal variability in the photosynthetically active radiation (400 to 700 nm) light intensity within the

upper mixed layer of the surface ocean into one parameter. Irradiance from the sun typically follows a sinusoidal pattern at the surface over the course of a day while decreasing exponentially with depth in the ocean. Phytoplankton cells are mixed freely within the upper mixed layer of the surface ocean and so this parameter represents the acclimation irradiance for phytoplankton and provides an indication of their 'light history' which influences various photosynthetic and metabolic processes.

**OV60 : Surface ocean potential density anomaly of seawater** ($\sigma_{0,\mathrm{sw}}$)

Potential density anomaly referenced to 0 dbar pressure ($\sigma_{0,sw}$) is calculated from merged surface ocean temperature ($T_{\mathrm{sw}}$) and





salinity ($S_{sw}$) data (Haumann et al., 2020c) using the Matlab GSW toolbox (TEOS-10; McDougall and Barker, 2011). Here, the anomaly is defined with respect to $1000\,\mathrm{kg\,m^{-3}}$.

**OV61 : Surface ocean salinity of seawater** ($S_{sw}$)

Surface ocean salinity ($S_{sw}$) is largely derived from the thermosalinograph connected to the underway line (Haumann et al., 2020c), which has been corrected using discrete seawater salinity samples collected from the underway line (Haumann et al., 2020a), whenever both products were available. Missing data in the thermosalinograph time series is filled with the measurements from the discrete seawater salinity samples whenever no thermosalinograph salinity was available within $\pm 0.25$ days. Any remaining gaps are filled with interpolated values, whenever at least one salinity measurement was available prior and after each time step within a 6-hour window. The filling values are interpolated using Matlab's "Modified Akima cubic Hermite interpolation", which is designed to avoid overshoots and is based on a piecewise function of third-order polynomials.

**OV62 : Sea-ice concentration** ($C_i$)

Sea-ice concentration ($C_i$) is the fraction of the surface area covered by sea ice. It is linearly interpolated to the cruise track Haumann et al. (2020c) from satellite-derived, daily (25 km by 25 km resolution) NOAA/NSIDC Climate Data Record of Passive Microwave Daily Southern Hemisphere Sea Ice Concentration (version 3; Meier et al., 2013; Peng et al., 2013), which was created on November 30[th], 2017. We use the 1-minute GPS date, latitude, and longitude record (Thomas and Pina Estany, 2019) to find the closest points in space and time in the gridded satellite product. We interpolate the four spatially closest satellite data points to the ship's location using a distance weighted mean. This is done for the two closest fields in time, i.e. the daily average (centered at noon UTC) before and after the GPS record. These two points are then interpolated to the GPS time using a distance weighted mean. Note that while the resolution of the record is 1-minute along the cruise track, the actual temporal and spatial resolution is determined by the original product. That means that the dataset for example does not capture any daily cycle as the satellite data consists of daily means.

**OV63 : Surface ocean $\delta^{18}$O of seawater** ($\delta^{18}\mathrm{O}_{sw}$)

The surface ocean seawater oxygen isotopic composition ($\delta^{18}\mathrm{O}_{sw}$, OV202) was measured in discrete seawater samples collected from the underway line (Haumann et al., 2019). All samples were analysed for their oxygen isotopic composition (reported as permille deviation of the oxygen-18 to oxygen-16 ratio from VSMOW2) by mass spectrometry at the British Geological Survey. $\delta^{18}\mathrm{O}_{sw}$ provides insights into the type of surface freshwater fluxes (precipitation, evaporation, sea-ice melting and freezing, iceberg and land-ice melting) that determine the salinity of the seawater.

**OV64 : Total chlorophyll *a*** (TChl *a*)

Total chlorophyll *a* concentration ($\mathrm{mg\,m^{-3}}$) is the sum of monovinyl and divinyl forms of chlorophyll *a*. Chlorophyll is the main light harvesting pigment found in phytoplankton, photosynthetic microbes found in the sunlit layers of the ocean and responsible for primary production. The concentration of total chlorophyll *a* is often used as a proxy for overall phytoplankton





abundance and biomass however the concentration per cell can vary depending on the species and environmental factors such as light and dissolved nutrient availability.

**OV65 : Chlorophyll *a* fluorescence** (Chl $a_{\mathrm{fluo}}$)

Chlorophyll *a* fluorescence here refers to the in vivo pigment fluorescence as determined with a WetLabs ECO sensor located online in the ship's underway surface seawater flow. The sensor emits a light beam at 470 nm and measures backscatter fluorescence at 695 nm, which is specific for chlorophylls. Being an in vivo fluorescence measurement, it is subject to light-driven fluctuations due to non-photochemical quenching. Thus, the signal is higher at night and lower during the day.

**OV66 : Chlorophyllide *a* pigment concentration** (Chlide $a$)

Chlorophyllide *a* (mg m$^{-3}$) is a transformation product of chlorophyll *a* (see TChl $a$), the dominant pigment found in phytoplankton, and has been associated with scenscent phytoplankton cells and damaged diatoms.

**OV67 : Pheophorbide *a* pigment concentration** (Phaeob $a$)

Pheophorbide *a* (mg m$^{-3}$) is a transformation product of chlorophyll *a* (see TChl $a$), the dominant pigment found in phytoplankton, and has been associated with zooplankton grazing activity and found in sediments, as well as material trapped in sea-ice.

**OV68 : Phaeophytin *a* pigment concentration** (Phaeophy $a$)

Pheophytin *a* (mg m$^{-3}$) is a transformation product of chlorophyll *a* (see TChl $a$), the dominant pigment found in phytoplankton, and has been associated with, and has been associated with scenscent phytoplankton cells and damaged diatoms.

**OV69 : Slope of the particle size distribution** (PSD$_{\mathrm{slope}}$)

The particle size distribution in the surface ocean was measured by counting particle concentrations (living and detrital) in discrete size bins between 2 to 60 µM although the maximum particle size observed in the ACE dataset in the surface ocean was about 20 µM. The slope of the particle size distribution, estimated using a power law (or 'Junge type' approximation), provides a numerical index which indicates shifts in the relative proportion of small vs large size particles where a steeper slope indicates a greater proportion of small vs large size particles. In the open ocean, this parameter typically agrees with major shifts in the composition of the microbial community, particularly the phytoplankton. In locations closer to land masses or sea ice, the particle size distribution slope may also be influenced by additional contributions of detrital particles.

**OV70 : Photochemical efficiency of photosystem II measured during the night** ($F_V F_M$)

$F_V F_M$ is the maximum photochemical efficiency of photosystem II in phytoplankton when the cells are in their fully relaxed state (or after a period of darkness). More generally this parameter is considered as the maximum photosynthetic efficiency or utilisation of absorbed light to drive the electron transport chain. $F_V F_M$ is influenced by environmental factors such as light





and dissolved nutrient availability where excess light can result in an a reduction in $F_V F_M$ which can last for hours after the event and a lack of available nutrients can also result in lower $F_V F_M$. Additionally, the maximum $F_V F_M$ achievable can be constrained by taxonomic differences and hence the $F_V F_M$ is influenced by phytoplankton community structure. Note that this is a bulk water measurement.

**OV71 : Photochemical efficiency of photosystem II during the day** ($\Phi_{PSII}{}'$)

$\Phi_{PSII}{}'$ is the effective photochemical efficiency of photosystem II in phytoplankton when the cells are exposed to light. $\Phi_{PSII}{}'$ is lower than the $F_V F_M$ (see $F_V F_M$) of the sample sample because light exposure results in an increase in the baseline fluorescence due to closed photosynthetic reaction centres and quenching of the maximum fluorescence as a means to dissipate excess light. More generally this parameter $F_V F_M$ is considered as the photosynthetic efficiency or utilisation of absorbed light to drive the electron transport chain. Like $F_V F_M$, $\Phi_{PSII}{}'$ is influenced by environmental factors such as light and dissolved nutrient availability where excess light can result in an a further reduction in $\Phi_{PSII}{}'$ which can last for hours after the event and a lack of available nutrients can also result in lower $\Phi_{PSII}{}'$. Additionally, the maximum $\Phi_{PSII}{}'$ achievable can be constrained by taxonomic differences and hence the $\Phi_{PSII}{}'$ is influenced by phytoplankton community structure. Note that this is a bulk water measurement.

**OV72 : Functional absorption cross section of PSII during the night** ($\sigma_{PSII}$)

$\sigma_{PSII}$ ($\text{Å}^2\,\text{RCII}^{-1}$) is the functional absorption cross section of photosystem II in phytoplankton when cells are in a fully relaxed state (or after a period of darkness). More generally this parameter is the area of the cell harvesting incoming light and actively directing the light energy towards photosynthesis. The size of the functional absorption cross section reflects differences in taxa as well as short-term and long term responses to environmental conditions, namely light and dissolved nutrient concentrations. Note that this is a bulk water measurement.

**OV73 : Functional absorption cross section of PSII during in the day** ($\sigma_{PSII}{}'$)

$\sigma_{PSII}{}'$ ($\text{Å}^2\,\text{RCII}^{-1}$) is the functional absorption cross section of photosystem II in phytoplankton when the cells are exposed to light. More generally this parameter is the area of the cell harvesting incoming light and actively directing the light energy towards photosynthesis. $\sigma_{PSII}{}'$ measured in the light is generally lower than $\sigma_{PSII}$ (see $\sigma_{PSII}{}'$) measured after darkness (or when cells are fully relaxed) due to fluorescence quenching processes dissipating excess light. The size of the functional absorption cross section reflects differences in taxa as well as short-term and long term responses to environmental conditions, namely light and dissolved nutrient concentrations. Note that this is a bulk water measurement.

**OV74 : Chlorophyte contribution to chlorophyll biomass** (Chloro)

The parameter Chloro, shortened from 'CHLORO-1 pigment type', is an estimation of the contribution of phytoplankton species within the division *Chlorophyta* to the chlorophyll *a* pigment biomass ($\text{mg}\,\text{m}^{-3}$) in a sample, and by proxy contribution to the total phytoplankton biomass. Phytoplankton populations are composed of many species which have different





biogeography in response to changing environmental conditions (e.g. light, temperature, dissolved macro- and micro-nutrients) and different biogeochemical or function roles in the ocean. Examples of chlorophytes found in the Southern Ocean include *Chlorella sp.*


**OV75 : Cryptophyte contribution to chlorophyll biomass** (Crypto)

The parameter Crypto, shortened from 'CRYPTO-1 pigment type', is an estimation of the contribution of phytoplankton species within the division *Cryptophyta* to the chlorophyll *a* pigment biomass (mg m$^{-3}$) in a sample, and by proxy contribution to the total phytoplankton biomass. Phytoplankton populations are composed of many species which have different biogeography in

response to changing environmental conditions (e.g. light, temperature, dissolved macro- and micro-nutrients) and different biogeochemical or function roles in the ocean. Examples of cryptophtes found in the Southern Ocean include *Cyptomonas cf. actuta* and other *Cryptomonas sp.*

**OV76 : Cyanobacteria type 2 contribution to chlorophyll biomass** (Cyano)

The parameter Cyano, shortened from 'CYANO-2 pigment type', is an estimation of the contribution of phytoplankton species within the division *Cyanophyta* to the chlorophyll *a* pigment biomass (mg m$^{-3}$) in a sample, and by proxy contribution to the total phytoplankton biomass. Phytoplankton populations are composed of many species which have different biogeography in response to changing environmental conditions (e.g. light, temperature, dissolved macro- and micro-nutrients) and different biogeochemical or function roles in the ocean. Examples of cyanobacteria found in the Southern Ocean include *Synechococcus*

*sp.*

**OV77 : Diatom type contribution to chlorophyll biomass** (DiatA)

The parameter DiatA, shortened from 'DIATOM-1 pigment type' (chlorophyll *c1*, chlorophyll *c2* and fucoxanthin containing), is an estimation of the contribution of phytoplankton species within the class *Bacillariophyceae* (commonly known as diatoms)

to the chlorophyll *a* pigment biomass (mg m$^{-3}$) in a sample, and by proxy contribution to the total phytoplankton biomass. Phytoplankton populations are composed of many species which have different biogeography in response to changing environmental conditions (e.g. light, temperature, dissolved macro- and micro-nutrients) and different biogeochemical or function roles in the ocean. Examples of pigment type-1 bacillariophytes (commonly known as diatoms) found in the Southern Ocean include *Chaetoceros sp.* such as *C. debilis, C. brevis* and *C. dichaeta* and *Phaeodactylum tricornutum.*


**OV78 : Diatom type 2 contribution to chlorophyll biomass** (DiatB)

The parameter DiatB, shortened from 'DIATOM-2 pigment type' (chlorophyll *c2*, chlorophyll *c3* and fucoxanthin containing), is an estimation of the contribution of phytoplankton species within the class *Bacillariophyceae* (commonly known as diatoms) to the chlorophyll *a* pigment biomass (mg m$^{-3}$) in a sample, and by proxy contribution to the total phytoplankton biomass.

Phytoplankton populations are composed of many species which have different biogeography in response to changing environmental conditions (e.g. light, temperature, dissolved macro- and micro-nutrients) and different biogeochemical or function





roles in the ocean. Examples of pigment type-2 bacillariophytes (commonly known as diatoms) found in the Southern Ocean include *Pseudonitzschia sp*. Such as *P. hemeii, P. barkeyi*.

**OV79 : Dinoflagellate type 1 contribution to chlorophyll biomass** (Dino)


The parameter DinoA, shortened from 'DINO-1 pigment type' (lacking chlorophyll *c3*), is an estimation of the contribution of phytoplankton species within the class *Dinophyceae* (commonly known as dinoflagellates) to the chlorophyll *a* pigment biomass (mg m$^{-3}$) in a sample, and by proxy contribution to the total phytoplankton biomass. Phytoplankton populations are composed of many species which have different biogeography in response to changing environmental conditions (e.g. light, temperature, dissolved macro- and micro-nutrients) and different biogeochemical or function roles in the ocean. An example of a pigment type-1dinophyte (commonly known as dinoflagellates) found in the Southern Ocean is *Amphidinium carterae*.


**OV80 : Haptophyte type 8 contribution to chlorophyll biomass** (Hapto8)

The parameter Hapto8, shortened from 'HAPTO-8 pigment type' (lacking 19'-hexanoloxyfucoxanthin), is an estimation of the


contribution of phytoplankton species within the division *Haptophyta* and class *Prymnesiophyceae* to the chlorophyll *a* pigment biomass (mg m$^{-3}$) in a sample, and by proxy contribution to the total phytoplankton biomass. Phytoplankton populations are composed of many species which have different biogeography in response to changing environmental conditions (e.g. light, temperature, dissolved macro- and micro-nutrients) and different biogeochemical or function roles in the ocean. An example of a pigment type-8 prymnesiophyte found in the Southern Ocean is *Phaeocystis antarctica*.


**OV81 : Haptophyte type 6&7 contribution to chlorophyll biomass** (Hapto67)

The parameter Hapto67, shortened from 'HAPTO-67 pigment type' (containing 19'-hexanoloxyfucoxanthin), is an estimation of the contribution of phytoplankton species within the division *Haptophyta* and class *Prymnesiophyceae* to the chlorophyll *a* pigment biomass (mg m$^{-3}$) in a sample, and by proxy contribution to the total phytoplankton biomass. Phytoplankton popu-


lations are composed of many species which have different biogeography in response to changing environmental conditions (e.g. light, temperature, dissolved macro- and micro-nutrients) and different biogeochemical or function roles in the ocean. An example of a pigment type-67 prymnesiophyte found in the Southern Ocean is *Emiliania huxleyi*.

**OV82 : Prasinophyte type 3 contribution to chlorophyll biomass** (Prasino)


The parameter Pras3, shortened from 'PRAS-3 pigment type' (containing prasinoxanthin), is an estimation of the contribution of phytoplankton species within the class *Prasinophyceae* to the chlorophyll *a* pigment biomass (mg m$^{-3}$) in a sample, and by proxy contribution to the total phytoplankton biomass. Phytoplankton populations are composed of many species which have different biogeography in response to changing environmental conditions (e.g. light, temperature, dissolved macro- and micro-nutrients) and different biogeochemical or function roles in the ocean. An example of a pigment type-3 prasinophyte


found in the Southern Ocean is *Micromonas sp*.





**OV83 : Pelagophyte contribution to chlorophyll biomass** (Pelago)

The parameter Pelago, shortened from 'PELAGO-1 pigment type', is an estimation of the contribution of phytoplankton species within the class *Pelagophyceae* to the chlorophyll *a* pigment biomass ($\text{mg m}^{-3}$) in a sample, and by proxy contribution to the

total phytoplankton biomass. Phytoplankton populations are composed of many species which have different biogeography in response to changing environmental conditions (e.g. light, temperature, dissolved macro- and micro-nutrients) and different biogeochemical or function roles in the ocean. An example of a pigment type-1 pelagophyte found in the Southern Ocean is *Pelagomonas sp.*

**OV84 : Total bacterial abundance** ($N_{\text{totalbacteria}}$)

Total bacteria actually refers to total heterotrophic prokaryotes, which include non-chloroplast-containing bacteria and archaea. However, the abundance of bacteria largely overcomes the abundance of archaea. Because of the method used (flow cytometry of single cells), particle-attached cells are not counted and the variable is to be considered free-living heterotrophic prokaryotes.

**OV85 : Abundance of synechococcus** ($N_{\text{synechococcus}}$)

This is the abundance of the cosmopolitan unicellular cyanobacteria within the genus Synechococcus. They are identified by their unique autofluorescence vs size signal in a flow cytometer. Synechococcus are primary producers that belong in picophytoplankton. They are one of the most abundant organisms on Earth, yet they thrive better in warmer waters than in polar waters.

**OV86 : Abundance of nanoeukaryotes** ($N_{\text{nanoeukaryotes}}$)

Nanoeukaryotes here refer to nano-sized (3 to $10\,\mu\text{m}$) eukaryotic phytoplankton. Because of the method used to count them (autofluorescence vs size in a flow cytometer) they only include chloroplast-containing cells. Nanoeukaryotes is an operational, not taxonomical, definition. Actually, the group embraces a great variety of taxonomic groups, including haptophytes, chlorophytes, small dinoflagellates and small diatoms.


**OV87 : Abundance of picoeukaryotes** ($N_{\text{picoeukaryotes}}$)

Picoeukaryotes here refer to pico-sized ($< 3\,\mu\text{m}$) eukaryotic phytoplankton. Because of the method used to count them (autofluorescence vs size in a flow cytometer) they only include chloroplast-containing cells. Picoeukaryotes is an operational, not taxonomical, definition. Actually, the group embraces the smallest representatives of a great variety of taxonomic groups,

typically including large numbers of haptophytes and prasinophytes.

**OV88 : Dissolved nitrate concentration in seawater** (Nitrate)

Nitrogen is an essential component of amino acids in phytoplankton and essential for their growth and productivity. In the ocean it is available to phytoplankton in three inorganic forms, nitrate ($NO_3^-$), nitrite ($NO_2^-$) and ammonium ($NH_4^+$), as well

as organic forms. Dissolved nitrate ($\mu$M) is usually the most abundant form of the dissolved inorganic nitrogen sources in the ocean and generally concentrations in the Southern Ocean are some of the highest of any oceans across the globe but do vary





spatially and temporally.

**OV89 : Dissolved nitrite concentration in seawater** (Nitrite)

Nitrogen is an essential component of amino acids in phytoplankton and essential for their growth and productivity. In the ocean it is available to phytoplankton in three inorganic forms, nitrate ($NO_3^-$), nitrite ($NO_2^-$) and ammonium ($NH_4^+$), as well as organic forms. Nitrite ($NO_2^-$; µM) is an intermediate form of inorganic nitrogen formed when microorganisms oxidise organic nitrogen back to inorganic nitrate. Dissolved nitrite is typically found in low concentrations but concentrations in the Southern Ocean are generally higher than other oceans and varies spatially and temporally.

**OV90 : Dissolved phosphate concentration in seawater** (Phosphate)

Dissolved inorganic phosphate in seawater ($PO_4^{3-}$; µM) is one of the essential macronutrients required for phytoplankton growth. Phytoplankton utilise phosphorus for protein synthesis, construction of phospholipids in cellular structure, storage and transmission of genetic information, metabolic signaling, energy transduction (adenosine triphosphate; ATP) and stress responses. Generally phosphate concentrations in the Southern Ocean are some of the highest of any ocean across the globe, however the distribution is spatially and temporally variable.

**OV91 : Dissolved silicate concentration in seawater** (Silicate)

Dissolved inorganic silicate in seawater ($Si(OH)_4$; µM) is one of the essential macronutrients required for phytoplankton growth. Phytoplankton have a small demand for silicon for protein synthesis, but more importantly the phytoplankton phylum Bacilliariophyta (more commonly known as Diatoms) require silicon for the synthesis of their cell walls. Diatoms play a significant biogeochemical role in the Southern Ocean and are responsible for increased productivity and carbon export. Generally silicate concentrations in the Southern Ocean are some of the highest of any ocean across the globe, however the distribution is spatially and temporally variable.

**OV92 : Dissolved ammonium in seawater** (Ammonium)

Nitrogen is an essential component of amino acids in phytoplankton and essential for their growth and productivity. In the ocean it is available to phytoplankton in three inorganic forms, nitrate ($NO_3^-$), nitrite ($NO_2^-$) and ammonium ($NH_4^+$), as well as organic forms. Ammonium ($NH_4^+$) is an indicator of biological activity where the mixed-layer concentration is controlled by the balance between excretion from zooplankton and uptake by phytoplankton. Higher dissolved ammonium concentrations are associated with areas of increased biological activity, often in close proximity to landmasses.

**OV93 : Particulate organic carbon** (POC)

Particulate organic carbon (POC; µM) measurements are estimations of the organic carbon content of living and detrital particles (including fecal pellets, senescent cells, aggregated material and terrestrially-derived organic matter), and for this dataset include particles $> 0.7$ µm in size in the surface ocean. Under the right conditions, particulate organic carbon is exported to the





sea-floor as part of the biological pump and the concentrations in the surface ocean are linked to the abundance and productivity of phytoplankton who synthesise POC through photosynthesis. Processes such as heterotrophy and grazing by zooplankton can also impact the POC concentration.


**OV94 : Particulate organic nitrogen** (PON)

Particulate organic nitrogen (PON; µM) measurements are estimations of the organic nitrogen content of living and detrital particles (including fecal pellets, senescent cells, aggregated material and terrestrially-derived organic matter), and for this dataset include particles $> 0.7\,\mu m$ in size in the surface ocean. Particulate organic nitrogen enters the ocean cycle through the uptake

of various forms of nitrogen by microbes including phytoplankton and conversion into organic matter or the introduction of terrestrially-derived organic matter containing nitrogen near landmasses or sea-ice.

**OV95 : Particulate organic carbon to nitrogen ratio** (C:N)

Particulate organic carbon (POC) and particulate organic nitrogen (PON) are key components of particulate organic matter

found in the ocean. In this dataset, organic matter content was measured in particles $> 0.7\,\mu m$ in size in the surface ocean including living and detrital particles ((including fecal pellets, senescent cells, aggregated material and terrestrially-derived organic matter). The ratio of POC:PON or simply C:N ratio is often considered in the context of the ecological stoichiometry of plankton and in comparison to the Redfield Ratio (C:N of $\sim 6$), a constant ratio of C:N (and other elements) reflective of the elemental requirements for phytoplankton. Deviations from Redfield Ratio can indicate nutrient limitation and/or acclimation

by phytoplankton to the nutrient supply but may also vary with taxa.

**OV96 : Transparent exopolymeric particles** (TEP)

Transparent exopolymeric particles (TEP) are polysaccharide-rich, gel-like substances that account for a remarkable proportion of particulate organic carbon in the ocean. TEP is an operational definition, referring to the particles stainable with Alcian blue.

Their concentration is calibrated against xanthan gum. TEP are produced mainly by phytoplankton and degraded by bacteria and UV radiation. They can make it to the atmosphere as part of primary organic aerosols.

**OV97 : Coomassie stainable particles** (CSP)

Coomassie stainable particles (CSP) are protein-rich, gel-like substances that account for a not yet quantified proportion of

particulate organic carbon and nitrogen in the ocean. CSP is an operational definition, referring to the particles stainable with Coomassie brilliant blue. Their concentration is calibrated against bovine serum albumin. CSP are produced mainly by phytoplankton and expected to be degraded mainly by bacteria.

**OV98 : Absorption at 350 nm by coloured dissolved organic matter** ($a_{\mathrm{CDOM}}$)

Light in the ocean is attenuated (absorbed and scattered) by seawater, particles and dissolved matter, including coloured dissolved organic matter (CDOM). CDOM is the coloured optically active component of dissolved organic matter (DOM) also





known as dissolved organic carbon (DOC). CDOM absorbs most strongly in the ultraviolet and blue wavelengths, decreasing exponentially towards the red and infrared wavelengths and is generally strongly positively correlated with the concentration of DOC, although this relationship can vary. The parameter absorption by CDOM at 350 nm ($m^{-1}$) can be considered as both

an indicator of factors controlling underwater light attenuation as part of the light absorption budget and as a proxy for DOC.

**OV99 : Slope of detrital particulate absorption** ($a_{\mathrm{napslope}}$)

Detrital particles in the ocean are contributed by many inorganic and organic sources including phytoplankton through the decomposition of senescent cells, food-web interactions with zooplankton, from terrestrial run-off and melting sea-ice, and

sediment resuspension. Absorption by detrital particles is strongest at ultraviolet wavelengths and decays exponentially with longer visible wavelengths. The spectral slope ($nm^{-1}$) parameterises the exponential decay and can offer additional information on the composition and source of the material. Some observations suggest that higher proportions of mineral to organic matter result in higher spectral slopes.

**OV100 : Ratio of detrital absorption to total particulate absorption at 440 nm** ($a_{\mathrm{nap}}/a_{\mathrm{p}}$)

Light in the ocean is attenuated (absorbed and scattered) by seawater, dissolved matter and particles, including phytoplankton and non-phytoplankton particles, sometimes referred to as detritus. Total particulate absorption is equal to the sum of absorption by phytoplankton particles and non-phytoplankton particles. Detritus comes from a variety of sources including the decomposition of senescent phytoplankton cells, from food-web interactions with zooplankton, from terrestrial run-off and

melting sea-ice, and sediment resuspension. Generally in the open ocean the non-algal particulate contribution to the co-varies with the phytoplankton component or chlorophyll $a$, and the non-algal particulate absorption is proportional to the phytoplankton absorption. Increases in the ratio of detrital to total absorption can indicate the additional input of material from terrestrial sources, melting sea-ice or sediment resuspension, as well a shift in the optically active components controlling the attenuation of underwater light.


**OV101 : Acrylate** (acrylate)

The anion acrylate ($CH_2$=CHCOO$^-$) is produced (along with dimethyl sulfide) as a breakup product of the algal osmolyte dimethylsulfoniopropionate. It is toxic at high concentrations but rapidly consumed by bacteria at the nanomolar concentrations at which it occurs in the surface ocean.


**OV102 : Dimethylsulfoniopropionate** (DMSP)

Dimethylsulfoniopropionate (($CH_3$)$_2$S+$CH_2$$CH_2$COO$^-$) is a zwitterion that occurs in the surface ocean at nanomolar concentrations. It is a cosmopolitan compatible solute and osmolyte in phytoplankton and macroalgae, most abundant in haptophytes and dinoflagellates. When released from the algal cell, it is rapidly utilized by bacteria as a source of reduced sulfur for amino

acids, with dimethyl sulfide, acrylate and methanethiol as by-products of DMSP catabolism.





**OV103 : Dimethyl sulfide** (DMS)

Dimethyl sulfide (DMS, $H_3C$-S-$CH_3$) is a biogenic volatile compound produced by the breakup of its major precursor, the algal osmolyte dimethylsulfoniopropionate. Enzymes for DMS production occur in phytoplankton and bacteria. DMS occurs
at nanomolar concentrations in the surface ocean and accounts for 90% of the oceanic emission of volatile sulfur. Once in the atmosphere, it is oxidized within hours mainly to sulfuric and methanesulfonic acids, which participate in aerosol formation and growth.

**OV104 : Carbonyl sulfide** (CSO)
Carbony sulfide (OCS) is a volatile compound produced in the surface ocean (where it occurs at picomolar concentrations) mainly through photochemical reactions of dissolved organic matter. It is by far the most abundant and stable sulfur gas in the atmosphere, from which it can return to the ocean, be taken up by terrestrial plants or reach the stratosphere, where it represents a major source of sulfate.

**OV105 : Carbon disulfide** ($CS_2$)

Carbon disulfide ($CS_2$) is a volatile compound produced by anaerobic decomposition of organic matter or, in the surface ocean (where it occurs at picomolar concentrations), by photochemical reactions of dissolved organic matter. It oxidizes rapidly in the atmosphere.

**OV106 : Air mixing ratio of isoprene** ($Isoprene_{sea}$)

Isoprene, or 2-methyl-1,3-butadiene ($C_5H_8$) is a biogenic volatile organic compound produced by terrestrial plants and oceanic phytoplankton as a by-product of photosynthesis. It occurs in the surface ocean at picomolar concentrations, and, when emitted to the atmosphere, oxidizes within hours, thereby contributing to regulate the atmospheric oxidative capacity and participating in aerosol growth.


**OV107 : Bromoform** ($CHBr_3$)

Bromoform, or tribromomethane ($CHBr_3$) is a volatile compound produced in the surface ocean (where it occurs at picomolar concentrations) by macro- and microalgae, probably as a by-product of oxidative stress. It is short-lived in the troposphere, where it acts to regulate ozone concentration.


**OV108 : Dibromomethane** ($CH_2Br_2$)

Dibromomethane ($CH_2Br_2$) is a volatile compound produced in the surface ocean (where it occurs at picomolar concentrations) by macro- and microalgae, probably as a by-product of oxidative stress. It is short-lived in the troposphere, where it acts to regulate ozone concentration.






**OV109 : Net community production** (NCP)

Net community production (NCP) reflects the balance between photosynthesis and respiration. It is a key parameter describing the marine carbon cycle. At steady state, NCP represents the carbon export production from the sunlit surface ocean. During ACE, NCP was estimated from biological oxygen saturation based on $O_2$/Ar ratios. In brief, surface seawater dissolved $O_2$ and Ar were continuously measured from the ship's flow-through seawater system by an Equilibrator Inlet Mass Spectrometer (**Cassar et al., 2009**). NCP in units of $O_2 \, m^{-2} \, day^{-1}$ was then derived as described in Eveleth et al. (2017) with a modification to wind parameterization (**Teeter et al., 2018**) and a linear correction for sea ice.

**OV110 : Distance to the nearest shore** ($d_{land}$)

The distance to the nearest shore (in metres) has been derived from the GPS track of the R/V-*Akademik Tryoshnikov* (10.5281/zenodo.3772377). Shapefiles of continents have been downloaded from NaturalEarth physical (50 m grid, version 4.1.0). Small islands were added manually as single GPS locations. The distance of the GPS track with shores has been computed with NNJoin plugin (v3.1.2) of qGIS (v3.2.3).

**OV111 : Water depth** ($d_{water}$)

Depth of the seabed along the five-minute averaged cruise track was calculated from the General Bathymetric Chart of the Oceans (GEBCO Compilation Group, 2019) gridded 30-arc second bathymetry data. The nearest gridded value from the bathymetry dataset was used to find the depth at the averaged position.



*Author contributions.* JS, DW, AT, NC, RS conceived the idea. JS supervised the project. DW coordinated the ACE research cruise. CT, IT, JT, CB, MHD, GC, NC, VF, CMR, YL, RA, AA, FAH, AB, PRR, RS, IG, SF, HF, AM, PCG, PRR, LX, MZ, JS, SL, and KCL provided the datasets which were used in this study. MV, FPC and SL developed the analysis and implemented the data ingestion. SL and MV produced the results of the sPCA. SL and JS organized meetings with the co-authors where the results were interpreted and discussed and coordinated the writing process. SL, MV, FAH, IT, CMR, CT, SH, RM, JS, VF, AB, RS, IG, HF, CSH, RA, NC and JT wrote the manuscript together. All authors commented on the manuscript.

*Acknowledgements.* This work would not have been possible without the following contributions: Our gratitude goes first to David Walton, Chief scientist of the ACE cruise and early supporter of this project; we also thank Danièle Rod, Philippe Gillet and Franziska Aemisegger for their contribution to the ACE-DATA proposal. We thank the members of ACE Project 1: David Antoine, Sandy Thomalla, Thomas Ryan-Keogh, Nina Schuback, Hazel Little, David Berliner, William Moutier, Alexandra Olivier-Morgan for providing datasets to this project. We thank members of ACE Project 7: Fiona Tummon for installing the iDirac instrument and operating it during leg 1; Markus Hartmann and André Welti for operating instruments on leg 1 and 3, respectively; Martin Gysel-Beer for great discussions on data analysis and interpretation. We thank Heini Wernli (ACE Project 11) for his ideas and advice during the starting phase of the ACE-DATA project; we thank the members and collaborators of ACE Project 8 Gonzalo L. Pérez, Eva Ortega-Retuerta, Miguel Cabrera, Yaiza Castillo and David J Kieber for assistance with data processing; and we thank the crew of the Akademik Tryoshnikov. The Antarctic Circumnavigation Expedition was funded by the Swiss Polar Institute and Ferring Pharmaceuticals. The collection of meteorological and oceanographic data was supported by Swiss National Science Foundation (SNSF) grant number PZ00P2_142684; MV, SL, and JS were supported by the Swiss Data Science Center; JT, SL, IT were supported by the Swiss Polar Institute; JT was supported by the ACE Foundation; FAH was supported by Swiss National Science Foundation (SNSF) grant numbers P2EZP2_175162 and P400P2_186681, and by the National Science Foundation (NSF) Southern Ocean Carbon and Climate Observations and Modeling (SOCCOM) Project under the NSF Award PLR-1425989, and a grant from the BNP Paribas Foundation; AB was supported by the SNSF Grant No. 200021_169090. IG thanks SNSF grant PZ00P2_142684, SPI, and also FCT/MCTES for the national support to CESAM (UIDP/500017/2020+UIDB/500017/2020) and FCT Project ATLACE (CIR-CNA/CAC/0273/2019) through national funds; SF and HF were supported by the South African National Antarctic Programme and National Research Foundation (SF, HF). SF also recognizes the support of a Royal Society/African Academy of Sciences FLAIR fellowship; AA and AT were supported by the Australian Antarctic Science Program (project 4434). AA is supported by the Japanese Society for the Promotion of Science (PE19055); GC was supported by the cost action of Chemical On-Line cOmpoSition and Source Apportionment of fine aerosol (COLOSSAL, CA16109), a COST related project of the Swiss National Science Foundation, Source apportionment using long-term Aerosol Mass Spectrometry and Aethalometer Measurements (SAMSAM, IZCOZ0_177063), as well as the EU Horizon 2020 Framework Programme via the ERA-PLANET project SMURBS (grant agreement no. 689443); RA was supported by the UK Research and Innovation Natural Environment Research Council grants NE/P008526/1 and NE/P021409/1; and NH, CB and VF were supported by NERC grants NE/K016377/1 and NE/S00579X/1. A workshop funded by NE/P008526/1 aided early discussions for combining ACE datasets; CSH was supported by the Swiss National Foundation for scientific research professor Fellowship PP00P2_166197; NC was supported by the "Laboratoire d'Excellence" LabexMER (ANR-10-LABX-19) and cofunded by a grant from the French government under the program



"Investissements d'Avenir"; CT and SH were supported by the Deutsche Forschungsgemeinschaft (DFG) in the framework of the priority

programme "Antarctic Research with comparative investigations in Arctic ice areas" by grant STR 453/12-1 and HE 6770/3-1; This research was partially supported by the Australian Government through the Australian Research Council's Discovery Projects funding scheme (project DP DP160103387), a South African CSIR Parliamentary Grant (SNA2011112600001) and the CSIR Southern Ocean Carbon and Climate Observatory (SOCCO) programme. CMR was supported by the Australian Government through the Australian Research Council's Discovery Projects funding scheme (project DP DP160103387). RS, MZ, PCG and PRR were supported by the Spanish Ministry of Science

through the BIOGAPS project (CTM2016–81008–R). JS holds the Ingvar Kamprad Chair for Extreme Environments Research.





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
