# Peer review of "Exploring the ocean and atmosphere coupled system with a data science approach applied to observations from the Antarctic Circumnavigation Expedition"

_Earth System Dynamics, 2021_

## Author Comment (AC1)

**Summary of Revisions**

RC = Reviewer comment

We thank all three reviewers for their positive and constructive feedback. In order to provide a quick overview of the changes to the to-be-revised manuscript, we give a summary here:

- The title has been changed to: "Exploring the ocean and atmosphere coupled system with a data science approach applied to observations from the Antarctic Circumnavigation Expedition" (following RC3.3).
- We have added research questions in the introduction for a framework that better structures the manuscript as a whole (following RC1.6).
- The methods description has been revised substantially to make the language more accessible to non-data scientists (following the general and several targeted comments of Reviewer #1).
- Section 5 (description of individual LVs) will be moved to a new appendix A to substantially shorten the manuscript. We now summarize the outcome of all LVs briefly in a revised section 4.1, and highlight the novel aspects we found there as well. We give give one condensed description of LV9 as example in a revised section 4.2.  (following RC1.7, 3.1, 3.4)

RC = Reviewer Comment, AC = Author Comment, new suggested text in blue

**Answers to Reviewer 1**

Anonymous Referee #1, 28 May 2021

Summary and overall impression

**RC 1.1:** This manuscript makes use of a large interdisciplinary dataset from the Antarctic Circumnavigation Expedition, a 90-day cruise from December 2016 to March 2017, in combination with the sparse PCA (sPCA) method to extract process understanding from this comprehensive dataset. The study has a very broad scope, aiming to obtain a holistic understanding of the process biogeochemical and physical processes in the Southern Ocean and atmosphere. The method (sPCA), goes beyond standard PCAs, which are commonly used in oceanography and meteorology. sPCAs aim to increase interpretability when dealing with many variables and processes. In addition, the authors apply a bootstrapping approach in order to quantify the uncertainty of their sPCA results.

I find this a very exciting study and it has the potential to be relevant and valuable to the community. I see three main strengths of the manuscript. First, it presents a method

(sPCA) that is relatively new in Earth System Science and may be useful for further studies analyzing ship data. Second, the method allows the authors to conduct an extremely multidisciplinary analysis including a broad range of observed variables and are able to extract an understanding of the dominant processes in the study region. Third, the study is based on a new comprehensive observational dataset from a historically under-sampled region (the Southern Ocean), and includes measurements in the ocean, atmosphere, and cryosphere, covering all sectors, different interfrontal zones, both open ocean and near islands and continents, and covering a broad range of physical and biogeochemical variables.

At the same time, I have several major comments that I believe need to be addressed before publication. My main concern is the description of the method. I have to admit that I am not too familiar with standard PCAs, and sPCAs are completely new to me. Assuming that this may be the same for many readers, I believe the manuscript can gain considerable clarity by improving the description of the methods (see general comments for more specific details on this and other major comments).

**AC1.1:** We thank the reviewer for the positive and very constructive review. Their comments made our study much more targeted, structured, and understandable. We agree that the description of the method was too technical and we have therefore rewritten the text following the major and general comments below. Please refer to our direct answers there.

General comments

RC1.2: sPCA method: I suggest expanding Section 3.1. I would appreciate a discussion on why setting some weights to zero is ok and why this does not lose crucial information. In a standard PCA, we say e.g., 80% of the variability is linked to OV1, 5% each to OV2 and OV3. We then know that there is a remaining 10% of variability due to other processes. With sPCA (the way I understand it from the manuscript) we reduce the complexity, ignoring some variables, to explain all of the remaining variability. Here, we get to 100%, but we actually know that we weighted many variables with 0 in order to do so. Isn't the standard approach more complete in its interpretation? What are the pros and cons of each? It should also be mentioned if the user chooses which weights are set to zero, or if the algorithm does that. (My apologies if I have misunderstood the sPCA method. If that is the case, I suggest you clarify it).

**AC1.2:** We thank the Reviewer for the valuable comments, these points greatly help in clarifying the methodological sections. We have divided the comment into three points and answer them separately here below.

Point 1 : Is the standard approach not more complete in its interpretation?

The standard approach is only more complete in the sense that in the limit where #LV = #OV, the explained variance will always be 100%, as the Reviewer correctly pointed out. With sPCA, one trades off this full explanation with interpretability, by using an algorithm that sets some OV weights to 0. That is, although standard PCA could explain 100% of the variability, many OV have small associated weights which make it hard to appreciate

their contribution to a given LV, whereas forcing the algorithm to set those unimportant variables to a weight of 0, one can safely reduce the amount of OV contributing to a given LV and therefore facilitating interpretation of each LV.

Proposed manuscript change, L163. New text:

"The standard PCA has the ability to extract 100% of the data variance, when considering a number of LVs which is equal to the number of OVs. While at a first glance this might be a strength of the standard PCA, in fact this comes at the cost of having typically a large number of OVs associated with small weights, which makes it difficult to unambiguously select a subset (or cluster) of OVs relevant for a specific LV. By using the sPCA approach presented here, the algorithm instead optimises these weights, so that some are exactly 0. This approach makes it possible to interpret groups of OVs that contribute to any given LV, and their associated strength, by looking at the subset of OVs with a nonzero weight. Note that if one would discard OVs associated with small weights in standard PCA solutions, the explained variance would decrease and there is no guarantee that the resulting LVs are as different from each other as possible, and therefore containing the least redundant information. In practice, sPCA optimises this thresholding process."

Point 2: What are the pros and cons of each?

The main point in favour of sPCA, as opposed to the standard PCA, is the automatic reduction of the number of OVs contributing to the LVs. This has several benefits. The first advantage is the increased ease of interpretation, which is the main motivation for its use in our context. By discarding a large number of OVs, which the algorithm does not deem necessary, to the construction of a given LV, the users only have to focus on a smaller set of input OVs. In our case, by using such a high dimensional and heterogeneous input set, this was a must, because we cannot select which OV features are important in each LV a priori, in an objective and unbiased manner. Secondly, when only few data points are available, the estimation of the data covariance might become an ill-posed problem when many input dimensions are considered. sPCA circumvents this problem by reducing the dimensionality of the estimation problem. This acts as a regularization, which makes it harder for the sPCA to summarize LV corresponding to noise and minor variations, which usually relate to the low variance components of the standard PCA.

These benefits, however, come at the cost of a harder optimisation problem, which does not guarantee that running the method twice will produce the same solution (non-convex), as opposed to standard PCA which generally has a unique solution. To alleviate this issue, and to actually take advantage of this, we developed a bootstrap approach to quantify uncertainty of the sPCA. In addition, since decomposition weights can be 0, LVs could be correlated, although in practice they are close to orthogonal (uncorrelated). This also makes it impossible to summarize 100% of the variance in the same number of components as for the traditional PCA. However, as the algorithm is not forced to do so, noise components and minor variability modes are automatically discarded, making this robustness to noise also a strength.

Proposed manuscript changes:

- L258: Change subsection 3.5 title as "Model limitations and advantages"
- L277, new paragraph: "The main advantage of the sPCA approach over its standard counterpart is the automatic selection of OVs by assigning non-zero weights for a given LV. The automatic optimisation of the weights associated with the OVs is done sequentially for each LV, starting from the one corresponding to the largest mode of variance. This ensures that, although not exactly, all the LVs are as uncorrelated as possible. The use of sPCA has also the advantage of being less susceptible to noise and unimportant data variations. This advantage can be understood when contrasting the sPCA results with the large number of principal components with very low explained variance of the standard PCA. Although by considering these components the standard PCA is able to fully explain the data variance, such variance directions are of little practical use in our case, as it would be difficult to link them to natural processes. Compared to the standard PCA, sPCA is less likely to return components with very small explained variance, which are usually corresponding to noise. This advantage is further strengthened by our novel use of the bootstrap analysis, which promotes robustness to noise, meaning that OVs which contribute mainly through noise are identified as such. Data is resampled randomly, and the influence of noise can be observed in large fluctuations of the solution. Therefore, analyses relying on aggregated bootstrapped solutions are more robust to the influence of noise than the traditional PCA or even a single run sPCA. Moreover, using sPCA over the standard PCA has also the benefit of not being susceptible to rank-deficient covariance matrices, in particular when the number of data points is smaller compared to the number of OVs. And last, but not least, the exploratory character of the sPCA allows researchers to conduct an untargeted analysis and potentially find relationships or (spatial / temporal) patterns which would have been left undiscovered in a targeted analysis because one did not think of the possibility."

Point 3: It should also be mentioned if the user chooses which weights are set to zero, or if the algorithm does that.

In most implementations of sparse algorithms, the user does not manually set the input weights corresponding to the OV, but these are set by the algorithm itself. This is a step that sPCA does automatically, as being part of its internal optimization routine, which aims at maximizing the variance explained by each LV (starting from the first), under the constraint of using a small subset of all available OV. The user usually has indirect control over it, by selecting a hyperparameter controlling the strength of such an effect. We state how we select the hyperparameters in lines 241-247 in the manuscript.

Proposed manuscript changes:

- L163 as for Point 1.
- L161, new text: : " … hence promoting sparsity. Sparsity is obtained automatically as the solution of Eq. 2 leads to the selection of the smallest possible subset of OVs to maximize the variance of the LV."

**RC1.3:** LVs: I find the current explanation of what an LV is quite confusing (L85-87), which led to further confusion later in the document. I recommend making it really clear here what an LV is in an sPCA and how it is different to the OVs. I recommend explicitly stating that the LVs are the processes we want to understand (i.e., the output from the sPCA) with the help of OVs (i.e., the input to the sPCA). (it becomes clearer later in the document, but is needed early on).

**AC1.3:** Thanks for the comment. This is indeed an important point, and we clarified it in the text. The Reviewer is correct: LVs are the processes, as estimated by the sPCA algorithm, while OVs are the input variables, i.e. the measurements.

Proposed manuscript changes:

- L87, new text: "… of maximal variance. In practice, LVs can be seen as artificial output variables returned by the sPCA algorithm that are linear combinations of the input OVs, i.e the actual measurements. Therefore, LVs are the target variables that we aim to interpret in this study, where each LV summarises a specific aspect of the data, which we relate to natural processes. This approach has the advantage of reducing the 111 OVs that we measured during the cruise to 14 LVs that we can interpret in terms of the processes that they represent."
- L95, new text: "... sparse weight matrix. The OVs with non-zero weights form a subset (a cluster) of variables that are related to each other and compose a specific LV, which can be interpreted with one or several underlying natural processes.

**RC1.4:** Please add a section that summarizes what happens during the sPCA to add clarity on the method for people unfamiliar with it. The way I understand how the sPCA works from your manuscript, the user chooses a set number of processes they want to know about (here: 14), feeds all OVs (here: 111) into the algorithm. Some of the OVs are set weighted 0 to reduce the number of OVs for each LV. (→ This should be discussed and mentioned if this happens randomly.) The algorithm then identifies 14 different sets of OVs. The users then see which OVs have non-zero weights in each LV to determine which process each LV represents. i.e. the user has to make a choice: if sea surface temperature, salinity, and MLD are OVs in an LV, then the LV might represent a process linked to ocean circulation. (→ For each LV, it would be good to know which OVs are in it so that the reader can understand how the label for each LV was chosen). We can then also see the percentage of the variability that process has on the variability in all of the 111 variables.

→ Is this correct? If yes, it might give you hints about which pieces of information the reader might want to hear about. If not, my understood explanation might give you hints about which parts were confusing.

**AC1.4:** The Reviewer is correct in their summary. The only minor feedback we can give, is about the estimation of the weights, which is given in the reply to RC1.1. This is a step that sPCA does automatically, as being part of its internal optimization routine: maximize the variance explained by each LV (starting from the first), under the constraint of using a small subset of all available OVs.

In the original Figures 6, 7, 8, 9, 10, 11, 12, 13, 15, 16, 17, 18, 19, and 20 in the manuscript we show the weights of those OVs that are larger than 2 standard deviations. In the SI we provide lists of all OVs with non-zero weights for each of the 14 LVs. Note that these figures have been moved to the appendix, following a comment by reviewer 3 (RC3.4).

We also think that adding a summary of the main steps of the overall approach is a good idea, and we did so in Section 3.3 (corresponding to subsection 3.4 in the new version of the manuscript).

Proposed manuscript change:
- Switch sections 3.3 "Data preprocessing and model setup" and 3.4 "Missing data and imputation"
- L247, new paragraph. "Our analysis pipeline can be summarized as follows: First, the measurements are preprocessed as described above in order to obtain the input OVs. Then, for each bootstrap, a random subset of data points is sampled, with replacement. This subset is used to compute an sPCA solution with the settings described above. Once all 30 bootstrap solutions are obtained, we perform the alignment of the principal components described in Section 3.2 and compute the distribution of the weights associated with each OV, the distributions of the LV activations, and the average explained variance per principal component. We then interpret these three outputs of the bootstrapped sPCA to understand the underlying natural processes that cause the variability described by each LV."

**RC1.5:** Unimportant variables for an LV "are forced to be zero": could we accidentally lose information here? Is this a subjective choice by the authors or done by the algorithm? This should be discussed further.

**AC1.5:** As the reviewer remarked, the process of setting weights to 0 does not come without caveats. First, this process is automatically done by the algorithm, so there is as little human bias as possible. Usually, OVs that strongly contribute to a given LV (i.e. to a given variance direction) are assigned a non-zero weight, while OVs that do not strongly correlate with the given LV are almost always assigned a weight of 0, because they are noisy and do not carry substantial information. But there is indeed a risk to lose information for variables "in between", and in particular for OVs that are undersampled. In order to minimize this risk, in our setting, we use bootstrapping: resampling and estimation of model weights provides a measure for how much the solutions vary, which tells us about the stability of the assignments. The more stable the solution is, the smaller the risk of losing information. However, as the algorithm does not have guarantees to converge to the global minimum of the optimization, we cannot exclude that some minor information is lost. By bootstrapping, controlling the optimization through hyperparameter selection, and performing missing data imputation, we believe that the obtained LVs are stable and as rich in information as possible.

We proposed to change the manuscript as written in AC1.2.

RC1.6:Research Question(s): Another concern is linked to the research question(s) the article wants to answer. It is such a broad study that scratches on so many topics that it becomes a bit blurry in the introduction where this is all going. The way it is currently presented, it appears as a data mining approach of plugging in all the data and seeing what happens. Were there some hypotheses before that you wanted to test? I would find it helpful to add a (couple of) specific research question(s) and build on that in the introduction why we want to know about that. E.g., Is it about the processes? Is it about showing that sPCAs are a good tool? (or both). Are there some processes we are unsure about, which the sPCA might shine a light on?

**AC1.6:** Thank you for pointing this out. Including some more structure in the manuscript by means of our targeted research questions is a very good idea. We have now included the following in the introduction in l. 42:

"To explore interactions between the Southern Ocean system components, we apply an unsupervised learning method, sparse principal component analysis (sPCA). Application of the sPCA has two objectives: i) conducting an untargeted and therefore more objective analysis of data, where the method is less tailored to the science question as compared to more traditional regression analysis, and ii) to target a set of specific research questions (RQ):

RQ1: Is sparse principal component analysis an adequate tool to extract interaction processes inherent to a heterogeneous and short data set, which describes environmental variability?

RQ2: Is it possible to identify geographic locations ("hotspots") that are common to several interaction processes?

RQ3: Which are the key observed environmental variables that strongly contribute to several interaction processes?

Specific answers to RQ1 are given in section 3.5, with respect to model limitations and advantages, and 6.2, with respect to interaction processes. RQ2 is answered in section 6.1 and RQ3 in section 6.3. Note that we focus on the proof of concept of the sparse principal component method by basing the interpretation primarily on the known processes of the Southern Ocean climate system. New scientific insights from this novel approach are described in section 4.1.

To make the structure of the introduction a bit more evident we introduced the following key words:

L. 43: "Southern Ocean Processes:"

L. 72: "The Expedition:"

L. 82: "Unsupervised learning approach:"

In addition, we would like to point out that one of the key strengths of the sPCA is to allow for a more untargeted and therefore more objective analysis of data, where the method is less tailored to the science question as compared to more traditional regression analysis. Therefore, it arises naturally that our study is not following a clear hypothesis that identifies a specific air-sea interaction process. We also now better clarify this aspect of the analysis in the abstract (lines 24/25):

"The sPCA processing code is available as open-access. As we show here, it can be used for an exploration of environmental data that is less prone to cognitive biases, and confirmation biases in particular, compared to traditional regression analysis that might be affected by the underlying research question."

RC1.7: Linked to my previous comment: it is not clear to me which findings are confirmations of processes we already knew, and which findings are new insights. This should be clarified.

**AC1.7:** This is indeed important and needs more highlighting. We have restructured the manuscript significantly, following this remark and that of reviewer 3, RC3.4. Now all of section 5 has been moved to appendix A. We keep part of the text from former section 5.8 and moved this up to section 4.1. Section 4.1 is now "Short summary of all latent variables and new insights". and contains the text here below, which is merged from the original section 4.1 first paragraph and section 5.8 "Short summary of all latent variables", and contains new additions to highlight the new insights. We also provide a condensed description of LV9 in a new section 4.2 to give one prominent example with new insights. We highlight the new text in blue.

[revised manuscript text omitted]

RC1.8: Eddies: One process that doesn't seem to be covered in this study, but is a known
driver behind variability in the Southern Ocean are mesoscale eddies. This should be
discussed.

**AC1.8:** Thank you for pointing out this limitation of our study, which we had not yet discussed
in detail. The main issue why eddies are not captured by our analysis is the resolution for two
reasons. 1) Most data are sampled only every couple of hours and 2) even if we have
continuous measurement for certain OVs, the 3-hour subsampling/interpolation of the sPCA
input data would filter all mesoscale (eddy) activity. For example, given a ship speed of about
10kn (about 19km/h), the 3-hour interval translates to a spatial resolution of about 57 km,
which is too coarse to capture eddies. At 50degS, the baroclinic Rossby radius of deformation
ranges between 10 and 25km (Chelton et al., 1998). Apparently, this is a substantial limitation
of our study that is not able to capture mesoscale and submesoscale variability in the ocean
and if future studies wanted to focus on the influence of eddy/mesoscale or submesoscale
processes, they would need to use a much higher resolution data set. We have added a
respective comment to clarify this limitation (l. 982 in the original manuscript):

"Moreover, our study is limited to the spatio-temporal scales of the ACE cruise (single
season), the sampling intervals along the cruise track (varies among variables), and the
chosen 3-hourly resolution for the sPCA analysis. This limitation has the important
implication that we cannot identify variations and processes on longer scales, such as
interannual variations, or shorter scales, such as the meso- or submeso-scale. For example,
meso-scale eddies that are an important driver of Southern Ocean variability are not
resolved by our analysis, because the 3-hour interval (about 57 km if the ship moved at 10
knots) is larger than the Rossby radius of deformation at these latitudes (Chelton et al.,
1998)."

References:

Chelton, D. B., deSzoeke, R. A., Schlax, M. G., El Naggar, K., and Siwertz, N. (1998). Geographical Variability of the First Baroclinic Rossby Radius of Deformation. *Journal of Physical Oceanography*, 28(3), 433-460. https://doi.org/10.1175/1520-0485(1998)028%3C0433:GVOTFB%3E2.0.CO;2

RC1.9: Seasonality: Please add a discussion on the fact that the cruise is only 90 days long (i.e., during one season) and that the ship is moving during that time, making it difficult (or impossible?) to conduct a seasonal analysis. The discussion should include why it is possible (or not possible?) to robustly conclude on any seasonal signals with this data.

**AC1.9:** We agree that conclusions on seasonality need to be further discussed and that the constant movement of the ship limits a detailed seasonal interpretation of signals. We have added the following to the manuscript in l. 983:

"Even though the ACE cruise covered a relatively long time period from late December to late March, the robustness of the derived seasonal signals from this dataset is limited. This limitation arises from the ship's movement, thereby covering a wide range of environmental conditions. Thus, signals on time scales such as the seasonal signal depicted by LV7 need to be interpreted as integrated signals occurring on sufficiently large scales. For example, the seasonal variation of the intensity of solar radiation in LV7 shows a decrease anywhere across the Southern Ocean towards austral fall. We can also attribute a seasonal signal to phenomena which only occur during a certain period and certain location. For example, the melting of sea ice discussed in LV9 only occurred in a limited region at the time of the cruise, but it would have been a much more widespread signal if the cruise had taken place in austral spring when the sea ice cover was more extensive. Therefore, it is important to note that we cannot discuss the full seasonal evolution of the signal, because we only spent a few days in the sea ice region, but the input of freshwater from the melting sea ice emerges as an important seasonal phenomenon in our analysis."

Specific and minor comments to the text:

RC1.10: L. 131: In this section, I would have liked to also find out a bit more about the measurements, e.g., if the ocean measurement are at the sea surface only (same for atmosphere) and I recommend adding a sentence or two stating the nature of the measurements (sensors, air/water/ice samples… were some data collected by platforms other than the ship, such as satellites/planes…?).

**AC1.10:**

This is a good point. We have added the following information at the end of section 2.1:

"Generally, all atmospheric measurements were taken from either the container or monkey deck, i.e. 15 m and up to 31.5 m above sea level, respectively. Ocean measurements were either obtained from the underway water line, with an intake at the front of the ship at about 4.5 m below sea level, or from CTD casts. Details on the

sampling locations are given in the cruise report (Walton and Thomas, 2018), whereas details on the measurement methodologies are given in the supplementary information section S1."

RC1.11: L. 145: It should also be mentioned here (and possibly in the abstract/introduction) that this is an unsupervised machine learning approach (as stated in the Conclusion).

**AC1.11:** Thank you for pointing this out. We have rephrased the sentence in L. 146 to: "Sparse PCA, an unsupervised machine learning approach,  was used to…". In the abstract in l. 9, we added: "Sparse PCA is an unsupervised machine learning method."

---

## Author Comment (AC2)

**Summary of Revisions**

RC = Reviewer comment

We thank all three reviewers for their positive and constructive feedback. In order to provide a quick overview of the changes to the to-be-revised manuscript, we give a summary here:

- The title has been changed to: "Exploring the ocean and atmosphere coupled system with a data science approach applied to observations from the Antarctic Circumnavigation Expedition" (following RC3.3).
- We have added research questions in the introduction for a framework that better structures the manuscript as a whole (following RC1.6).
- The methods description has been revised substantially to make the language more accessible to non-data scientists (following the general and several targeted comments of Reviewer #1).
- Section 5 (description of individual LVs) will be moved to a new appendix A to substantially shorten the manuscript. We now summarize the outcome of all LVs briefly in a revised section 4.1, and highlight the novel aspects we found there as well. We give give one condensed description of LV9 as example in a revised section 4.2.  (following RC1.7, 3.1, 3.4)

RC = Reviewer Comment, AC = Author Comment, new suggested text in blue

**Answers to Reviewer 2**

Anonymous Referee #2, 01 Jun 2021

Comments on "Biogeochemistry and Physics of the Southern Ocean-Atmosphere System Explored With Data Science" by Landwehr et al.

**RC2.1:** This manuscript presents a detailed exploration of a very large ensemble of measurements of in-situ variables from the Southern Ocean and from the Southern Atmosphere.  It emphasizes the technique of sparse principal component analysis which indicates possible causal relationships and tries to identify underlying processes explaining how the variations of the observed variables. As it is now the manuscript is well written but it could benefit from incorporating the minor remarks I have below. I also propose to more clearly delineate the advantages of the sPCA method to guide the reader about the choice of analysis made here.

**AC2.1:** We thank the reviewer very much for their positive and constructive remarks. We address all comments below in detail.

Major comments:

**RC2.2:** The following sentence at the end of the discussion (Page 60, lines 1102-1103) would need to be better backed up by the authors: "In summary, we find that the sPCA is not only capable of resolving many of the complex connections between the OVs (Observed Variables) but also to provide estimates of their relative importance for the observed variability of each OV."

**AC2.2:** The reviewer makes a good point. Rereading the section and the first summary sentence, we find that this particular sentence is misplaced here and partly non-sensical. This is because the complex connections are discussed in detail in the individual LV descriptions, where we highlight a number of processes involving several OVs, but this is not the topic of this section 6.3. The second half of the sentence makes grammatically no sense, because "their" refers to "OVs" resulting in "... but also to provide estimates of OVs' relative importance for the observed variability of each OV." What we meant to say is that the occurrence of OVs in several LVs provides insight into where the OV variability might stem from. In order not to repeat information from the individual OV discussions and the section 6.3, we removed this sentence and start the summary as follows:

"In summary, we find that state variables of the environment such as the air-sea temperature difference,..."

**RC2.3:** I would welcome a paragraph stating, with possible examples from the results and the discussion, the strengths of the sPCA method. The weaknesses are well described but the reader would also like to have the view of the authors on what guided them to select this method for an analysis.

**AC2.3:** This is an important point, which we apparently did not communicate very clearly. To make this clearer, we have added the following short paragraph in section 3.5 for a general description of the advantages, and a second paragraph in the introduction to Section "6 Discussion" in l. 982. The discussion section actually highlights some of the aspects which we consider to be key advantages, that is the identification of "hotspots" and of "key OVs". The attribution of a number of processes that explain the variability of each OV is discussed further upfront in the manuscript and is shown in Fig. 5. And last, but not least, the exploratory character of the sPCA allows researchers to conduct an untargeted analysis and potentially find relationships or (spatial / temporal) patterns which would have been left undiscovered in a targeted analysis because one did not think of the possibility.

- L277, new paragraph: "The main advantage of the sPCA approach over its standard counterpart is the automatic selection of OVs by assigning non-zero weights for a given LV. The automatic optimisation of the weights associated with

the OVs is done sequentially for each LV, starting from the one corresponding to the largest mode of variance. This ensures that, although not exactly, all the LVs are as uncorrelated as possible. The use of sPCA has also the advantage of being less susceptible to noise and unimportant data variations. This advantage can be understood when contrasting the sPCA results with the large number of principal components with very low explained variance of the standard PCA. Although by considering these components the standard PCA is able to fully explain the data variance, such variance directions are of little practical use in our case, as it would be difficult to link them to natural processes. Compared to the standard PCA, sPCA is less likely to return components with very small explained variance, which are usually corresponding to noise. This advantage is further strengthened by our novel use of the bootstrap analysis, which promotes robustness to noise, meaning that OVs which contribute mainly through noise are identified as such. Data is resampled randomly, and the influence of noise can be observed in large fluctuations of the solution. Therefore, analyses relying on aggregated bootstrapped solutions are more robust to the influence of noise than the traditional PCA or even a single run sPCA. Moreover, using sPCA over the standard PCA has also the benefit of not being susceptible to rank-deficient covariance matrices, in particular when the number of data points is smaller compared to the number of OVs. And last, but not least, the exploratory character of the sPCA allows researchers to conduct an untargeted analysis and potentially find relationships or (spatial / temporal) patterns which would have been left undiscovered in a targeted analysis because one did not think of the possibility."

- L.982, new paragraph: "The key strengths of the method are: (a) Sparse PCA has an untargeted exploratory character, i.e. the possibility of relating many different OVs with each other and identifying correlations, which one might not intuitively address in a targeted analysis. (b) Because sPCA can easily relate geographical information with all OVs, it is possible to explore spatial patterns and obtain a geographic overview. This also allows us to identify geographical hotspots, as discussed in Section 6.1. (c ) Sparse PCA can help to identify original variables which are key to many processes, as discussed in section 6.3. Due to the possibility of exploring a large number of OVs at the same time, it becomes straightforward to isolate those OVs that stand out. And finally (d), we can explore which processes (LVs) contribute to explaining the variability of the OVs, as is shown in Fig. 5."

**RC2.4:** The distance to the continent (Latent Variable 5, LV5) is not the best indicator of land influences as the authors seem to suggest. A much better indicator would be a $^{222}$Rn concentration measurement. Radon-222 is a radiogenic gas which emission flux is 100 times more important over land than over ocean. As such, you can use the concentration of $^{222}$Rn to trace how long ago an air parcel was over a continent. Several

authors have used this property as a measure of the continental influence of an air parcel travelling over the ocean (Heimann et al. (1990) Balkanski and Jacob (1990)).

**AC2.4:** We agree with the reviewer that 222Rn would be much better suited as a terrestrial tracer. Unfortunately, there were no such measurements undertaken. Hence, we resorted to a simple metric such as distance to land.

Minor comments:

**RC2.5:** Caption of Figure 1: do you really mean "microbial gases" or is it rather "biogenic gases". If you use the terms 'microbial gases' you imply that these gases are exclusively emitted by microbial organisms.

**AC2.5:** This is correct, it should read "biogenic gases". We have corrected the caption.

**RC2.6:** Was there any attempt made to tag the air masses or use back-trajectories to know how long ago this air mass was above continents? It could (for example) explain why certain air masses have a higher $O_3$ content as discussed in lines 457-458 page 26.

**AC2.6:** The ozone mixing ratio is relatively invariant across most of the expedition with two exceptions, that is during the passage of the Balleny Islands for a few days (see Fig. 8 negative activation of the LV East of 180°E) and from South Georgia to Cape Town. Particularly the latter, long period is reflective of the air mass transported between 60 °S and 50 °S as shown with the 48 hrs back trajectories in Fig. 8. This is not necessarily evidence of continental influence. However, using CO as a semi-conservative tracer for continental influence (combustion) beyond 48 hrs, we find a relative concentration increase between South Georgia and Cape Town, which might be an indication of continental influence. The first instance of higher ozone concentrations near Balleny is clearly characterized by Antarctic air mass outflow, where the higher elevation of the continent, from which air masses descended, might have played a role. In light of these observations, we have added the following in l. 459 after "(see Figure 8c)":

"This might indicate enhanced vertical mixing in the marine boundary layer during cold air advection, which might lead to the entrainment of free tropospheric air masses with higher O3 concentration into the marine boundary layer (see Figure 8c). Such entrainment is particularly likely for the high ozone concentrations observed during a cold air outbreak from Antarctica, where air masses descended from further aloft. For elevated ozone concentrations between South Georgia and Cape Town, continental pollution outflow from South America cannot be ruled out, because CO concentrations are also slightly elevated."

**RC2.7:** Lines 480-482: did you check whether the values of RH for these warm air masses. Could the values of RH be an indicator for prior precipitation?

**AC2.7:** As RH in LV3 is a measure of the strength and direction of air-sea moisture fluxes and a tracer of large-scale moisture advection, it is not positively correlated with in situ measurements of rainfall. This is in contrast with LV4, which represents changes in RH due to precipitation events. RH and the amount of precipitation in the five days prior to arrival of the trajectories in the marine boundary layer are indirectly related as can be seen in a very weak, but significant correlation of the two variables in our dataset (Pearson correlation of 0.11 with a p-value of 0.005). We interpret this as a signal of precipitation occurring in the advected warm air mass, that is characterised by high RH. Due to several processes (meridional advection of moist air over a cold ocean surface, precipitation, and long-distance moisture advection), which affect RH in an air mass, and due to high variability of RH during a time period of five days, RH cannot directly be used as a tracer of precipitation during transport. As we can see in this study, the sPCA analysis succeeded in identifying these different time periods, which were affected by the aforementioned moist processes (see LV3, LV4 and LV9). Please, also be aware, that the simulated rainfall along the backward trajectories is only poorly constrained in the study area due to a lack of observations. Therefore, our results regarding precipitation during large-scale transport need to be interpreted carefully and further research is needed to understand the role of precipitation on the cycling of water vapour during warm air advection.

**RC2.8:** Page 31, lines 531-534, the following sentence comes a bit out of nowhere:

"There is no apparent explanation for the inclusion of carbon monoxide (CO), the mass concentration of sulfate in nonrefractory particulate matter (SO$_4^{2-}$), and the atmospheric isoprene concentration (Isoprene**air**), and further analysis is beyond the scope of this work."

You might be missing something important here relative to isoprene. It would be worth investigating or asking other groups to think about this positive correlation between extratropical cyclone activity and isoprene in air. Isn't it simply that isoprene sources are abundant in the subtropical regions and the cyclones channel rapidly air from lower latitudes to the latitudes at which you are making these measurements?

**AC2.8:** It is true that this sentence might appear to come out of nowhere. In fact, it is there, because we consider OV contributions, if their median value of the contributing weight is larger than their single standard deviation from the bootstrap runs. We state this in l. 324f, but we cannot expect the reader to remember this. Hence we added after the sentence:

"We mention them here, because their contributing weight to the LV is larger than their single standard deviation from the bootstrap runs."

Thank you for the hint on the potential transport of isoprene from lower latitudes. This is an interesting point. However, if that were the case one would expect a similar behaviour

for CO (more sources in the subtropics, and it is longer-lived than isoprene). Instead we see CO anticorrelating with LV13 (i.e., low CO and high isoprene when LV13 is activated). In addition to that, previous measurements (albeit sparse) of marine isoprene in subtropical regions (as summarised by Hackenberg et al., GBC, 2017) do not show that isoprene mixing ratios are higher in these regions than at higher latitudes. There are of course higher terrestrial emissions of isoprene in the subtropics, but 1) the short lifetime of isoprene at these latitudes would limit how far it can be transported and 2) one would expect a similar behaviour from CO (see also the previous point).

Reference: Hackenberg, S. C., et al. (2017), Potential controls of isoprene in the surface ocean, *Global Biogeochem. Cycles*, 31, 644– 662, doi:10.1002/2016GB005531.

**RC2.9:** With regards to the results described for LV2: Drivers of the cloud condensation nuclei population. You do not mention that small particle in the nucleation mode will eventually end up in the accumulation mode upon growth and coagulation. Condensation nuclei (CN) that are not activated will join the accumulation mode aerosol.

A very noteworthy reference concerning CCN is the one from Lee et al (2013). The authors studied twenty right parameters that cover all important aerosol processes to understand the cause of uncertainty for CCN.

**AC2.9:** It is correct that we have not discussed the nucleation and Aitken modes in the LV 2 section. This is because we limit the discussion to the OVs which are displayed in the specific LV figures, those are the ones that contribute with their weight beyond one single standard deviation from the bootstrap runs. We understand that for an audience who is more focused on aerosol science it might be unsatisfactory that the discussion is short from an expert's perspective. Given that the manuscript is already lengthy and we have been asked to shorten the discussion by Reviewer 3, we only added the following sentence in l. 680:

"We refer the reader to Lee et al. (2013) for a comprehensive investigation on aerosol processes relevant to CCN number concentrations and their uncertainty."

**RC2.10:** Lines 685-687 why is your hypothesis limited to rainout and does not include washout? "To check our hypothesis concerning rainout, we investigated the precipitation rate along the backward trajectories for the previous three days (see Figure 14)"

**AC2.10:** We actually meant "washout" in general, not specifically removal by rain. We have hence replaced "rainout" by "washout".

**RC2.11:** Paragraph 5.5 why is LV12 not related to $N_{ccn,0.15}$, $N_{ccn,0.30}$ and $N_{ccn,1.0}$? Monahan et al. (1986) parametrization of sea salt emission predicts that these small seasalt aerosols would be abundantly produced at high wind speeds.

**AC2.11:** This is indeed an interesting point. The reviewer is correct that the Monahan et al., (1986) source function predicts the emission of small sea spray particles at high wind speeds. Furthermore, this is supported by more recent sea spay source functions that also predict substantial emissions of small sea spray particles (e.g. de Leeuw et al., 2011). These particles are composed of sea salt and organics, and are therefore hygroscopic and efficient CCN. However, recent aerosol-focused ship-based studies have found that on a number basis and excepting very high wind speed events, sea spray particles still only form minor fractions of the total marine aerosol (typically less than 20%), and consequently, only minor fractions of marine CCN populations (Modini et al., 2015; Quinn et al., 2017; Schmale et al., 2019). Instead, it appears that under typical conditions marine CCN populations are composed primarily of non-sea-salt sulfate aerosols.

The overall sPCA results are consistent with this picture. LV2 contains strong contributions from Nccn at all 3 supersaturations, as well as accumulation mode aerosol number concentrations and aerosol sulfate concentrations. This suggests high correlation between these variables and supports the recent ship-based studies mentioned in the paragraph above. On the contrary and as noted by the reviewer, the Nccn variables do not show up in LV12, which is the LV related to sea spray aerosol. We believe that this is because, on average, sea spray only contributes minor fractions to the Nccn populations, and thus, to a first order, the variability in Nccn is not driven by variability in the number of sea spray particles. This does not preclude the occurrence of very high wind speed events where sea spray completely dominates CCN populations (such extreme cases are discussed for the ACE cruise in Schmale et al., 2019), but it does suggest that these events do not occur frequently enough to be picked up the sPCA analysis.

This picture is also reflected in Fig. 5, which shows that the variability in the Nccn variables is dominated by LV2 and not LV12.

To answer the reviewers question we have added the following brief summary on line 700:

"Since SSA particles contain sea salt they are hygroscopic and efficient CCN. Therefore, it is interesting to note that all of the CCN OVs are absent in LV12. The absence can be explained by recent studies that suggest that, on average, SSA particles only form a minor fraction of the total marine CCN budget (Modini et al., 2015; Quinn et al., 2017; Schmale et al., 2019), which instead appears to be dominated by accumulation mode non-sea-salt sulfate aerosols (e.g. see discussion of LV2 in Appendix A)."

References: De Leeuw et al. (2011), doi: 10.1029/2010rg000349; Monahan et al. 1986, doi: https://doi.org/10.1007/978-94-009-4668-2_16, Modini et al. (2015), doi: 10.1002/2014JD022963, Quinn et al. (2017), doi: 10.1038/ngeo3003, Schmale et al. (2019), doi: 10.1175/bams-d-18-0187.1

**RC2.12:** Page 41, line 713: please be more specific than 'The relatively large size of airborne SSA droplets' since particles much larger than 2 or 3 um do not scatter as efficiently at visible wavelengths than particles between 0.2 and 2 um.

**AC2.12:** Thanks for pointing out this lack of clarity. We rephrased this sentence to indicate the specific size range of SSA particles that we were referring to in l. 713 :

"The size distributions of dried SSA particles peak at diameters of around 0.2 µm and therefore contain substantial contributions from particles with diameters in the range from ~0.1 to 1 µm (Prather et al., 2013). The strong contribution to this size range means that SSA particles are effective at scattering solar radiation and thereby reducing visibility through the atmosphere."

**RC2.13:** FVFM is defined line 1664**: ''FVFM** is the maximum photochemical efficiency of photosystem II' and used line 738 without definition.

**AC2.13:** Thank you for spotting this. We have added the definition in line 738 and removed it from l. 1664.

**RC2.14:** Lines 762-764: explain for the non-specialist what to look for in Figure 5: " Bacterial abundance has a relatively high negative contribution to LV11 (see Figure 5), as bacterial concentrations are linked to the availability of dissolved organic matter (a product of particulate organic matter including POC and PON) and nutrients (Church et al., 2000; Kirchman et al., 2009)."

**AC2.14:** We apologize, the reference should point to Figure 16. We have corrected it accordingly.

**RC2.16:** Page 54, line 989: You wrote "strong precipitation even", did you mean "strong precipitation event"?

**AC2.16:** Yes, this has been corrected to "event".

---

## Author Comment (AC3)

**Summary of Revisions**

RC = Reviewer comment

We thank all three reviewers for their positive and constructive feedback. In order to provide a quick overview of the changes to the to-be-revised manuscript, we give a summary here:

- The title has been changed to: "Exploring the ocean and atmosphere coupled system with a data science approach applied to observations from the Antarctic Circumnavigation Expedition" (following RC3.3).
- We have added research questions in the introduction for a framework that better structures the manuscript as a whole (following RC1.6).
- The methods description has been revised substantially to make the language more accessible to non-data scientists (following the general and several targeted comments of Reviewer #1).
- Section 5 (description of individual LVs) will be moved to a new appendix A to substantially shorten the manuscript. We now summarize the outcome of all LVs briefly in a revised section 4.1, and highlight the novel aspects we found there as well. We give give one condensed description of LV9 as example in a revised section 4.2.  (following RC1.7, 3.1, 3.4)

RC = Reviewer Comment, AC = Author Comment, new suggested text in blue

**Answer to Reviewer 3**

Anonymous Referee #3, 02 Jun 2021

General comments:

**RC3.1:** I find this paper uses an interesting approach that has a potentially high value and high impact for the ocean-atmosphere interdisciplinary research community.  The paper takes the observations from the Antarctic Circumnavigation Expedition (ACE, austral summer 2016/2017) cruise and combines them with a sparse Principal Component Analysis (sPCA) to understand how different observed variables are linked together and to the general context (e.g. distance from land, cyclone activity, etc.).  The paper is also very long, which makes reading and understanding the entire content of the paper and really getting into the new conclusions that result from this study extremely difficult.

I support this paper as a proof of concept for this approach, but I find the science questions posed (or hypotheses) and conclusions in the study are very weak. This paper should be published after the comments from the other reviewers and the comments below are addressed.

**AC3.1:** We thank the reviewer for the positive and constructive feedback. We appreciate the reviewer's remark that the approach presented in our paper might be a valuable addition to the way that our community analyses large, heterogeneous data sets. We also agree that the paper is very long, which might be a disadvantage to clearly communicate our message. Therefore, we have taken multiple measures (see summary of revisions and detailed responses) to shorten the paper (e.g. moved the individual presentation of all 14 LVs to the appendix A). In addition, we have taken an effort to more clearly state the science questions (see AC3.2 and AC1.6) and made a dedicated effort to clearly state the novel results (in the new section 4.1, see AC3.4 and AC1.7) and the advantages of the sPCA (in the new section 3.5, see AC1.2). We hope that these changes will help to better bring across our key messages. Our detailed responses are provided below.

**Major comments:**

**RC3.2:** Most of the conclusions made using this very complex analysis are simplified statements of well known phenomena. So, I'm not sure what is the added value of this approach compared to what is already known. This is seen in the various "In summary" statements that come at the end of each section that focuses on the latent variables (LVs). This is seen most clearly in the summary for LV7 and LV10, which mostly put things into a seasonal and diurnal cycle context. I do not see what we have learned by using this "data science" approach. One way to address this would be to acknowledge in the abstract and very early in the study that there are no main scientific conclusions using data science in this study, but that this sets up the methodology that can be used in the future for this purpose.

**AC3.2:** We agree that the value of this manuscript lies first and foremost in setting up the method for future studies, which were designed a priori around interdisciplinary research questions. The sPCA fills an important gap in this regard, because it is more powerful than simple correlation analysis, and it allows to relate a large number of variables, which reflect processes of different time scales and at a level of detail that comprehensive Earth System Models cannot address.

One of the key aspects of this analysis is that it provides the possibility for an untargeted, and therefore more objective and unbiased, analysis, whereas traditional methods are often biased by a certain method that is tailored to a specific question. We should have pointed this aspect out more clearly and also more clearly state which of the results are novel and which are well known aspects.

We now highlight these new aspects that the analysis was able to depict in section 4.1, the abstract, and conclusions. They include:

- New insights into the Southern Ocean water cycle, where surprisingly, our large-scale assessment of concurrent precipitation and salinity measurements does not yield a direct response of the surface ocean salinity to precipitation events. Instead, we here show that variations in surface ocean salinity are driven by the climatological (long-term) patterns set by surface freshwater fluxes integrated over time-scales longer than synoptic events (LV1) and seasonal melting on sea ice (LV9).
- We also find a latitudinal distribution of the nutrient availability and its effect on the productivity, which is highlighted in LV11, LV6 and LV8. This shows, at the largest scale ever reported, nutrient limitation regimes for the subantarctic front, south of the polar front and associated with the island mass effect as previously reported.
- The sPCA produced unexpected results for some of the reactive trace gases, notably isoprene (LV7). This result points towards a complex interplay between the seasonality of emissions (sources) and seasonality of oxidation pathways (sinks), which, coupled with the potential effect of transport from terrestrial sources, paint a very complex picture for atmospheric isoprene in the Southern Ocean.

We also think that this contribution has provided a valuable overview of Southern Ocean processes on different time and spatial scales. In addition, the published datasets are a benchmark for the current state of the Southern Ocean, against which data in several years or decades time can be compared. We have added the following to the abstract in l. 11ff:

"Our results provide a proof of concept that sPCA with uncertainty analysis is able to identify temporal patterns from diurnal to seasonal cycles, as well as geographical gradients and "hotspots" of interaction between environmental compartments. While confirming many well known processes, our analysis provides novel insights into the Southern Ocean water cycle (freshwater fluxes), trace gases (interplay between seasonality, sources and sinks), and microbial community (nutrient limitation and mass island effects at the largest scale ever reported). Our results establish…"

And in l. 24:

"It thereby fills an important gap between simple correlation analyses and complex Earth System Models. The former would not be able to relate such a large number of variables, while the latter is less constrained by observations and comes with analytical challenges to depict single processes."

And in the introduction in l.42 (This addition in l. 42 is also a response to reviewer comment RC1.6):

"To explore interactions between the Southern Ocean system components, we apply an unsupervised learning method, sparse principal component analysis (sPCA). Application of the sPCA has two objectives: i) conducting an untargeted and therefore more objective analysis of data, where the method is less tailored to the science question as compared to more traditional regression analysis, and ii) to target a set of specific research questions (RQ):

RQ1: Is sparse principal component analysis an adequate tool to extract interaction processes inherent to a heterogeneous and short data set, which describes environmental variability?

RQ2: Is it possible to identify geographic locations ("hotspots") that are common to several interaction processes?

RQ3: Which are the key observed environmental variables that strongly contribute to several interaction processes?

Specific answers to RQ1 are given in section 3.5, with respect to model limitations and advantages, and 6.2, with respect to interaction processes. RQ2 is answered in section 6.1 and RQ3 in section 6.3. Note that we focus on the proof of concept of the sparse principal component method by basing the interpretation primarily on the known processes of the Southern Ocean climate system. New scientific insights from this novel approach are described in section 4.1.

Just as a point of clarification, the summaries at the end of each section 5.x are meant for the quick reader to grasp the essence. There are more interesting and potentially novel details in the descriptions, which can inspire researchers with an interest in the specific processes to explore those further. And of course, in a way the temporal patterns in LV7 and LV10 are trivial, but thinking this the other way around, it would not be a good sign if LV7 and LV10 did not feature, because this is an obvious performance check.

**RC3.3:** The paper should be re-titled to more clearly reflect the paper content. The paper focuses on all of the aspects of the ACE cruise, not just biogeochemistry and physics. I would recommend something more general like "Understanding processes observed in the southern ocean-atmosphere system using ACE observations combined with data science".

**AC3.3:** Thank you for the suggestion. We have retitled the paper:

Exploring the ocean and atmosphere coupled system with a data science approach applied to observations from the Antarctic Circumnavigation Expedition

We spell out ACE, because there was another cruise a couple of decades ago in the Southern Ocean called the Aerosol Characterization Experiment (ACE).

**RC3.4:** I recommend that the authors work on shortening the paper by moving some of the very lengthy discussion into supplementary materials or into an annex to make this paper more readable. I would like the authors to get to the point of what was learned in addition to what is already known more quickly.

**AC3.4:** We appreciate the reviewer's point of view and suggest the following: Section 4 has been renamed "Sparse PCA results", Section 4.1 is now "Short summary of all latent variables and new insights". and contains the text here below, which is merged from the

original section 4.1 first paragraph and section 5.8 "Short summary of all latent variables", and contains new additions to highlight the new insights. We also provide a condensed description of LV9 in a new section 4.2 to give one prominent example with new insights. The remainder of section 4 stays in the main manuscript. The manuscript then continues with the former section 6 "Discussion". We highlight the new text in blue.

This is the new section 4.1:

[revised manuscript text omitted]

**RC3.5:** The authors should discuss how different timescales of processes that occur in nature that control the observed variables that were seen as a snapshot in space and time on the ship. Is it fair to group things into a data science approach variables that are observed in the atmosphere, ice, and ocean that have very different lifetimes and controlling factors that may not be co-located (i.e. relating them in the same space and time may give the wrong correlations/dependencies compared to what happens in nature)?

**AC3.5:** This is an important question and has been addressed in section 3.5 "Model limitations and advantages", and section 6.2 "Atmosphere-ocean interactions". The two main limitations we highlight are:

a) There is no underlying temporal model, meaning two observations sampled within a short period of time are more related than two observations sampled within a longer period of time - one example is the lack of observation of the relation between dissolved DMS and aerosol MSA. The sPCA does not model time, and therefore lags and nonlinear temporal effects between measurements are not taken into account. However, note that we perform an independent temporal resampling prior to sPCA (as preprocessing) in order to homogenize temporal resolution. This comes with the drawback of potentially increasing relatedness between measurements acquired within the resampling time window, but also has the benefit of increasing the temporal correlation of each OV.

b) The strict linearity means non-linear process cannot be considered. We highlight the key observations on time scale relations from section 6.2 here below. In essence, we find if processes happen on sufficiently different time scales, sPCA succeeds in not relating them, as their covariance is usually low. To prevent solutions driven by noise and spurious correlations also along different spatio-temporal scales, we introduce the use of bootstrapping, which allows us to focus only on significant relationships.

Here below follow excerpts from the manuscript that address these points:

L. 1023: *In most LVs, we find a coinciding activation of variables in the Atmospheric dynamics and thermodynamics and in the Oceanic dynamics and thermodynamics category, which are related to local coupling of wind and waves, larger-scale variations of air and water temperature, and characteristics of the ocean currents. These LVs only activate OVs from the Atmospheric dynamics and thermodynamics category, but not from the Oceanic dynamics and thermodynamics. One possible explanation for the absence of a clear influence on the ocean is that the precipitation (LV4) and the diurnal cycle (LV10) represent strong variation of atmospheric OVs on time scales of less than a day, which might be too short to trigger considerable oceanic variability of detectable strength.*

L. 1033: *Links between ocean and atmosphere are visible for LVs with a strong low-frequency (> 1month) component like the climatic zones (LV1; Figure 22a), the seasonal signal (LV7; Figure 22g), and intermediate frequencies (in the order of days) such as sea ice cover (LV9; Figure 22i), and cyclone activity (LV13; Figure 22m). LVs which happen*

*on short time scales, for example strong precipitation related variations of LV4, trigger only a weak (w < 1) marine reaction…*

*L. 1049: The above observations show that our analysis targets processes that manifest themselves in rather local correlations, such as the established link between wind speed and sea state or correlations based on smooth variations over time and space, such as the large-scale horizontal gradients in the air and sea water temperature and the hydrological cycle. To include processes occurring with a time lag or those affected by transport across larger scales, the coupling with air mass back trajectory analysis*

*provides a valuable extension to infer potential relations of the observed signals with up-wind conditions and air mass history, for example the advection of cold or warm air (see section 5.2.1),...*

**RC3.6:** How do non-local processes get integrated into this approach?  This is not currently clear for me.

**AC3.6:** This is a good point. The method succeeds by itself in including several large-scale and longer temporal features: climatic zones and large-scale horizontal gradients are represented by LV1 and LV14, large-scale weather systems feature in LV13, LV7 highlights seasonal patterns. In addition to that, we have included back trajectory analyses to understand how in situ observations carry signatures of air mass history (i.e. larger spatial and temporal extent). This was somewhat addressed in the discussion section in  l. 1051. Following the reviewer's comment, we have made the formulation more explicit (l. 1051):

To better understand the ability of the sPCA to capture non-local processes occurring with a time lag or those affected by transport across larger scales, we analyse air mass back trajectories. This analysis provides a valuable extension to infer potential relations of the observed signals with up-wind conditions and air mass history. Two examples are the advection of cold or warm air (see LV3 - Meridional cold and warm air advection, Appendix A), and the removal of accumulation mode aerosols during successive precipitation events (see LV2 - Drivers of the cloud condensation nuclei population, Appendix A)."

**RC3.7:** The authors should expand their discussion of missing data and the influence this has on their analysis (as noted by reviewer 1).

**AC3.7:** To answer this question, we have to distinguish between two types of *missing data:* Data measured by sensors but filtered or dropped due to quality control or sensor failures, and data that is missing, because their temporal resolution is too low. For the former, we deal with them explicitly by using imputation strategies and temporal

averaging to cope with uneven sampling and spurious missing data. The temporal interval of imputation has been selected based on the overall OV temporal resolution, and selected by comparing different strategies. We describe this in detail in Sections 3.3 and 3.4.

For the second category, i.e. data that has been sampled less frequently than our temporal resampling interval of 3 hours, we perform iterative imputation by sPCA model inversion. We employ this strategy in order to provide continuous LVs along the temporal dimension, but we cannot verify the quality of this imputation strategy or identify variations on time scales shorter than the actual sampling frequency. For this reason, OVs with very low temporal resolution have a lower number of datapoints, which affects the strength of the correlations between different OVs. In addition, such missing values generally reduce the significance of the results after bootstrapping, which tends to assign larger standard deviations and lower median weights to sparser OVs. It results that OVs with lower correlations are generally discarded by the sPCA. The lower significance and correlations, result in the tendency of assigning lower importance of sparsely measured OVs for the corresponding LVs. For example, the mixed-layer depth is only derived from the relatively sparse CTD and XBT profile locations and therefore has a much lower temporal resolution compared to the other OVs in our data set. As a consequence, it appears to be less important for air-sea exchange processes and biological production in our results as one might expect. This issue is a clear limitation of our study that is important to consider when interpreting results.

Future work can be devoted to the inclusion of OVs temporal models within a sPCA like strategy, in order to better estimate the contribution of missing data of the two types described above. Ideally, imputation will not only be made based on linear dependencies between the input OVs, but also accounting temporal co-variations, potentially providing more robust decomposition solutions with respect to gaps in measurements.

Proposed manuscript changes:

- New sentences, L266: "The data filling performed at the preprocessing step is complementary to the data imputation performed by sPCA. While the former is an independent data filling based on temporal averages, the latter can be seen as an estimation based on inverting the sPCA model on missing data, corresponding to a regression from non-missing OVs. The more correlated the OVs to the one containing a missing data point to be estimated, the better the estimation. The lower significance and correlations, result in the tendency of assigning lower importance of sparsely measured OVs for the corresponding LVs. For example, the mixed-layer depth is only derived from the relatively sparse water column profile locations and therefore has a much lower temporal resolution compared to the other OVs in our data set. As a consequence, it appears to be less important for air-sea exchange processes and biological production in our results as one might expect. This issue is a clear limitation of our study that is important to consider when interpreting results. "

**Minor comments:**

**RC3.8:** There are a few small typos as noted by reviewer 2. I suggest a careful re-reading before publication.

**AC3.8:** Thank you for pointing this out. We have corrected all typos we found.